# A MEC-2/stomatin condensate liquid-to-solid phase transition controls neuronal mechanotransduction during touch sensing

Neus Sanfeliu-Cerdán [1], Frederic Català-Castro[1], Borja Mateos[2],
Carla Garcia-Cabau [2], Maria Ribera[2], Iris Ruider[1],
Montserrat Porta-de-la-Riva [1], Adrià Canals-Calderón[2], Stefan Wieser [1],
Xavier Salvatella [2,3] ✉ & Michael Krieg [1] ✉

A growing body of work suggests that the material properties of biomolecular condensates ensuing from liquid–liquid phase separation change with time. How this aging process is controlled and whether the condensates with distinct material properties can have different biological functions is currently unknown. Using *Caenorhabditis elegans* as a model, we show that MEC-2/stomatin undergoes a rigidity phase transition from fluid-like to solid-like condensates that facilitate transport and mechanotransduction, respectively. This switch is triggered by the interaction between the SH3 domain of UNC-89 (titin/obscurin) and MEC-2. We suggest that this rigidity phase transition has a physiological role in frequency-dependent force transmission in mechanosensitive neurons during body wall touch. Our data demonstrate a function for the liquid and solid phases of MEC-2/stomatin condensates in facilitating transport or mechanotransduction, and a previously unidentified role for titin homologues in neurons.

Recent studies have shown that proteins with intrinsically disordered domains can separate from the cytoplasm or nucleoplasm to form liquid-like condensates in a process akin to phase separation[1–3]. These liquid-like properties are usually transient and the condensates can undergo a transition to viscoelastic solids[4,5]. Such rigidity transitions occur in proteins that form amyloid fibres[6,7] and, possibly, drive neurodegeneration[8,9]. Not all rigidity transitions are pathological, however, because they endow biomolecular condensates with inducible material properties that might be physiologically important. For example, simple fluids cannot sustain mechanical forces, and it is plausible that liquid-to-solid transitions occur during mechanotransduction[10,11]. Whether or not the distinct liquid and solid phases have particular functions in the same cell is still unknown.

Stomatins are highly conserved membrane-associated scaffolding proteins that are required for membrane mechanics[12], modulate ion channel activity[13–15] and contain a characteristic stomatin domain[16] important for the sense of touch in mouse and *C. elegans*[17–19]. The N and the C termini of *C. elegans* stomatins have regions of low sequence complexity, predicted to be intrinsically disordered (Extended Data Fig. 1)[20]. In MEC-2, the C-terminal domain has been proposed as a gating tether thought to modulate mechanosensitive ion channel open probability by an elusive protein–protein interaction[21–23]. Here, we use a combination of genetic and biophysical experiments to show that MEC-2/stomatin binds to the SH3 domain of UNC-89 (titin/obscurin) and induces a liquid-to-solid phase transition with consequences in neuronal mechanotransduction during touch.

[1]ICFO - Institut de Ciencies Fotoniques, The Barcelona Institute of Science and Technology, Castelldefels (Barcelona), Spain. [2]Institute for Research in Biomedicine, The Barcelona Institute of Science and Technology, Barcelona, Spain. [3]ICREA, Barcelona, Spain. ✉e-mail: xavier.salvatella@irbbarcelona.org; michael.krieg@icfo.eu

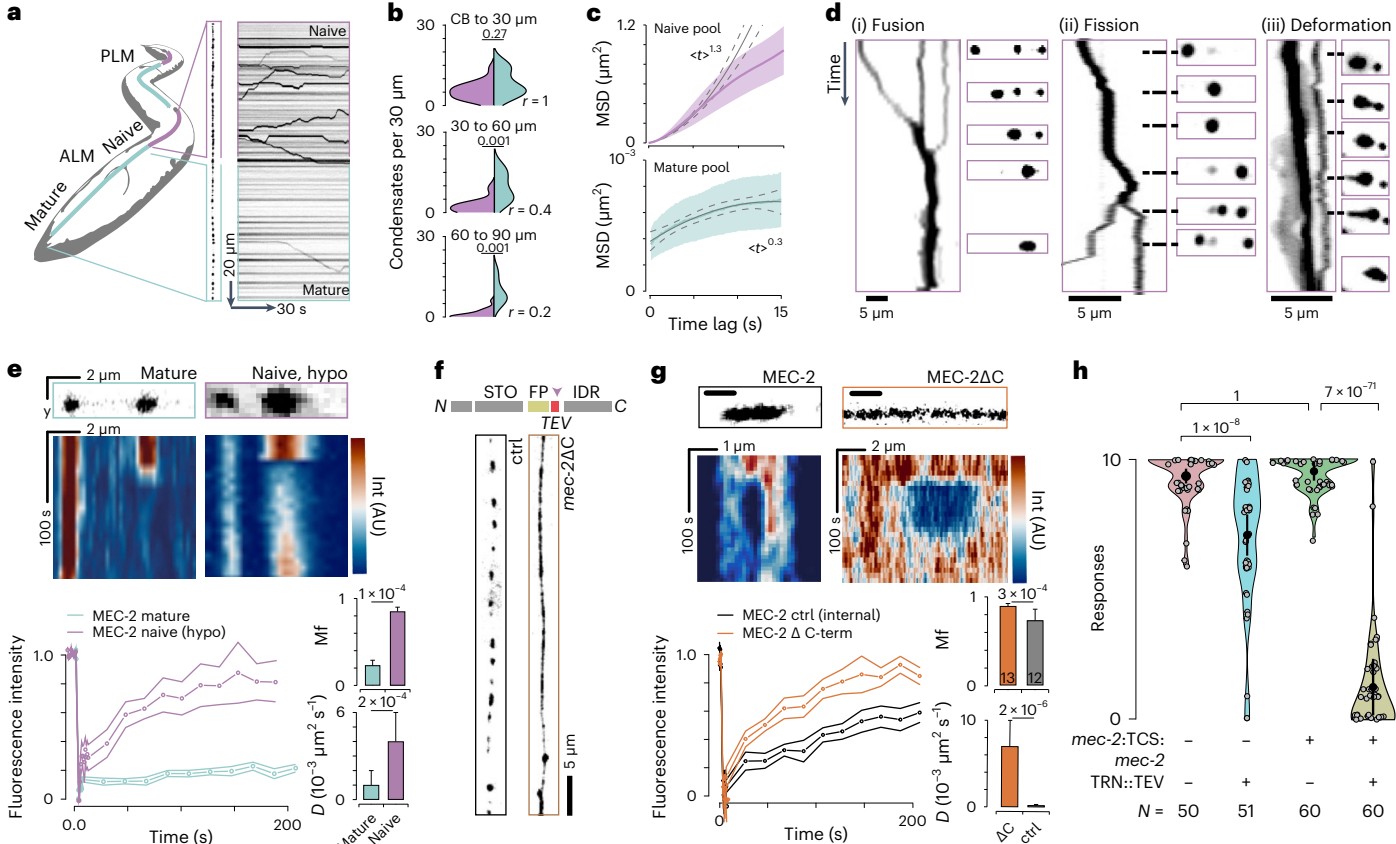

**Fig. 1 | Dynamics of the liquid-to-solid transition. a**, Sketch indicating the location of MEC-2 pools within an anterior (ALM) touch receptor neuron. The posterior (PLM) neuron is sketched for completeness. Representative image and kymograph of MEC-2 along the whole TR neurite. **b**, Fraction of mobile (purple) versus immobile (green) puncta in three different distances respect to the cell body (CB): CB–30 μm, 30–60 μm and 60–90 μm and ratio (r) of mobile/immobile populations. N = 19 TRNs from different individual animals pooled over four independent experiments. P-value derived from a one-sided Tukey HSD (honestly significant difference) test. **c**, Mean square displacement (MSD) versus time lag for MEC-2 in the naive and the mature fractions. The exponent of $<x>^n$ indicates the best fit to a power law. Error band indicates mean ± s.d. **d**, Representative examples of MEC-2 condensates in the naive fraction of the TRN (labelled in purple in the scheme): (i) fusion, (ii) fission and (iii) deformation events plotted in a kymograph and snapshots during the course of the videos. Scale bar, 5 μm; duration of the kymograph: t = 5, 13 and 22 s, respectively. Size of the magnified inset: 20 μm, 10 μm and 4 μm, respectively. **e**, Representative kymographs (top) and the FRAP dynamics (bottom) of *mec-2*::mCherry in the mature pool of TRNs (green, N = 10 condensates from 5 individual animals from 1 experiment) and *mec-2*::mCherry in the naive pool expressed in hypodermal cells (purple, N = 10 condensates from 5 individual animals over 4 independent experiments). Mean ± s.d. Right, mobile fraction (Mf) and diffusion coefficient (D) derived

from an exponential fit to the experimental recovery curves on the left. P-value derived from two-sided t-test. **f**, Schematic of the construct and representative images of MEC-2 distribution in TRNs of a *mec-2*::TSMod::TEV::*mec-2* animal (see scheme of the protein domains) in absence (left) and presence (right) of TEV protease. STO, stomatin domain; FP, fluorescent protein; TEV, tobacco etch virus cleavage site; IDR, intrinsically disordered region. See quantification of interpunctum distances in Extended Data Fig. 6. Scale bar, 5 μm for both images. **g**, Representative still images and kymographs (top). Scale bar is the same for still image and kymograph. Colour scale indicates fluorescence intensity. FRAP dynamics (bottom) for full-length MEC-2 (inside mature puncta, black, N = 12 condensates from 4 individual animals over 2 independent experiments and conditionally truncated MEC-2 (orange, N = 13 condensates from 8 individual animals over 2 independent experiments). Mean ± s.d. At the right, Mf and diffusion coefficient derived from an exponential fit to the curves at the left. P-value derived from two-sided t-test. **h**, Violin plot of the body touch response derived from wild-type and conditionally truncated animals in absence or presence of TEV protease, and wild-type animals with only TEV protease expression. TCS, TEV cleavage site. Circle indicates mean, black vertical bar indicates the 95% CI on the mean. N, number of different animals tested over three independent experiments. P-value derived from a two-sided t-test with Bonferroni adjustment for multiple comparison.

## Results

### MEC-2/stomatin forms two distinct condensates in vivo

First, we created transgenic animals carrying a single copy of fluorescently labelled MEC-2 and investigated its dynamics in the touch receptor neurons (TRNs; Supplementary Video 1) of immobilized animals. Like others before, we found that MEC-2 was distributed in discrete puncta along the sensory neurite[24,25] (Fig. 1a–c). We observed, however, two distinct populations within the TRN: static immobile puncta all along the neurite, but also larger fluorescent puncta close to the cell body, which rapidly trafficked towards the distal neurite with high processivity and occasional reversals, indicative of motor-driven, directed transport on short timescales (Supplementary Video 1 and Fig. 1a–c).

We also found that the number of mobile MEC-2 puncta increased with developmental age and hence neurite length (Extended Data Fig. 2a), suggesting that the mobile pool facilitates the directed transport of MEC-2 into longer neurites. In all developmental stages we also found MEC-2 clusters in the cell body, which were larger and more mobile than those found in neurites (Extended Data Fig. 2b). Based on their motility behaviour and predominant location, we termed the two pools mature and naive.

Interestingly, the clusters of the naive pool were larger and more spherical than those of the mature pool (Extended Data Fig. 2c), with a behaviour reminiscent of liquid-like droplets[3]. We observed fusion and fission events and deformation of single puncta, similar to

observations made in other systems[26] (Fig. 1d, Extended Data Fig. 2d,e and Supplementary Videos 2–4). Thus, we hypothesized that MEC-2 exists as phase-separated condensates with spatially distinct material properties.

To establish that MEC-2 forms different condensates along neurites in vivo, we compared the behaviour of the mature and naive populations. Because the mobility of condensates in the naive pool of TRNs precludes fluorescence recovery after photobleaching (FRAP) experiments, we modelled the naive pool in a heterologous tissue and expressed MEC-2 in hypodermal cells, where these condensates remain primarily static and TRN-specific genes such as *mec-4* and neuronal microtubules are not expressed. We then performed FRAP on individual condensates of the neuronal mature pool and the naive condensates in hypodermal cells (Fig. 1e). We observed that the MEC-2 in the hypodermal cells recovered quickly, whereas the mature, immobile pool in TRNs did not recover during the course of the experiment. This suggests that MEC-2 in naive condensates is more fluid than in mature ones (Fig. 1e). Importantly, these results were independent on the position on the neurite (Extended Data Fig. 3a–d).

### The MEC-2 C-terminal domain drives condensate formation

Because biomolecular condensation is often driven by phase transitions of proteins with intrinsically disordered regions (IDRs) we asked if the MEC-2 C terminus drives condensate formation. We engineered a fluorescent protein followed by a proteolytic tobacco etch virus (TEV) cleavage site into MEC-2 between the stomatin domain and the C terminus (Fig. 1f, arrow). In the absence of TEV protease this construct formed condensates along the sensory neurite, indistinguishably from wild-type MEC-2. However, when we co-expressed the TEV protease in TRNs, the labelled N-terminal MEC-2 fragment formed a continuous phase, suggesting that the intrinsically disordered C-terminal domain is the driver of MEC-2 condensation (Fig. 1f). Next, we compared the behaviour in FRAP experiments of the C-terminally truncated MEC-2 with that of the mature condensates formed by full-length MEC-2 in the distal neurite. Whereas the cleaved MEC-2 lacking the C-terminal domain recovered quickly and completely after photobleaching, the condensates formed by intact MEC-2 recovered incompletely over the course of the experiments (Fig. 1g and Supplementary Video 5). The propensity of MEC-2 to phase separate into distinct condensates is important for its function in sensing mechanical touch as the conditional C-terminally truncated MEC-2 was unable to transduce external touch into avoidance behaviour (Fig. 1h); importantly, we made the same observations with a constitutive truncated MEC-2 construct lacking the C-terminal domain (Extended Data Fig. 3e–i).

Protein structure prediction algorithms such as AlphaFold2 (ref. 27) did not propose a well-defined three-dimensional structure for the MEC-2 C-terminal domain (low predicted local distance difference test, pLDDT, confidence values), which is predicted to undergo liquid–liquid phase separation (LLPS)[28] (Fig. 2a, phase separation index[29]). To confirm this, we expressed and purified the MEC-2 C-terminal domain (residues 371–481) and studied its structural and phase separation properties in vitro. As expected for an intrinsically disordered protein, the nuclear magnetic resonance (NMR) $^1$H–$^{15}$N correlation spectrum had low $^1$H chemical shift dispersion (Fig. 2b) and the main chain chemical shifts largely corresponded to those of a statistical coil (Fig. 2c)[30]. The intensities of the NMR signals across the sequence of the C-terminal domain were however non-uniform: those corresponding to the region $^{382}$KKIRSCCLYKY$^{392}$ appeared broad beyond detection (Fig. 2b,c). This behaviour has been observed in IDRs that can undergo LLPS, where low intensities help identify the regions of sequence involved in the transient interactions that stabilize the phase-separated state[31,32].

Next, we tested the phase separation properties of the MEC-2 C-terminal domain and indeed observed that it forms liquid droplets (Fig. 2d(i) and Extended Data Fig. 4a,b) upon heating (Fig. 2e(i) and Extended Data Fig. 4e) and at high ionic strength (Extended Data

Fig. 4c), suggesting that hydrophobic interactions contribute to driving the phase transition. The droplets fused and their fluorescence quickly recovered in FRAP experiments, confirming their liquid character (Fig. 2d(ii),e(ii) and Extended Data Fig. 4f and Supplementary Video 6); these results are qualitatively equivalent to those observed for the condensates formed by the full-length protein in vivo (Fig. 1e).

### A proline-rich motif is critical for mechanotransduction

Several MEC-2 point mutations cause a strong touch defect without affecting the trafficking of MEC-2 into the sensory neurite of TRNs[24]. One of these alleles, *u26*, encodes an arginine-to-histidine substitution at position 385 of its C-terminal (R385H) IDR, in the vicinity of a proline-rich motif (PRM) with the consensus PPxxP sequence reminiscent of SH3-binding domains[33,34] (Fig. 2c,f): animals carrying this mutation did not display a behavioural response to touch (Fig. 2g). We also overexpressed a peptide derived from the MEC-2 C-terminus encompassing wild-type and mutant MEC-2 (Methods) and found that the wild-type PRM competitively interfered with the touch response, whereas the mutant PRM domain did not (Extended Data Fig. 4k). To visualize mechanoreceptor activity, we immobilized individual animals in a microfluidic device designed to apply precise mechanical waveforms to the cuticle while monitoring calcium transients in single neurons using GCaMP6s[35]. TRNs did not mobilize visible calcium under slow stimuli or steps, but robustly activated to rapidly oscillating cuticle deformations. TRNs from mec-2(R385H) mutant animals, however, never activated to any stimuli (Fig. 2h–j and Supplementary Video 7) despite the fact that mutant MEC-2 (R385H) protein has superficially wild-type dynamics and still colocalized with MEC-4, the pore-forming subunit of the MeT channel (Extended Data Fig. 4d). As we observed for the wild-type MEC-2, the mutant also has a naive and mature pools with different FRAP recovery dynamics (Extended Data Fig. 4g–j). However, we observed that the mutant MEC-2 exhibited a higher mobile fraction (Mf) in both the mature and naive populations compared with the wild-type MEC-2 (Extended Data Fig. 4h, j), even though the diffusion coefficient derived from the recovery curves of mature puncta formed by the mutant MEC-2 remained unchanged compared with those formed by the wild type (Extended Data Fig. 4i,j), which mirrored the findings obtained in the in vitro experiments (Extended Data Fig. 4f).

To determine how a point mutation leads to a loss of function without detectable gross defects in trafficking and axonal localization, we expressed the R385H mutant MEC-2 C-terminal domain and studied its structural propensities and LLPS behaviour in vitro. The NMR spectra of the R385H mutant displayed shifted peaks and changes in intensity relative to wild-type MEC-2 (Extended Data Fig. 4l–o), suggesting that the mutation affects MEC-2 homotypic interactions. We found, however, that the R385H mutation did not alter the phase separation of MEC-2 (Extended Data Fig. 4a–c,e,f). We thus concluded that the mutation is unlikely to lead to the observed touch defect by changing the phase separation capacity of the MEC-2 C-terminal domain.

### MEC-2 forms UNC-89 co-condensates

Based on the stereotypic PRM present in the C-terminal part, we hypothesized that MEC-2 interacts with an SH3 domain[36]. We thus performed a TRN-specific RNAi feeding experiment (Fig. 3a, inset)[37] to knock-down 35 of 41 proteins with an SH3-domain[38] that are reportedly expressed in TRNs of *C. elegans*[39,40] (Supplementary Data Table 1) and available in the feeding RNAi libraries[41,42]. Culturing these animals on bacteria expressing *mec-4* or *mec-2* RNAi constructs[43,39], but not the empty vector, lead to a significantly decreased touch response (Fig. 3a). Surprisingly, we found that only the knock-down of *unc-89*, a member of the titin family (Fig. 3a and Extended Data Fig. 5a)[40,41], gave a robust reduction in the response to touch.

We then confirmed *unc-89* expression in neurons and found noticeable expression in TRNs, motor neurons and some neurons in the head and the tail, apart from the previously described expression

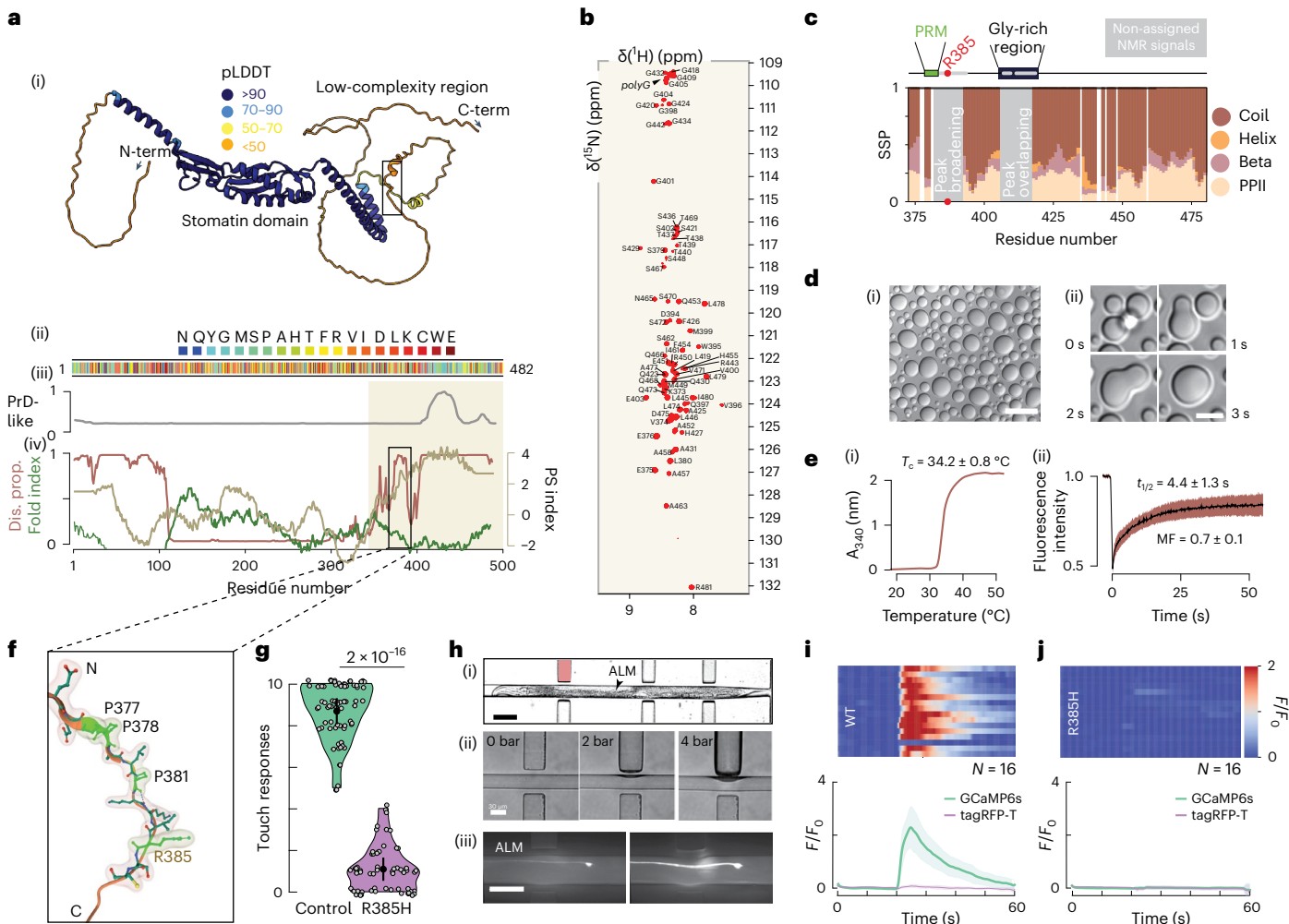

**Fig. 2 | MEC-2 forms biomolecular condensates and is critical for the touch response. a**, (i) AlphaFold2 prediction of MEC-2/stomatin. Low-confidence pLDDT values < 50 suggest the presence of disorder in the N- and the C-terminal regions. The black box indicates the location of the PRM. (ii) Colour-coded primary sequence of MEC-2 visualizes clustering of residues implicated in the formation of biomolecular condensates. (iii,iv) Prediction of sequences involved in oligomerization or amyloid formation (iii) and phase separation index (right axis, beige), disorder property calculation (left axis, red) and fold index (left axis, olive) (iv). The beige area highlights the C-terminal domain used throughout this study. **b**, 2D NMR $^1$H–$^{15}$N correlation spectrum of MEC-2 C-terminal domain, shown as the shaded area in **a**, (iv). **c**, Secondary structure propensity of MEC-2 based on NMR chemical shifts. Schematic representation of MEC-2 motifs. Grey boxes represent the non-assignable signals by NMR. **d**, (i) DIC microscopy image of MEC-2 liquid droplets in vitro, of a 400 μM sample with 2 M NaCl at 37 °C. Scale bar, 20 μm. Representative for $N = 3$. (ii) MEC-2 droplets fusion in vitro at the indicated time points from Supplementary Video 4. Scale bar, 5 μm. **e**, (i) Apparent absorbance measurement as a function of temperature of 200 μM MEC-2 with 2 M NaCl. The indicated cloud temperature ($T_c$) value is the mean and standard deviation of three independent measurements. (ii) FRAP experiment

in vitro of 370 μM MEC-2 with 2 M NaCl at 20 °C. Average fluorescence intensity, mean ± SD of $N = 11$ measurements from one experiment. **f**, Localization of the PRM and the conserved arginine at position 385 (R385). AlphaFold2 model with transparent space-filling residues highlighted. **g**, Average touch response for N2 wild-type ($N = 72$ animals) and MEC-2 (R385H) mutants ($N = 60$ animals). The black vertical bar indicates s.e.m. *P*-value derived from one-way Kruskal–Wallis test. **h**, (i) Microfluidic device for mechanical stimulation and calcium imaging holding an immobilized animal. Scale bar, 100 μm. (ii) Pressure sequence for 0, 2 and 4 bar. Scale bar, 30 μm. (iii) Still images of TRN::GCaMP transgenic animals before and after application of a mechanical stimulus inside the chip. Scale bar, 30 μm. Representative image of at least $N = 10$ individuals from 2 independent experiments. **i,j**, Average GCaMP6s and calcium-independent TagRFP-T intensities recorded from TRNs of wild-type (i) and MEC-2 (R385H) (j) mutant animals trapped inside the body wall chip. A 300 kPa 'buzz' stimulus was delivered for 2 s after recording 10 s baseline fluorescence. Top panels show colour-coded normalized fluorescence intensity traces: individual recordings visualized as stacked kymographs (*N*, number of recordings, animals tested during an experimental day). Bottom panels show the average fluorescence intensity, mean ± s.d.

in body wall muscles (Extended Data Fig. 5b and ref. 40). Next, we generated a knockout of the largest SH3-containing isoforms using CRISPR/Cas9 (Extended Data Fig. 5c,d). Even though these knockout animals had a modest reduction in the behavioural response to touch compared with wild-type animals (Extended Data Fig. 5e), they consistently displayed lower calcium activity after mechanical stimulation in the body wall chip[35] relative to controls (Fig. 3b and Extended Data Fig. 5f). To further confirm the cell-specific role of UNC-89, we conditionally deleted *unc-89* through the combination of site-specific

CRE/lox recombination and auxin-induced degradation (AID[42]). Co-expression of a pan-neuronal (Extended Data Fig. 5g) or a TRN-restricted CRE recombinase and TIR ligase after feeding the animals with auxin led to significantly decreased touch response (Fig. 3c) without a detectable deficit in crawling locomotion (Extended Data Fig. 5h,i).

Several components of the MeT channel complex associate into a punctate pattern[24,25] and we asked if the *unc-89* knockout disrupts the punctate distribution of MEC-2 in the distal neurites. Even though the

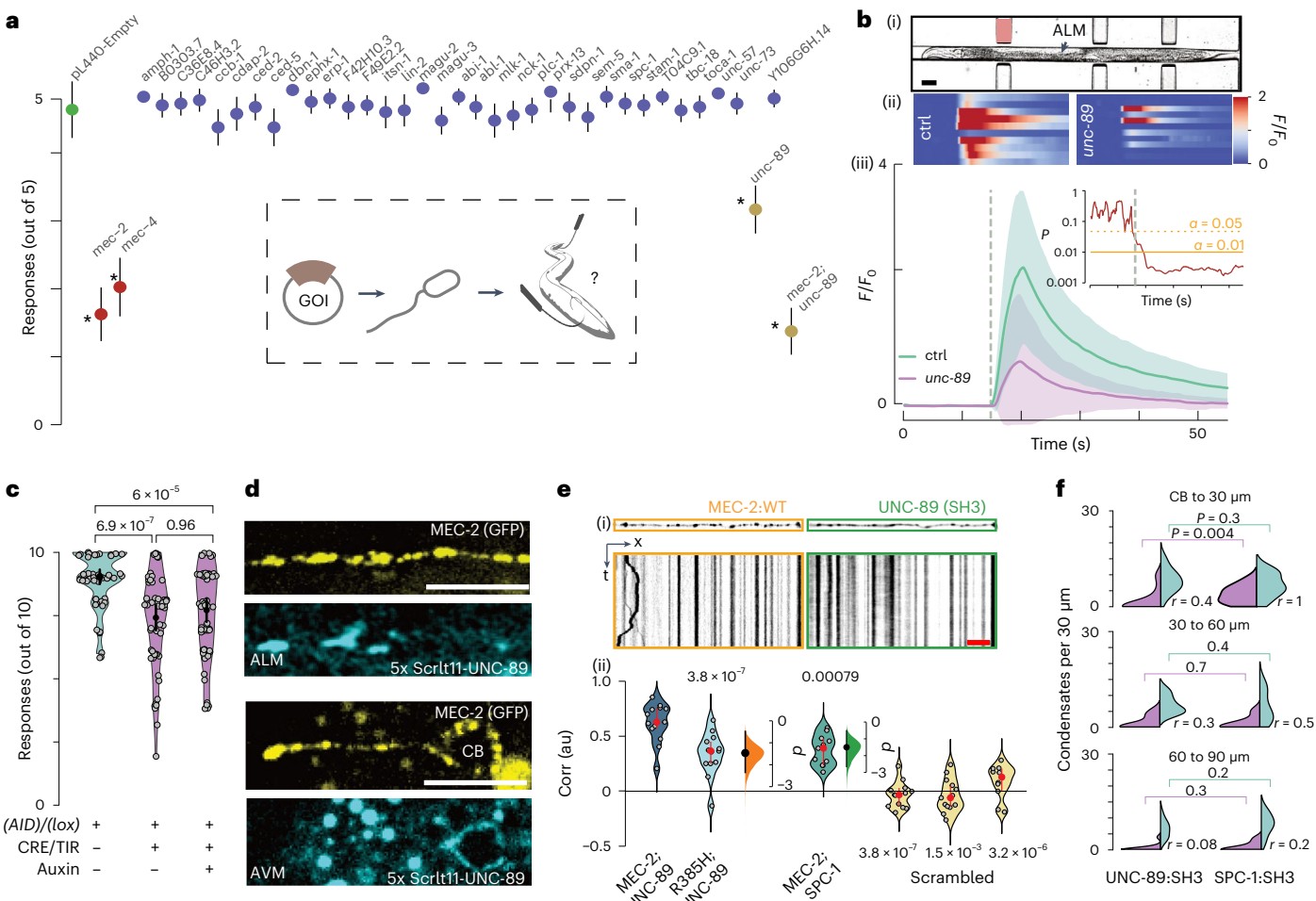

**Fig. 3 | UNC-89 is a component of the mechanoreceptor complex in TRNs.**
**a**, Neuronal feeding RNAi screen with SH3-containing proteins naturally expressed in TRNs (see Supplementary Table 1). Mean ± s.d. of touch responses, $N = 40$ animals; five touches each; pooled from two independent experiments. Asterisks indicate a significant $P$-value compared with the empty vector (negative control) derived from one-way Kruskal–Wallis followed by a Dunn's test for multiple comparison with Bonferroni correction. $P$-values were: $mec-2$: $1 \times 10^{-19}$; $mec-4$: $3 \times 10^{-15}$; $unc-89$: $1 \times 10^{-7}$; $mec-2/unc-89$: $5 \times 10^{-37}$ (Supplementary Table 3). Inset shows the scheme of the experimental pipeline. GOI, gene of interest. **b**,**c**, UNC-89 knockout influence in *C. elegans* touch sensation. **b**(i), Brightfield image of an animal inside the body wall chip. Scale bar, 50 μm. (ii) Stacked kymographs of individual calcium recordings for $N = 10$ wild-type animals from one experiment and 12 $unc-89$ animals from two independent experiments. (iii) Average normalized fluorescence intensity ± s.d. for the GCaMP signal in a wild-type (green) or a UNC-89 KO animal (purple) upon body wall touch inside the microfluidic device[35]. The calcium-independent mtagRFP-T as a control measurement is shown in Extended Data Fig. 5f. Inset shows the $P$-value from a one-sided $t$-test for each time point testing the hypothesis $H_0$: wild type = $unc-89$ with $\alpha = 0.01$. Pressure applied for 2 s at the time indicated by the grey dotted line. **c**, Violin plot showing the body touch response for TRN-specific knockout of $unc-89$ by using double effect of CRE/loxP and AID compared to the AID/loxP-flanked control animals in absence of the $mec-17$p::CRE, the $mec-4$p::TIR or the auxin. Circles indicate mean, vertical bars indicate 95% CI, $N = 60$ animals per condition, each tested 10 times, pooled over 3 independent experiments.

$P$-value derived from a two-sided $t$-test, Bonferroni adjusted for multiple comparison. **d**, Representative images (for $N = 5$ animals) of split wrmScarlet(11) x5:$unc-89$ complemented with TRN-specific $mec-4$p::wrmScarlet1–10 in $mec-2$::GFP background animals. Scale bar, 10 μm (for still image and kymographs). CB, cell body. **e**(i), Kymographs of MEC-2::mCherry and overexpressed UNC-89(SH3)::GFP from the same TRN. Scale bar in red, 10 μm; duration of the kymograph, $t = 60$ s. (ii) Altman–Gardner plot of the correlation between the SH3 domains and the MEC-2 condensates: SH3::GFP domain from UNC-89 and MEC-2::mCherry wild type ($N = 13$ animals) or MEC-2 (R385H) mutant ($N = 13$ animals), and SPC-1 α-spectrin SH3 domain::GFP together with the wild-type MEC-2::mCherry ($N = 11$ animals). Floating axes indicate estimated effect size (based on Cohen's $d$). Correlation coefficients from scrambled MEC-2 pictures (yellow violins) indicates that the correlation is not due to chance (Methods). $p$-value derived from Kruskal–Wallis test. Red circle indicates median ± 95% CI. **f**, Fraction of mobile (purple) versus immobile (green) aggregates in three different distances with respect to the cell body (CB): CB–30 μm, 30–60 μm and 60–90 μm and ratio ($r$) of mobile/immobile populations for two different conditions: animals with overexpression of UNC-89 SH3 domain (left) or SPC-1 SH3 domain (right). $N = 22, 17$ and 14 TRNs in distances 1-2-3, respectively, from UNC-89::SH3 individual animals pooled from four independent experiments. $N = 15, 16$ and 14 TRNs in distances 1-2-3, respectively, from SPC-1::SH3 individual animals pooled from three independent experiments. $P$-value derived from a two-sided Mann–Whitney $U$-test.

overall distribution of MEC-2 in the $unc-89$ knockout is similar to the wild-type animals (Extended Data Fig. 6a–c), we consistently observed that the median interpunctum interval (IPI) between two MEC-2 condensates is significantly smaller ($IPI_{wt} = 3.1$ μm versus $IPI_{unc-89} = 2.1$ μm), suggesting that the absence of UNC-89 interferes with proper localization of these condensates. We also visualized UNC-89 distribution, and

observed a faint signal of endogenously tagged 5xwrmScarlet::UNC-89 in TRNs that partially overlapped with MEC-2 (Fig. 3d).

We then tested if the SH3 domain of UNC-89 co-assembled with MEC-2 along the neurite and expressed the UNC-89 SH3 domain fused to GFP together with mCherry-labelled MEC-2. We observed that UNC-89:SH3, but not another SH3 domain (α-spectrin, SPC-1:SH3),

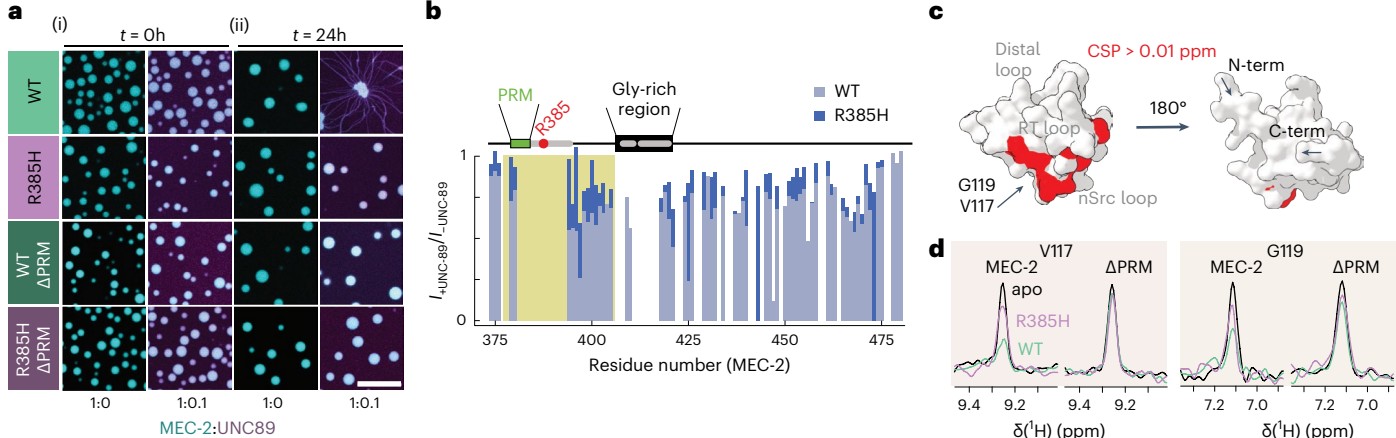

**Fig. 4 | UNC-89 interacts with MEC-2 through its SH3 domain. a**, Confocal fluorescence microscopy images of 200 µM MEC-2 C-terminus (wild-type or R385H mutant) labelled with Alexa Fluor 647 together with UNC-89 SH3 domain labelled with DyLight 488, at a molar ratio of 1:0.1 (MEC-2; MEC-2 (R385H); MEC-2 (ΔPRM); or MEC-2 (ΔPRM,R385H):UNC-89), with 2 M NaCl immediately, $t = 0$ h (i), or after 24 hours (ii) of sample incubation at 37 °C. Scale bar, 20 µm. Representative for $N = 3$ independent measurements. **b**, Intensity ratio between the $^1$H–$^{15}$N NMR spectra of the C-terminal domain of MEC-2 (wild type (light blue) and R385H mutant (blue)) in the presence or the absence of the SH3 domain of UNC-89. **c**, Structural representation of the MEC-2 binding to the UNC-89 SH3 domain, derived from the chemical shift perturbations (CSP) in Extended Data Fig. 7e. Red residues correspond to CSP higher than 0.01 ppm (threshold). The binding is located at the surface between the RT and nSrc loops of the SH3, as previously described for other SH3 domains (this terminology, like the name of the domain itself, refers to features, such as flexible loops of the Src tyrosine kinase). **d**, Binding of MEC-2 (wild-type or ΔPRM) to UNC-89 SH3 domain using V117 and G119 NMR signals as reporters. The green line indicates the presence of MEC-2 wild-type sequence. The purple line indicates the presence of MEC-2 R385H mutant. Black line indicates absence of MEC-2.

sorted into the same condensates and colocalized along the neurite with wild-type MEC-2. On the contrary, UNC-89:SH3 did not colocalize with the MEC-2 R385H mutant (Fig. 3e, Extended Data Fig. 5j). As seen in the kymograph, the colocalization was only observed for the static, immobile condensates, whereas the mobile MEC-2 puncta are not found to colocalize with UNC-89. This suggests that UNC-89 governs the transitions from mobile to immobile MEC-2 condensates. Indeed, we found significantly fewer naive, mobile MEC-2 condensates (compare Fig. 1b with Fig. 3e,f) in TRNs expressing the UNC-89:SH3 protein, and a higher IPI (Extended Data Fig. 6a–c), leading to a lower mature pool compared with wild-type neurons in distal neurites (wild type versus UNC-89::SH3, $P = 0.015$, one-sided $t$-test). This effect appeared to be specific to the UNC-89:SH3 domain, as the SH3 domain of α-spectrin SPC-1 did not interfere with formation of the mobile condensates (Fig. 3f).

Finally, we investigated whether MEC-2 and UNC-89 directly interact and co-phase separate in vitro. We first prepared MEC-2 and UNC-89 SH3 domain (residues 58–128) co-condensates at different molar ratios ranging from 1:0.1 to 1:1 (MEC-2:UNC-89; Extended Data Fig. 7a) and found that UNC-89 partitioned in the droplets formed by both the wild-type and R385H mutant MEC-2 C-terminal domain (Fig. 4a(i)). Similarly, a mutant MEC-2 C-terminal domain in which the proline residues in the PRM preceding the critical position 385 were substituted to alanines ($^{377}$PPxxP$^{381}$→$^{377}$AAxxA$^{381}$; MEC-2ΔPRM) phase separated with the same temperature dependence and co-partitioned with UNC-89 SH3 in the presence and the absence of mutation R385H (Extended Data Fig. 7a,b).

Next we tested the MEC-2/UNC-89 interaction by NMR and observed a weak but consistent intensity reduction in residues in the vicinity of the PRM. This effect was nearly absent in the MEC-2 R385H mutant, indicating that the mutation reduces the strength of the interaction with UNC-89 SH3 (Fig. 4b and Extended Data Fig. 7c–f). As expected for canonical SH3 binding motifs[34], the residues involved in the interaction are found between the RT and nSrc loops of the UNC-89 SH3 domain (Fig. 4c). The interaction depended on the presence of the PRM as we observed a complete abrogation of binding in the MEC-2 ΔPRM mutant (Fig. 4d) and, in vivo, animals carrying the mutant

ΔPRM were completely touch insensitive (0.23 ± 0.56 responses out of 10, mean ± s.d., $N = 60$ animals).

## UNC-89 promotes rigidity maturation of MEC-2 condensates

Finally, we assessed whether UNC-89 partitioning and binding to MEC-2 influenced the maturation process of MEC-2 droplets in vitro. We incubated samples of phase-separated wild-type and mutant MEC-2 (R385H and ΔPRM) in the presence and absence of UNC-89 SH3 (1:0.1) over 24 h and studied the morphology of the formed condensates. Strikingly, the condensates formed by wild-type MEC-2 C-terminal domain underwent a liquid-to-solid transition giving rise to the formation of fibrillar-like structures in the presence of the SH3 domain. However, this did not occur in the single (R385H or ΔPRM) and double mutants (ΔPRM and R385H) of the C-terminal domain of MEC-2, nor in samples not containing UNC-89 SH3 (Fig. 4a(ii)). Further, these droplets composed of MEC-2 and UNC-89 had significantly slowed recovery dynamics after photobleaching, as indicated by a longer $t_{1/2}$ in vitro (Extended Data Fig. 7g,h). We then performed FRAP experiments on mature MEC-2 condensates in vivo in the *unc-89* mutant background but could not detect significant changes in the recovery dynamics (Extended Data Fig. 7i), suggesting that the diffusion of MEC-2 condensates are not affected by the *unc-89* knockout (see Discussion, and ref. 44). Because FRAP experiments elucidate condensate fluidity through molecular mobility measurements[45], one explanation is that UNC-89 determines the elastic properties of these condensates, that are inaccessible to FRAP measurements[46].

Many complex fluids, including protein composites, glasses and active gels[47], exhibit viscoelastic material properties that depend on their age and the rate at which the stress is applied. To better understand the processes leading up to viscoelastic ageing, we performed single step relaxation (Extended Data Fig. 8) and active microrheology experiments in a dual optical trap[4,48]. The two optical traps hold a protein droplet by means of two optically trapped microspheres attached to it, one of which applies a sinusoidal perturbation onto the droplet with increasing frequencies, while the other clamps the droplet (Fig. 5a, Supplementary Video 8, Methods) and measures the transmitted force. The resulting force-displacement curves of the

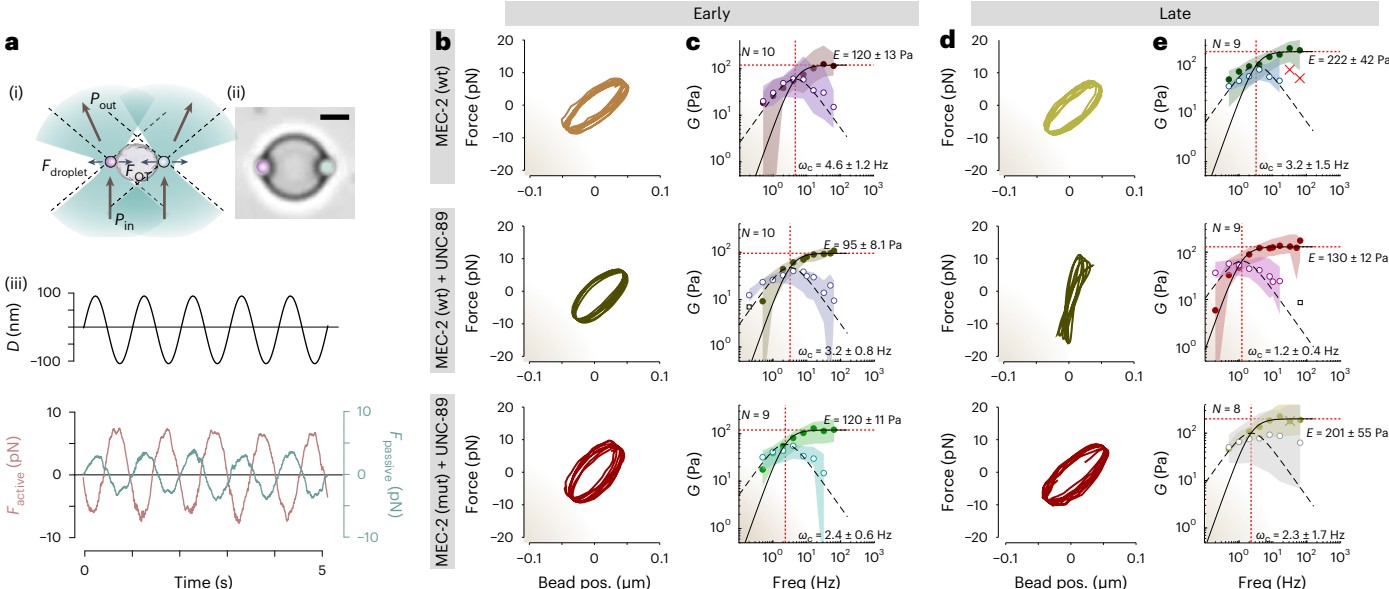

**Fig. 5 | UNC-89 promotes viscoelastic maturation of MEC-2 condensates.**
**a**(i), Schematic of the dual trap optical-tweezer-based active microrheology
assay, during which one of the trapped microspheres (pink) is actively
oscillated against the surface of a protein droplet, immobilized with a second
optically trapped bead (mint). (ii) Representative picture for at least $N = 10$
MEC-2 condensates showing the two microspheres in contact with the droplet.
Scale bar, 5 μm. (iii) Representative force–time signal of a typical sinusoidal
rheology test. The upper graph indicates the trap trajectory of the oscillating
bead; the lower graph shows the force exerted onto the active microsphere by
the droplet (pink) and the propagated force (mint). **b,d**, Force-displacement
plot (Lissajous-curve) showing the mechanical response of early (**b**, 0 h) and
late (**d**, 24 h) condensates at 4 Hz oscillation frequency composed of pure

MEC-2 (top), MEC-2:UNC-89 (middle) and MEC-2 (R385H):UNC-89 co-condensates
(bottom). The increase in slope indicates stiffening, whereas an increased
area under the ellipse indicates viscous dissipation. **c,e**, The complex shear
modulus as a function of deformation frequency. Solid circles represent the
median storage modulus ('elastic' behaviour), open circles represent the
median loss modulus ('viscous' behaviour), the shaded area embraces the 25th
percentile centred around the median for $N$ = number of droplets measured
from one experiment representative for three independent trials. Solid and
dashed lines indicate the fit for a Maxwell material (Methods) to the storage
and loss moduli, respectively. Red dotted lines indicate the fit parameters for
the plateau modulus $E$ and crossover frequency $\omega_c$ ± 95% CI, respectively. See
Supplementary Table 2 for statistics.

active trap provide rich insight into the mechanical response of the
protein condensates related to their frequency-dependent ability to
store and dissipate mechanical stresses[48], described as the storage
and loss modulus.

At low frequency, the storage modulus of all condensates was in
general smaller than the loss modulus, $G''(\omega)$, confirming the ability
of protein droplets to get deformed and dissipate stress at slow strain
rates: they tend to flow. At higher frequencies, by contrast, the storage
modulus dominates, indicating that the material response is primarily
elastic, and thus forces can effectively be transmitted. This crossover
frequency, $\omega_c$, is an important characteristic describing the upper limit
at which the material behaviour is dominated by fluid properties. We
found that immediately after condensate formation, MEC-2 alone had
a larger loss modulus up to the crossover frequency $\omega_c = 4.6 \pm 1.2$ Hz
(mean ± 95% confidence interval (CI), see Supplementary Table 2), con-
firming the ability of MEC-2 to dissipate slow stresses. The response was
unchanged 24 h after condensate formation ($\omega_c = 3.2 \pm 1.5$ Hz, $P = 0.19$,
two-tailed $t$-test, Fig. 5e, top), in agreement with our results from the
step indentations (Extended Data Fig. 8).

How does the presence of UNC-89 SH3 domain influence the
frequency-dependent mechanical properties of MEC-2/UNC-89
co-condensates? In naive droplets immediately after formation
($t = 0$ h), the co-condensates showed a response that was very similar
to those formed by MEC-2 alone ($\omega_c = 3.2 \pm 0.8$ Hz, mean ± 95% CI;
$P = 0.07$; Fig. 5c, middle). However, after 24 h of maturation, the
crossover frequency of the co-condensates was reduced relative naive
co-condensates ($\omega_c = 1.2 \pm 0.4$ Hz $P = 3 \times 10^{-4}$; Fig. 5e, middle), and to
that $t = 24$ h for the condensates formed by MEC-2 alone. This indicates
that in the presence of the SH3 domain, the wild-type condensates
behave as an elastic material even at slower strain rates, turning into

a stress-transmission element over a wider range of deformation fre-
quencies. In contrast, co-condensates prepared with mutant MEC-2
C-terminal domain showed no shift in the crossover frequency after
aging ($\omega_c = 2.3 \pm 2.7$ Hz, $p = 0.93$; Fig. 5c,e, bottom). In addition, age
had a small but significant effects on the elastic plateau modulus for
all conditions tested (Supplementary Table 2).

## MEC-2/UNC-89 co-condensates are under tension during touch

We engineered a genetically encoded FRET-tension sensor module
(TSMod)[49] into full-length MEC-2 between the stomatin domain and
the PRM (Fig. 6a), with the aim of visualizing force transmission across
MEC-2 during touch by measuring changes in FRET. This insertion did
not disrupt the localization of MEC-2 (Fig. 6a, Extended Data Fig. 6a–c)
and preserved partial touch sensitivity (Fig. 6b). Importantly, artificially
separating the FRET cassette with a 200-amino-acid spacer domain led
to constitutively low FRET signal (Fig. 6a(iv)), suggesting that intermo-
lecular FRET is minimal or absent. We then imaged the FRET signal in a
confocal microscope[50,51], while applying increasing pressure to animals
inside a microfluidic chip[35].

In animals expressing wild-type MEC-2::TSMod, FRET values
decreased as the deformation of the body wall increased ($\rho = -0.96$,
$P = 0.04$). Upon sudden pressure release, the FRET index increased
again to the same value as before the indentation, indicating that
MEC-2 can reversibly transmit force between the stomatin and the
C-terminal domains. We did not observe pressure-induced changes
in FRET with the C-terminal TSMod fusion (Extended Data Fig. 9a,c),
in UNC-89 mutant animals expressing the MEC-2::TSMod (Fig. 6c,d,
$\rho = -0.53$, $P = 0.24$), or in the transgenic animals bearing the MEC-2
(R385H) mutation (Fig. 6c,d).

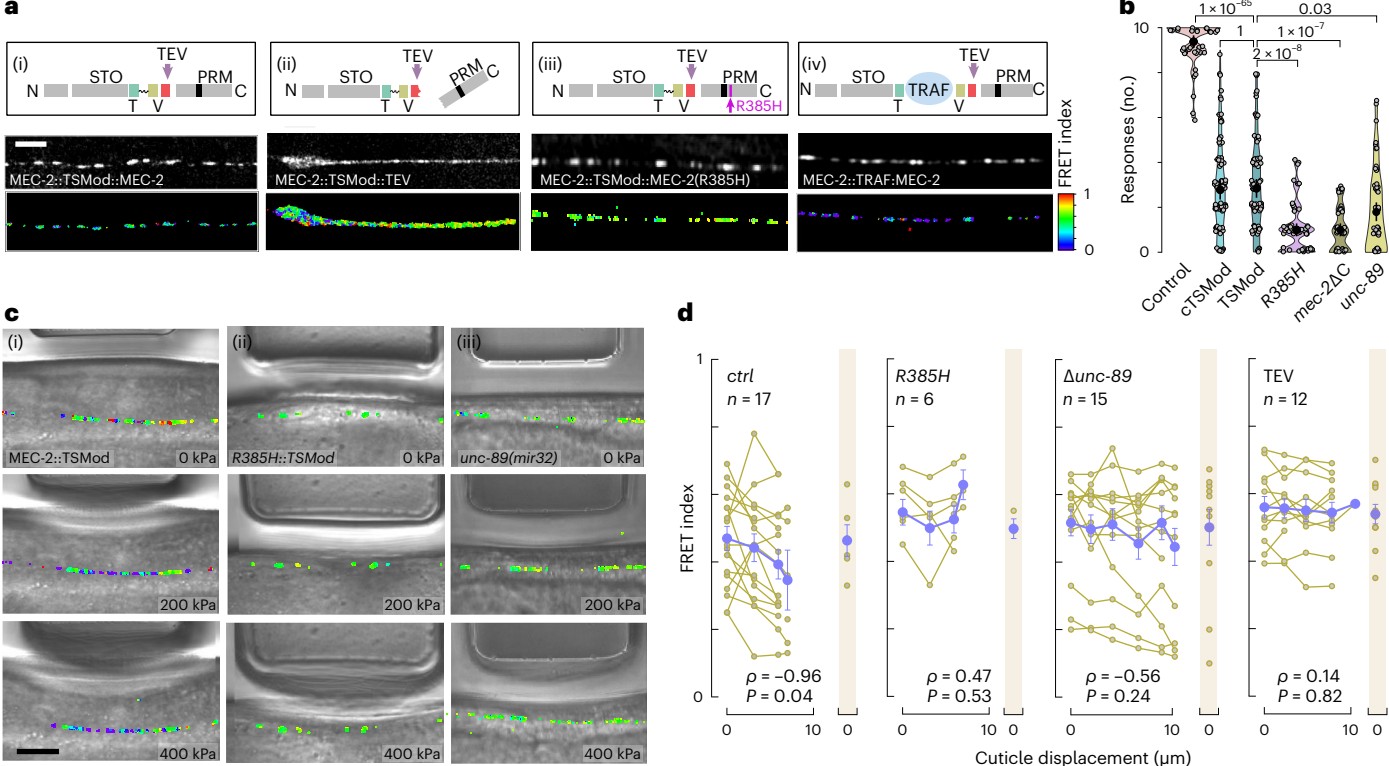

**Fig. 6 | MEC-2 sustains tension during body wall touch. a**, Schematics of the FRET-based tension sensing module (TSMod) integrated between the PRM and the stomatin domains for control (i), C-terminally truncated, TEV cleaved MEC-2::TSMod (ii), MEC-2 (R385H) mutant (iii) and the constitutive low FRET standard (iv). STO, stomatin domain; TEV, tobacco etch virus cleavage site; T, donor; V, acceptor fluorophore. Micrographs below show representative confocal picture highlighting the acceptor of the corresponding construct and the FRET map. Scale bar, 5 μm for all images. **b**, Violin plot of the body touch response derived from N2 wild-type control animals (*N* = 50), C-terminal TSMod (MEC-2:cTSMod, *N* = 95) fusion, internal (MEC-2::TSMod, *N* = 96) embedded between amino acids 370 and 371, MEC-2::TSMod in R385H mutant (*N* = 60), C-terminally truncated MEC-2::TSMod (*N* = 60) and MEC-2::TSMod in the *unc-89* mutant background (*N* = 60), being *N* the number of individual animals tested 10 times each, pooled over 3 independent experiments. The circle indicates mean and the black vertical bars indicate confidence intervals on the mean. *P*-value derived

from a two-tailed *t*-test with Bonferroni adjustment for multiple comparisons. **c**, Representative brightfield images of wild-type (i), R385H mutant MEC-2::TSMod (ii) and MEC-2::TSMod (iii); *unc-89* transgenic animals within the body wall chip under increasing force application (0–200–400 kPa, indicated by the actuator deflection) overlaid with their corresponding FRET index calculation. Scale bar, 10 μm. **d**, Quantification of the FRET index versus pressure delivered to the body wall for the indicated groups. Yellow lines represent sequences acquired for individual animals, purple circles the corresponding mean ± s.e.m. *r* = Pearson's correlation coefficient calculated between FRET versus cuticle deformation. Shaded column indicates the resting FRET value at 0 pressure after the pressure is released, indicating reversibility of the FRET response. *P*-value indicates statistical significance of the correlation coefficient derived from a two-sided *t*-test. *N* = number of animals tested pooled from 4, 2, 2, and 4 independent experiments for ctrl, R385H, *unc-89* and TEV, respectively.

Finally, we asked whether the naive fraction of MEC-2 would be able to transmit forces. Owing to the difficulty of performing the FRET measurements on moving spots under the application of an external pressure, we resorted to the conditionally, C-terminally truncated construct after TEV cleavage (Fig. 6d) and a MEC-2::TSMod ectopically expressed in the hypodermis (Extended Data Fig. 9b,d). The truncated protein could not mature into distinct solid-like condensates and is characterized by higher FRET values compared to wild type (Fig. 1f, Fig. 6a and Extended Data Fig. 3f–h). Similarly, the truncated and hypodermally expressed MEC-2::TSMod were not able to bear significant tension when tested under increasing mechanical load delivered to the body wall (Fig. 6).

## Discussion

Our notion that MEC-2 forms biomolecular condensates is in line with previous observations that stomatin forms higher order oligomers[14,16,52]. The intrinsically disordered C-terminal region of MEC-2 mediates protein–protein interactions[19,20] that eventually induce condensate maturation. What is the relevance of the naive phase? Because the diameter of axons is not constant along their length, and sometimes smaller than transported cargo[53], we propose that the liquid-like property facilitates

transport along the neurite with varying calibre, where material needs to be squeezed through constrictions along the way. Once at their cellular target, the MEC-2 condensates undergo a liquid-to-solid transition such that they are able to sustain mechanical stress over long timescales during body wall touch, which provides a focus for force transmission to the ion channel. Because MEC-2 is a membrane-associated protein with cholesterol binding ability[52,54], the membrane itself could direct the assembly process and the slowed diffusion within the plasma membrane might further delimit the extent and timescale of MEC-2 condensate formation[55]. Conversely, the liquid-to-solid transition might itself produce a stress that reshapes the neuronal membrane[56,57] and induce constriction of the axonal caliper[58], thus repositioning the MeT channel complex close to cytoskeletal elements that are critical for mechanosensation[59]. Our data is consistent with the previous observation that MEC-2 interacts transiently with MEC-4 in oocytes[60], where it may modulate the stiffness of cholesterol-enriched platforms surrounding the ion channel[61].

How does UNC-89 induce the rigidity transition? We propose that an aggregation-prone motif in the C-terminal domain of MEC-2 interacts with residues close to the PRM preserving the liquid character of the condensates by a mechanism known as heterotypic

buffering[62]. Binding of the UNC-89 SH3 domain to the PRM would, by competing against the interaction responsible for buffering, expose the aggregation-prone motif, thus leading to the liquid-to-solid transition. Indeed, in the absence of UNC-89, MEC-2 forms stiff condensates after 48 h in vitro but the addition of UNC-89 SH3 domain, at a 1:0.1 molar ratio, allows the transition to occur in only a few hours. This is also reflected in vivo, even though MEC-2 can form mature condensates with distinct mechanical properties in absence of UNC-89, because the overexpression of UNC-89 SH3 domain increases the relative size of the mature pool, in expense of the naive one. Although heterotypic buffering is considered key for preventing the liquid-to-solid transitions thought to be associated with neurodegeneration, we speculate that it here acts as a mechanism to induce a fast functional maturation of MEC-2 at mechanoelectrical transduction sites.

Many physiologically relevant responses of the cell depend on the rate at which stress is applied[63], including focal adhesion growth[64], bone morphogenesis[65] and the neuronal response to mechanical touch and hearing[66,67]. We speculate that the increase in relaxation time observed of the MEC-2/UNC-89 condensates in vitro (Fig. 5 and Extended Data Fig. 8) has profound consequences on their mechanical function as a tension sensor[4]. *C. elegans* displays a frequency-dependent touch response and is more sensitive to stimuli delivered fast as compared to slow stimuli[35,68]. Together, our data provides a compelling molecular mechanism for the observed frequency response and it offers a potentially unifying framework to understand how the ion channel kinetics are governed by the mechanical response of the viscoelastic environment.

Future work needs to address how mechanical force is transmitted to the MEC-2 condensates and how force modulates the energy landscape of the MEC-2:UNC-89 interaction within the condensates.

## Online content

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

## Methods

### *C. elegans* culture

Animals were maintained on nematode growth medium (NGM) plates seeded with *Escherichia coli* OP50 bacteria. Age-synchronized young adult animals were used for all the experiments and handled as described previously[69]. All strains generated in this study are listed in Supplementary Table 4.

### Molecular biology and transgenesis

All plasmids listed in Supplementary Table 5 were generated using the Gibson assembly method. All coding sequences were verified by sequencing. All details for plasmid cloning and CRISPR transgenesis can be found in Supplementary Note 2. Fluorophore knock-ins were performed as described previously[70], using a co-CRISPR strategy[71] or the mos1-mediated single copy insertion (MosSCI) method[72]. Multicopy transgenes were integrated using UV/TMP mutagenesis by exposing a cohort of animals with ultraviolett light and trimethylpsoralen[73].

### Prediction of phase separation behaviour

The sequence of the MEC-2 A isoform was modeled using AlphaFold2 and transform-restrained Rosetta was used for structure prediction with standard parameters (not shown). Prion-like sequences were predicted using the methodology described previously[28] with the *C. elegans* proteome as a background sequence. Phase separation index was calculated using software presented previously[29].

**In vitro assays.** Protein expression and purification. To obtain unlabelled MEC-2 protein, *E. coli* B834 (DE3) cells were transformed with the MEC-2 C-terminal (371–481) plasmid (pNS66). The cells were grown in LB medium at 37 °C until optical density (OD) = 0.6 and induced by the addition of isopropyl β-ᴅ-1-thiogalactopyranoside (IPTG) to a final concentration of 1 mM. The cultures were grown overnight at 25 °C. Cells were harvested upon centrifugation for 30 min at 4,000$g$. The cells were resuspended in 50 mM Tris, 50 mM NaCl and 1 mM dithiothreitol (DTT) buffer at pH 7.4. The cells were lysed by sonication and centrifuged for 30 min at 38,000$g$. The supernatant was loaded in a nickel affinity column (HisTrap HP 5 ml, Cytiva) and eluted with a gradient from 0 to 500 mM imidazole. The histidine affinity tag was cleaved with 3C protease through dialysis for 2 h in cleavage buffer (50 mM Tris-HCl, 50 mM NaCl, 1 mM DTT, pH 7.4). 8 M urea was added in order to separate NusA and MEC-2. The reverse nickel column was run with 8 M urea to remove the cleaved tag and uncleaved protein. After loading, MEC-2 was eluted with a buffer containing 50 mM Tris-HCl, 300 mM NaCl, 20 mM imidazole, 1 mM DTT and 8 M urea at pH 8.0. The eluted protein was injected in a size exclusion Superdex 75 (Cytiva), running in 20 mM sodium phosphate buffer with 1 mM TCEP and 0.05% (w/v) NaN₃ at pH 7.4. The fractions with protein were collected and concentrated to 400 µM, fast frozen in liquid nitrogen and stored at −80 °C.

To obtain unlabelled UNC-89 protein, *E. coli* B834 (DE3) cells were transformed with the UNC-89 SH3 (59–128) plasmid (pNS75). The cells were grown in LB medium at 37 °C until OD = 0.6 and induced by the addition of IPTG to a final concentration of 1 mM. After growing 4 h at 37 °C, the cells were centrifuged for 30 min at 4,000$g$ and resuspended in 50 mM Tris and 50 mM NaCl buffer at pH 7.4. The cells were lysed by sonication and centrifuged for 30 min at 38,000$g$. The pellet was washed twice with washing buffer (PBS, 1 mM DTT, 500 mM NaCl, 1% TritonX-100, PIC, PMSF and DNAse, at pH 7.4). The pellet was resuspended in resuspension buffer (25 mM Tris-HCl, 8 M urea, 500 mM NaCl, 10 mM imidazole, pH 8.0) and centrifuged for 30 min at 38000 xg. The supernatant was loaded in a nickel affinity column (HisTrap HP 5 mL, Cytiva) and eluted with a gradient from 0 to 500 mM imidazole. The histidine affinity tag was cleaved with 3C protease through dialysis for 2 h in cleavage buffer (50 mM Tris-HCl, 200 mM NaCl, 1 mM DTT, pH 8). The reverse nickel column was run to remove the cleaved tag and uncleaved protein. After loading, UNC-89 was eluted with a buffer

containing 50 mM Tris-HCl, 300 mM NaCl, 20 mM imidazole, 1 mM DTT and 8 M urea at pH = 8.0. The eluted protein was injected in a size exclusion Superdex 75 (Cytiva), running in 20 mM sodium phosphate buffer with 1 mM TCEP and 0.05% (w/v) NaN₃ at pH 7.4. The fractions with protein were joined and concentrated to 1 mM, fast frozen in liquid nitrogen and stored at −80 °C.

Isotopically ¹⁵N/¹³C- and ¹⁵N-labelled proteins were produced by growing transformed *E. coli* B834 cells in M9 minimal medium containing 1 g l⁻¹ of ¹⁵N-NH₄Cl and 2 g l⁻¹ of ¹³C₆-ᴅ-glucose.

**NMR experiments.** Backbone assignment. NMR experiments were recorded at 278 K on a Bruker Avance NEO 800 MHz spectrometer or a Bruker Avance III 600 MHz, both equipped with a TCI cryoprobe and the TopSpin software (v.4.0.8). A 400 µM ¹⁵N, ¹³C double-labelled MEC-2 (371–481) sample in NMR buffer (20 mM sodium phosphate (pH 7.4), 1 mM TCEP, 0.05% (w/v) NaN₃) was used for backbone resonance assignment. A series of 3D triple resonance experiments were recorded, including the BEST-TROSY version of HNCO, HN(CA)CO, HNCA, HNCACB, and HN(CO)CACB[74]. Chemical shifts were deposited in BMRB (ID:51491). Secondary structure propensities were derived from the H, N, C′, C$^\alpha$ and C$^\beta$ chemical shifts measured by using solution state NMR and using the δ2D software[30]. A 1200 µM ¹⁵N, ¹³C double labelled UNC-89 SH3 domain (59–128) sample in NMR buffer (20 mM sodium phosphate (pH 7.4), 1 mM TCEP, 0.05% (w/v) NaN₃) was assigned using the same NMR experiments as described above. Chemical shifts were deposited in BMRB (ID:51490). SSP values were computed by using δ2D[30].

Binding mapping. CSP and signal intensity changes ($I/I_0$) were extracted by measuring ¹H–¹⁵N correlation spectra of ¹⁵N, ¹³C double-labelled MEC-2 (371–481) with 10 molar equivalents of UNC-89 SH3 domain, and vice versa. Data analysis was performed with CcpNmr V3[75].

**Sample preparation for in vitro experiments.** All samples were prepared on ice as follows. First, a buffer stock solution consisting of 20 mM sodium phosphate, 1 mM TCEP and 0.05% (w/v) NaN₃ was pH adjusted to 7.4 and filtered using 0.22 µm sterile filters (buffer stock). A 5 M NaCl solution in the same buffer was also pH adjusted to 7.4 and filtered (salt stock). Then, the protein samples were thawed from −80 °C on ice, pH adjusted to 7.4 and centrifuged for 5 minutes at 22,000$g$. The supernatant (protein stock) was transferred to a new Eppendorf tube and the protein concentrations were determined by their absorbance at 280 nm. The indicated samples were prepared by mixing the right amounts of buffer stock, protein stock and salt stock to reach the desired final protein and NaCl concentrations.

**Turbidity assay.** Absorbance of the samples was measured at 350 nm ($A_{350nm}$) using 1 cm path length cuvettes and a Cary100 ultraviolet–visible spectrophotometer equipped with a multicell thermoelectric temperature controller. The temperature was increased progressively from 10 to 60 °C at a ramp rate of 1 °C min⁻¹. The cloud temperatures ($T_c$) were determined as the maximum of the first-order derivatives of the curves.

**Differential interference contrast microscopy.** 1.5 µl of sample was deposited in a sealed chamber comprising a slide and a coverslip sandwiching double-sided tape (3M 300 LSE high-temperature double-sided tape of 0.17 mm thickness). The used coverslips were previously coated with PEG-silane following the published protocol in ref. [76]. The differential interference contrast (DIC) images were taken using an automated inverted Olympus IX81 microscope with a 60x/1.20 water UPlan SAPo objective using the Xcellence rt 1.2 software.

**In vitro confocal fluorescence microscopy.** MEC-2 wild-type or R385H were labelled with Alexa Fluor 647 and UNC-89cys mutant with DyLight 488, following provider's instructions (Thermo Fisher

Scientific). The samples for fluorescence microscopy were prepared as previously described but containing 1 μM of labelled protein molecules. UNC-89cys mutant plasmid was generated by directed mutagenesis on top of pNS75 backbone to incorporate a Ser to Cys change in amino acid 62 for subsequent fluorescence labelling giving rise to pNS77.

Fluorescence microscopy images and FRAP experiments were recorded using a Zeiss LSM780 confocal microscope system with a Plan ApoChromat 63×1.4 oil objective. For the FRAP experiments, 10 or 11 droplets of similar size were selected for MEC-2 (R385H) or MEC-2 wild-type, respectively. The bleached region was 30% of their diameter, and the intensity values were monitored for ROI1 (bleached area), ROI2 (entire droplet) and ROI3 (background signal). The data was fitted using the EasyFrap software[77] (v.1.11) to extract the kinetic parameters such as the half-time of recovery and the mobile fraction.

## Optical tweezer microrheology of phase-separated protein droplets in vitro

**Preparation.** Samples containing microspheres and protein droplets were elaborated following the recipe described in the last section. PEG surface-modified latex microspheres (1 μm diameter, 01-54103, Micromod) were added to the buffer stock and were mixed with salt stock and protein stock (see above) for a final volume of 10 μl. Final microsphere concentration was $9.3 \times 10^7$ ml$^{-1}$, salt concentration was 2 M, MEC-2 was 200 μM and, for the mixed protein droplets, UNC-89 was 20 μM. Chambers containing the protein samples for microscopy and optical trapping were made by gluing a cover-glass (22 × 22 mm, #1.5, Menzel-Glaser) and a microscope slide (1 mm thick, Deltalab) with a 20 × 20 mm piece of double-sided adhesive tape with a hole (2 mm in diameter) in the middle. Cover glasses were incubated overnight for a PEG-silane coating following protocols in ref. 76. A protein sample of 5–6 μl was added into the cavity before closing the chamber, which was kept horizontal with the cover-glass facing down, for a specific time at room temperature. In this way, protein droplets settled onto the PEG-treated surface (Fig. 2d and Fig. 5a). In our measurements, droplets with a diameter of $D \sim 5 \pm 2$ μm were selected.

**Optical trapping protocol.** A detailed description of our optical trapping set-up can be found elsewhere[78].

Briefly, the optical traps were created at the focal plane of a water-immersion objective (60×, NA = 1.2, Plan Apo, Nikon) using an AOD-modulated optical micromanipulation unit (SENSOCELL, Impetux Optics) coupled to the rear epi-fluorescence port of an inverted microscope (Eclipse Ti2, Nikon). The system was equipped with a light momentum force detection module (SENSOCELL, Impetux Optics), which is transparent for visible light, enabling brightfield illumination. The trapping power and the initial light momentum variations across the field of view (FOV) were compensated using manufacturer's built-in routines (LightAce version 1.6.2.0, Impetux Optics), prior to the measurements, in an empty optical trapping chamber[78].

In stress–relaxation tests, one optical trap with $P_{trap}$ = 20 mW was used to trap one microsphere. For active microrheology, two optical traps (20 mW each) were created by time-sharing the steering angle of the laser beam at 25 kHz and trapped two microspheres. Trap stiffnesses, $k_1$ and $k_2$, were obtained by using the manufacturer's particle scan routine in the optical trapping software (LightAce, Impetux Optics), from which a line was fit over the linear regime of the optical force profile (following $F = kx$, −100 nm < $x$ < 100 nm). After that, the microspheres were brought into contact with a protein droplet (Fig. 5a), which was noticed as a force against the bead, normal to the droplet surface. Non-specific attachment was confirmed by a force of opposite sign when trying to pull the microsphere back (Supplementary Note 1).

**Rheological measurement of the protein droplet $G$ modulus.** We used the protocol introduced previously[79] to obtain the complex spring

constant, $\chi^*(\omega)$, and the $G$ modulus of our protein droplets, with a slight modification regarding the measurement of the surface tension, γ (Supplementary Note 1). We performed two in-series measurements using custom programs based on the manufacturer's SDK in LabView (LightAce, Impetux Optics). First, we obtained force–relaxation curves upon a constant step $A$ by applying step indentations to trap 1 (0.2 Hz, $A$ = 100 nm, Extended Data Fig. 8). The curves were fitted with an exponential decay, $F_{1,2}(t) = F_0^{(1,2)} + \left(F_p^{(1,2)} - F_0^{(1,2)}\right)e^{-t/\tau^{(1,2)}}$ for traps 1 and 2, respectively, to determine the time constant, $\tau$, and the peak, $F_p$, and resting, $F_0$, forces. The resting droplet spring constant, $\chi_0$, was obtained as shown in equation (1a). Surface tension, γ, was obtained by following the definition in ref. 80 (equation (1c)).

$$\Delta x_0 = A + \frac{F_0^{(1)}}{k_1} - \frac{F_0^{(2)}}{k_2} \tag{1a}$$

$$\chi_0 = \frac{F_0^{(1)} - F_0^{(2)}}{2\Delta x_0} \tag{1b}$$

$$\gamma \approx \frac{\chi_0}{\pi}(-\ln\theta_0 + 0.68) \tag{1c}$$

Second, trap 1 applied a sequence of sinusoidal indentations onto the protein droplet at increasing frequencies, $f$ = 0.5, 1, ..., 64 Hz, giving rise to forces at traps 1 and 2, $\tilde{F}_{1,2}(\omega)$, with the tilde denoting a Fourier transform. Being that the total length of the coupled system was $x_{sys} = x_{trap}^{(2)} - x_{trap}^{(1)}$, the coupled spring constant of the system, $\chi_{sys}^*(\omega)$, and the frequency-dependent complex spring constant of the droplet, $\chi^*(\omega)$, were determined as follows[79]:

$$\chi_{sys}^*(\omega) = \frac{\tilde{F}_1(\omega) - \tilde{F}_2(\omega)}{2\tilde{x}_{sys}(\omega)} \tag{2a}$$

$$\chi^*(\omega) = \frac{\chi_{sys}^*[4k_1k_2 + i\xi\omega(k_1 + k_2)]}{2k_1(2k_2 + i\xi\omega) - 4\chi_{sys}^*(\omega)(k_1 + k_2 + i\xi\omega)} \tag{2a}$$

Here, $\xi = 3\pi\eta D$ is the viscous drag of the droplet (with $\eta = 1$ mPa s$^{-1}$ being the medium viscosity). Finally, the complex shear modulus of the protein droplets was computed as follows[80]:

$$G^*(\omega) = \frac{\chi^*(\omega) - (1.25 + 4.36\theta_0^2)\gamma}{R(5.47\theta_0^5 - 29.28\theta_0^4 + 23.29\theta_0^3 - 5.08\theta_0^2 + 3.79\theta_0 - 0.02)} \tag{3}$$

From equation (3), storage and loss moduli were obtained as $G'(\omega)$ = Re[$G^*(\omega)$] and $G''(\omega)$ = Im[$G^*(\omega)$]. Data were fitted a Maxwell element, that is a dashpot in series with a spring[81], using a nonlinear regression algorithm in MATLAB. From the viscosity of the dashpot, $\eta$, and the spring constant, $E$, we obtained the crossover frequency as $\omega_c = E/(2\pi\eta)$ (Hz) and the characteristic timescale $\tau = 1/\omega_c$. All data processing was carried out with custom scripts in MATLAB (R2019b).

## Behavioural assays

**Gentle body touch assays.** Gentle body touch assays were carried out as described elsewhere[82]. An eyebrow hair was used to gently touch 20–30 young adult worms ten times with alternative anterior and posterior touches (five each). Unless otherwise stated, these experiments were repeated at least three different days to determine an average response and s.d. out five or ten sequential touches. Touch behaviour was scored after non-randomized stimuli presentations, consequential to successive head/tail touch events. Behaviour was scored blind to genotype in the RNAi screen, the MEC-2::TSMod rescue constructs and the unc-89 mutant animals comparing to the wild-type or untreated control conditions.

**TRN-specific RNAi feeding.** RNAi bacterial clones were obtained from Ahringer[43] (*mec-2, mec-4, ced-5, sem-5, toca-1, tbc-18, sdpn-1, F49E2.2, unc-73, spc-1, mlk-1, abl-1, sma-1, plc-1, itsn-1, nck-1, magu-3, ccb-1, C46H3.2, prx-13, erp-1, ephx-1*) or Vidal (ORFeome-Based)[83] (*BO303.7, abi-1, magu2, unc-89, stam-1, lin-2, F42H10.3, amph-1, dbn-1, C36E8.4, T04C9.1, unc-57, Y106G6H.14, ced-2*) libraries, donated by Cerón and Lehner labs. First, clones were verified by colony PCR and sequencing. NGM plates were supplemented with 6 mM IPTG (I1001-25 Zymo Research) and 50 μg ml⁻¹ ampicillin. Unseeded feeding plates were completely dried in a laminar airflow hood and kept in the dark at 4 °C. RNAi bacterial clones were grown in 4 ml LB with 50 μg ml⁻¹ ampicillin. Next day, each plate was inoculated with 200 μl of the corresponding bacterial culture and dried for 1 h under the hood. The expression of dsRNA was induced in the presence of IPTG overnight at room temperature, or in an incubator at 37 °C for 4 h, in the dark. Then, six gravid hermaphrodites (TU3403[37]) were transferred onto the plates and grown at 25 °C for 48 h. The progeny were tested for body touch sensitivity at young adult stage, a total of 40 worms on two different days. Importantly, wild-type animals are insensitive for neuronal RNAi due to the lack of dsRNA transporter in neurons. Thus, animals were sensitized to RNAi through a TRN-specific SID-1 rescue construct (*mec-18*p::*sid-1*(+)[37]) in a systemic RNAi mutant *sid-1*(qt2) background.

**Auxin-induced degradation experiment.** Auxin plates were prepared as described elsewhere[84], by adding 250 mM stock of 1-naphthaleneacetic acid (NAA Auxin, Sigma Aldrich 317918) dissolved in 95% ethanol to cooled NGM before pouring the plates at a final concentration of 1 mM auxin. Plates were dried out under a laminar airflow hood, seeded with 10× concentrated *E. coli* OP50 and dried for 1 h under the hood. The next day, six gravid hermaphrodites were transferred onto the plates and grown at 25 °C for 48 h. The progeny were tested for body touch sensitivity at young adult stage.

**Calcium imaging of TRNs from microfluidically immobilized animals after body wall touch**
**Device preparation.** Devices were replica-moulded from SU8 photolithography mould as described previously[35] using either 1:10 or 1:15 PDMS pre-polymer:curing agent mixtures. Devices prepared with 1:10 mixtures were used for the calcium-imaging experiments (Fig. 2i,j and Fig. 3b). 1:15 mixtures were used to obtain larger cuticle displacements during the Force-FRET experiments (Fig. 6). Diaphragm deflection versus pressure was calibrated previously for both mixtures[85].

**Animal loading into a microfluidic trap.** Loading of the animals in the body wall chip was performed as described in detail elsewhere[86]. Briefly, two to three young adult animals were transferred to a M9-filtered droplet. Then, worms were aspirated through a SC23/8 gauge metal tube (Phymep) connected to a 3 ml syringe (Henke Sass Wolf) with a PE tube (McMaster-Carr). Once the tube was inserted in the inlet of the chip, the animals were loaded on the waiting chamber by applying gentle pressure with the syringe. In general, animals were oriented head-first. The same device was used to compare between genotypes.

**Calcium imaging.** In vivo calcium imaging of TRNs was performed by positioning the worm-loaded microfluidic device in a Leica DMi8 microscope with a 40×/1.1 water-immersion lens, Lumencor Spectra X LED light source, fluorescence cube with beam splitter (Semrock Quadband FF409/493/573/652), a Hamamatsu Orca Flash 4 V3 sCMOS camera and HCImage software (version 4.4.2.7). Cyan-488 nm (≈6.9 mW) and yellow-575 nm (≈12.6 mW) illuminations were used to excite the green and red fluorescence of the TRN::GCaMP6s and mtagRFP, which was used to correct possible artifacts from animal movement and TRN identification. The incident power of the excitation light was measured with a Thorlabs microscope slide power meter head (S170C) attached to PM101A power meter console. Emission was

split with a Hamamatsu Gemini W-View with a 538 nm edge dichroic (Semrock, FF528-FDi1-25-36) and collected through two single-band emission filters, 512/525 nm for GCaMP (Semrock, FF01-512/23-25) and 620/60 for mtagRFP (Chroma, ET620/60 m). The emission spectra was split by the image-spliter, allowing different exposure times for each signal. For mechanical load application to the body wall of the animal, the stimulation channel was connected to a piezo-driven pressure controller (OB1-MK3, Elveflow) as described previously[87]. To follow calcium transients, videos were taken at 10 frames per second with 80 ms exposure time, using the master pulse from the camera. The camera SMA trigger out was used to synchronize the stimulation protocol in Elveflow sequencer, which consisted on 20 s pre-stimulation, 2 s stimulation (2500 mbar buzz) and 40 s post-stimulation.

**Calcium analysis.** Images were processed using MATLAB in-house procedures to extract GCaMP signal intensity[88]. First, the TRN was manually labelled based on the mtagRFP calcium insensitive channel. The position was automatically tracked in the following frames and used to extract the GCaMP intensity. A smooth filter (moving average filter) was applied. The calcium sensitive signal was normalized to the first 100 frames pre-stimulation ($F - F_0/F_0$).

**Fluorescence resonance energy transfer**
**Data acquisition.** Fluorescence resonance energy transfer (FRET) imaging of worms loaded within the body wall chip as described above was performed on a Leica DMI6000 SP5 confocal microscope using the 63×/1.4 NA oil immersion lens. As described in detail in ref. 50, three images were collected: the direct donor (mTFP2) excitation and emission, donor excitation and acceptor emission and direct acceptor (mVenus) excitation and emission. mTFP2 was excited with 458 nm (≈9 μW), mVenus with 514 nm (≈4 μW) line of an Argon ion laser at 80% and 11% transmission respectively (25% power). The incident power of the excitation light was measured with a Thorlabs microscope slide power meter head (S170C) attached to PM101A power meter console. A single set of images was collected before and after pressure delivery, while recording images for each pressure. Due to defocussing immediately after the pressure delivery, manual refocussing was necessary to keep the focus in the plane of the MEC-2 clusters. The pressure was converted into cuticle displacement using the table provided in ref. 35. for 1:10 PDMS or in ref. 85. for 1:15 PDMS mixtures.

**Data analysis.** Procedures as described in ref. 50 have been followed. In short, due to the spectral overlap of the donor and the acceptor, the resulting FRET images are contaminated with donor bleedthrough and acceptor cross-excitation. To eliminate this spurious signal, a linear unmixing procedure as a bleedthrough correction was employed. The images were first bleedthrough corrected with a factor that was predetermined prior to each experiment with animals that express either fluorophore alone (for details about the procedure, see ref. 50). The corrected FRET channel was then normalized by the sum of the background corrected donor channel and the corrected FRET channel on a pixel-by pixel basis. With the aim to eliminate pixels outside of the region of interest and ubiquitous autofluorescence inherent to living *C. elegans*, we applied a mask on the acceptor channel to separate the MEC-2 clusters from the background. To correlate the FRET signal with the cuticle displacement, we used the Pearsons's correlation $\rho$ coefficient and calculated the t-test statistics according to $t = \frac{\rho}{r\omega}$ with $r\omega$ as the CI of $\rho$.

**Fluorescence recovery after photobleaching**
**Data acquisition.** Fluorescence recovery after photobleaching (FRAP) imaging was performed on a Leica DMI6000 SP5 confocal microscope using the 63×/1.4 NA oil immersion lens. Animals were imaged live in 3 mM levamisole on 5–6% agarose pads. The FRAP protocol consisted on 5 frames pre-bleach (every 371 ms), 5 frames of bleach

(every 344 ms), 10 frames post-bleach 1 (every 371 ms) and 10 frames post-bleach 2 (every 20 s). mCherry or mVenus were excited with a 594 or 514 nm line of an argon ion laser, respectively, at 5% of total power, except for the bleach step for which it was set at 100%. For bleaching a small ROI within MEC-2 puncta, same protocol was applied, except for the bleach step where 60% 594 nm line of Argon ion laser was used. For bleaching a small ROI within MEC-2 condensates in hypodermis, the protocol consisted on 5 frames pre-bleach (every 97 ms), 5 frames of bleach (every 66 ms), 10 frames post-bleach 1 (every 1 s) and 10 frames post-bleach 2 (every 20 s).

**Data analysis.** Images were pre-processed using ImageJ/Fiji (v.1.53f51). First, ROI1 was manually drawn in the bleached area and used to track the intensity measurement in the following frames. The same was done for the total fluorescence area (ROI2) and the background area (ROI3). This data was processed by the easyFRAP online tool[77] (version 1.11) to compute the normalized recovery curves using full scale normalization, which corrects for differences in the starting intensity in ROI1, differences in total fluorescence during the time course of the experiment and differences in bleaching depth. When a small region within a phase-separated condensate was bleached (as in Fig. 1g; Extended Data Figs. 3, 4g,i and 7i), an extra step of normalization to the bleaching rate (considering first frames of pre-bleach from another ROI in a different punctum/condensate) was added, because the intensity of the whole punctum/condensate decreased during bleaching and could not be used to correct for fluorescence changes during the course of the experiment. Finally, all individual recovery curves were averaged and fit to an exponential with the form $I(t) = Mf(1 - \exp(-\tau \cdot t))$ to extract the mobile fraction and the half-time of recovery $t_{1/2} = -\ln 0.5/\tau$. As the recovery half-time strongly depends on the radius of the user-defined bleach spot, it cannot be compared among different studies and experimental conditions. The diffusion coefficient was calculated using the Soumpasis equation $D = 0.224 \frac{r^2}{t_{1/2}}$ directly from the fit parameters and the measured bleach spot size. Data from moving animals were removed due to the uncertainty in intensity extractions.

## Confocal microscopy

Fluorescence images were taken using an inverted confocal microscope (Nikon Ti2 Eclipse) with a 60×/1.4 NA oil immersion lens. Animals were imaged live in 3 mM levamisole on 5–6% agarose pads. mCherry was excited using the 561 nm laser, (2–3 mW at the sample plane) and transmitted through a 594 nm emission filter. Exposure time was 100–200 ms, depending on the strain to image. GFP was excited with a 488 nm laser, (2–4 mW at the sample plane) and transmitted through a 521 nm emission filter. Exposure time was 100–200 ms.

**Tracking of MEC-2 along TRN.** Fluorescently labelled MEC-2 was imaged in the axons of the TRNs using a Leica DMi8 microscope with a 63×/1.4 oil immersion lens. Imaging was performed at a frame rate of 100 ms. The naive pool was imaged in close proximity to the cell body and the mature pool at the periphery of the axon. MEC-2 trajectories of the mobile pool were extracted using MTrackJ[89] and the trajectories for the immobile pool were obtained with the ImageJ Plug-in TrackMate[90]. The resulting trajectories were further analysed using a Python script to compute the mean squared displacement defined as

$$\mathrm{MSD}(\tau) = \left\langle \left( \vec{x}(t) - \vec{x}(t + \tau) \right)^2 \right\rangle. \qquad (4)$$

The average was taken over the time and the ensemble of the measured trajectories.

The quantification of the area, roundness and number of aggregates per neurite during aging was performed using ilastik software (https://www.ilastik.org/ ref. 91) to segment and identify our aggregates of interest based on machine learning algorithms trained with reference images from the corresponding larval stage, followed by an in-house Python script to extract the above-mentioned parameters.

**Colocalization analysis.** Images of fluorescently labelled MEC-2::mCherry (wild-type or R385H mutant) and the SH3 domains of UNC-89::GFP or the negative control SPC-1::GFP were first treated with ImageJ to identify TRNs as ROI (50 μm length) and straighten them for posterior analysis. Colocalization values were extracted with the stats.pearsonr() function from Python to calculate the Pearson correlation coefficient and its associated $P$-value. In order to verify that the colocalization was specific, we generated scrambled images of the MEC-2::mCherry channel. We chopped the image of the neuron in several pieces and randomly shuffled them with an in-house Python script and tested the colocalization respect to the SH3 motifs as described before.

**Interpunctum interval analysis.** For calculating the MEC-2 interpunctum interval, a 160–190 μm length ROI was drawn in TRN axon images in ImageJ. A threshold value and background subtraction were applied to remove the particles outside TRNs. The ImageJ particle counting tool was used to infer the position of each particle and the difference between them was derived. The resulting values were used to calculate the mean difference using Python.

## Locomotion

Locomotion assays were performed as described in ref. 88. Briefly, 1 min long videos of young adult synchronized animals transferred to non-seeded NGM plates were recorded at 25 fps using a homebuilt tracking platform. Imaging processing was done by using eigenworms with custom MATLAB scripts. To compare the genotypes, animal locomotion behaviour was decomposed using the Eigenworm analysis[92]. To construct and visualize the 3D distributions of the modes ($a_1, a_2, a_3$), the 3D kernel density estimate of the first three modes was calculated in R using the ks package[93] with an unconstrained plug-in selector bandwidth. We choose to indicate the 10, 25 and 50% contours of the highest density regions in the manifold and the 2D projections of the floating data cloud along the corresponding planes. To compare two different datasets and test for the null hypothesis that the two kernel density functions are similar, we resampled the highly oversampled population by bootstrapping to avoid spurious significance due to long tailed outliers. The resampling and testing was performed 1,000 times to yield a distribution of $P$-values which is displayed as a violin plot summarizing each figure of locomotion data. Importantly, the resampling does not lead to significant discrimination of the downsampled and original dataset within the same population.

## Statistics and reproducibility

All statistical calculations were performed in R (v.4.2.2, 2022-10-31 with 'ks' package v.1.14.0), Python (v.3) or online software according to ref. 94. The paired mean difference or Cohen's $d$ was used to estimate the effect size in addition to the dichotomous decision guided hypothesis tests whenever indicated in the figure. We exclusively performed two-sided tests, unless otherwise indicated in the figure legend. The level of significance for all comparison was chosen 0.05, unless otherwise indicated in the figure or figure legend. No statistical methods were used to pre-determine sample sizes but our sample sizes are similar to those reported in previous publications. Data collection and analysis were not performed blind to the conditions of the experiments, except when indicated otherwise (for example touch behaviour). Data distribution was assumed to be normal but this was not formally tested. Non-parametric tests were used when data was deviating from the assumption of normality as indicated in the figure legends.

## Reporting summary

Further information on research design is available in the Nature Portfolio Reporting Summary linked to this article.

## Data availability

The chemical shifts in the NMR data of MEC-2 and UNC-89 were deposited in the Biological Magnetic Resonance Data Bank (https://bmrb.io/) under accession codes 51491 and 51490, respectively. The publicly available datasets used in this study are accessible through wormbase.org v.WS289. Protein sequences were obtained from UniProt (O01761). Calcium imaging data of the animals under a mechanical stimulus has been deposited on Zenodo (https://doi.org/10.5281/zenodo.8163972). Source data are provided with this paper. All other data supporting the findings of this study are available from the corresponding author on reasonable request.

## Code availability

Locomotion and calcium data were analysed using custom Matlab scripts (v.2021)[88]. Ratiometric FRET data analysis was performed in IgorPro 6.37 using algorithms described previously[50,51]. NMR data analyses were performed as described in detail using published open source software (CcpNmr v.3 and delta2D v.1). Computer code to analyse optical tweezer data (Matlab scripts (v.2021)) is available at https://gitlab.icfo.net/rheo/Tweezers/droplets. All other custom scripts are available upon request.

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

## Acknowledgements

We thank the Neurophotonics and Mechanical Systems Biology (NMSB) and Super-resolution Light Microscopy and Nanoscopy (SLN) labs for discussions and suggestion throughout the work and for use of their microscopes. We thank the Institut de Ciències Fotòniques (ICFO) Biolab and NanoFabrication Lab (NFL) for support with animal maintenance and SU8 lithography, respectively. We thank S. Karimi for fabrication of microfluidic devices, and NMSB lab members for help with Python and MATLAB analysis. We thank the Infraestructuras Científicas y Tecnológicas Singulares (ICTS) nuclear magnetic resonance (NMR) facility, managed by the scientific and technological centres of the University of Barcelona (CCiTUB), for their help in NMR and the Institute for Research in Biomedicine (IRB) Barcelona advanced digital microscopy facility, for their help with microscopy experiments. We thank M. Chalfie, M. Goodman, C. Carolis and the Caenorhabditis Genetics Center (CGC) (National Institutes of Health, Office of Research Infrastructure Programs (P40 OD010440)) for providing reagents; B. Lehner and J. Cerón for sharing their RNAi libraries. M.K. acknowledges financial support from the European Research Council (ERC, MechanoSystems, 715243), Human Frontiers Science Program (CDA00023/2018), ministerio de ciencia e innovación/agencia estatal investigacion (MCIN/AEI/10.13039/501100011033/ FEDER (European Regional Development Fund) 'A way to make Europe', PID2021-123812OB-I00), 'Severo Ochoa' programme for Centres of Excellence in R&D (CEX2019-000910-S), from Fundació Privada Cellex, Fundació Mir-Puig, and from Generalitat de Catalunya through the Centres de Recerca de Catalunya (CERCA) research programme, in addition to funding through the ministerio de economia, industria y competitividad (MINECO, FPIPRE2019-088840 and RYC-2016-21062 funded by MCIN/AEI/10.13039/501100011033 and ESF/European Social Fund 'Investing in your future' to N.S.-C. and M.K., respectively). X.S. acknowledges funding from Agència de Gestió d'Ajuts Universitaris i Recerca (AGAUR. 2017 SGR 324), MINECO (BIO2015-70092-R and PID2019110198RB-I00), and the ERC (CONCERT, contract number 648201). B.M. acknowledges financial support from the Asociación Española contra el Cáncer (FCAECC project #POSTD211371MATE). C.G.-C. acknowledges a graduate fellowship from MINECO (PRE2018-084684). IRB Barcelona and ICFO

are the recipient of a Severo Ochoa Award of Excellence from MINECO (government of Spain).

## Author contributions

N.S.-C. was responsible for animal husbandry, molecular biology (cloning, CRISPR, RNAi), behavioural experiments, FRET and FRAP assays, calcium imaging, data analysis and manuscript writing. F.C.-C. performed optical tweezer microrheology. B.M., C.G.–C., M.R. and A.C.-C. performed NMR and in vitro phase separation experiments. I.R. and S.W. performed particle tracking. M.P.-d.-l.-R. performed molecular biology and locomotion assays. X.S. was responsible for study conceptualization, acquisition of funding and manuscript writing. M.K. was responsible for study conceptualization, acquisition of funding, data acquisition, analysis and manuscript writing.

## Competing interests

X.S. is founder and scientific advisor of Nuage Therapeutics. All other authors declare no competing interests.

## Additional information

**Extended data** is available for this paper at https://doi.org/10.1038/s41556-023-01247-0.

**Correspondence and requests for materials** should be addressed to Xavier Salvatella or Michael Krieg.

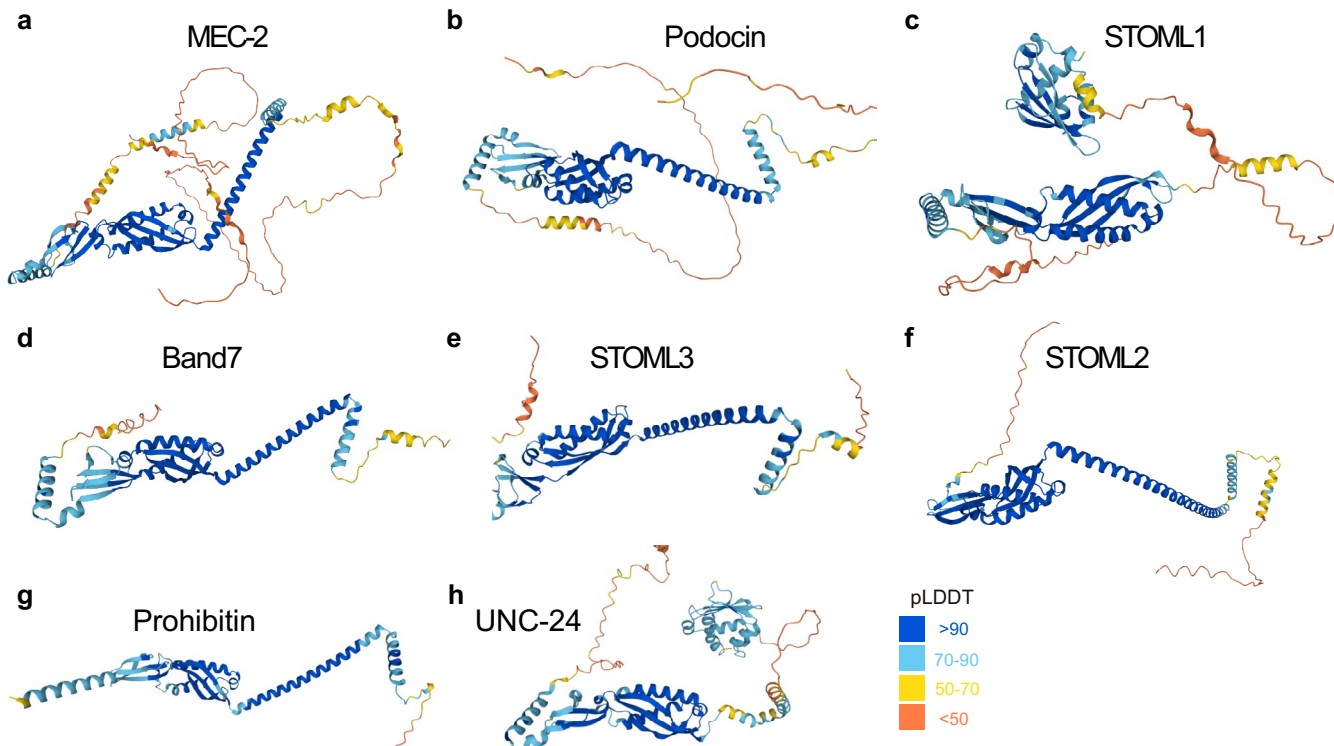

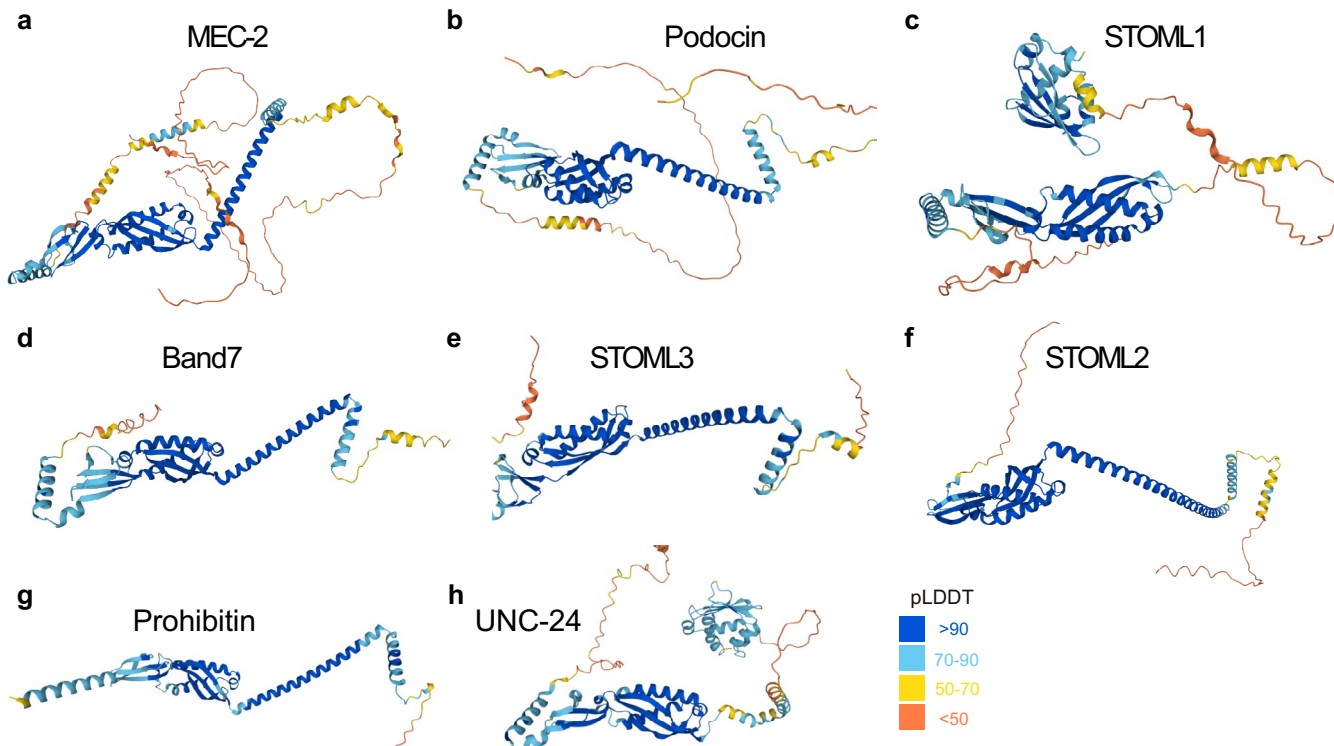

**Extended Data Fig. 1 | Stomatin protein structure predictions.** Structural prediction of **a**, *C. elegans* MEC-2 (Q27433), **b**, human Podocin (Q9NP85), **c**, human STOML1 (Q9UBI4), **d**, Band7 (P27105), **e**, human STOML3 (Q8TAV4), **f**, human STOML2 (Q9UJZ1), **g**, human Prohibitin (P35232) and **h**, C. elegans UNC-24 (G5ED76) generated with AlphaFold2. The position and conservation of the proline-rich motifs can be found in Supplementary Table 8. Colour scale of the per-residue confidence score (pLDDT) applied to all panels.

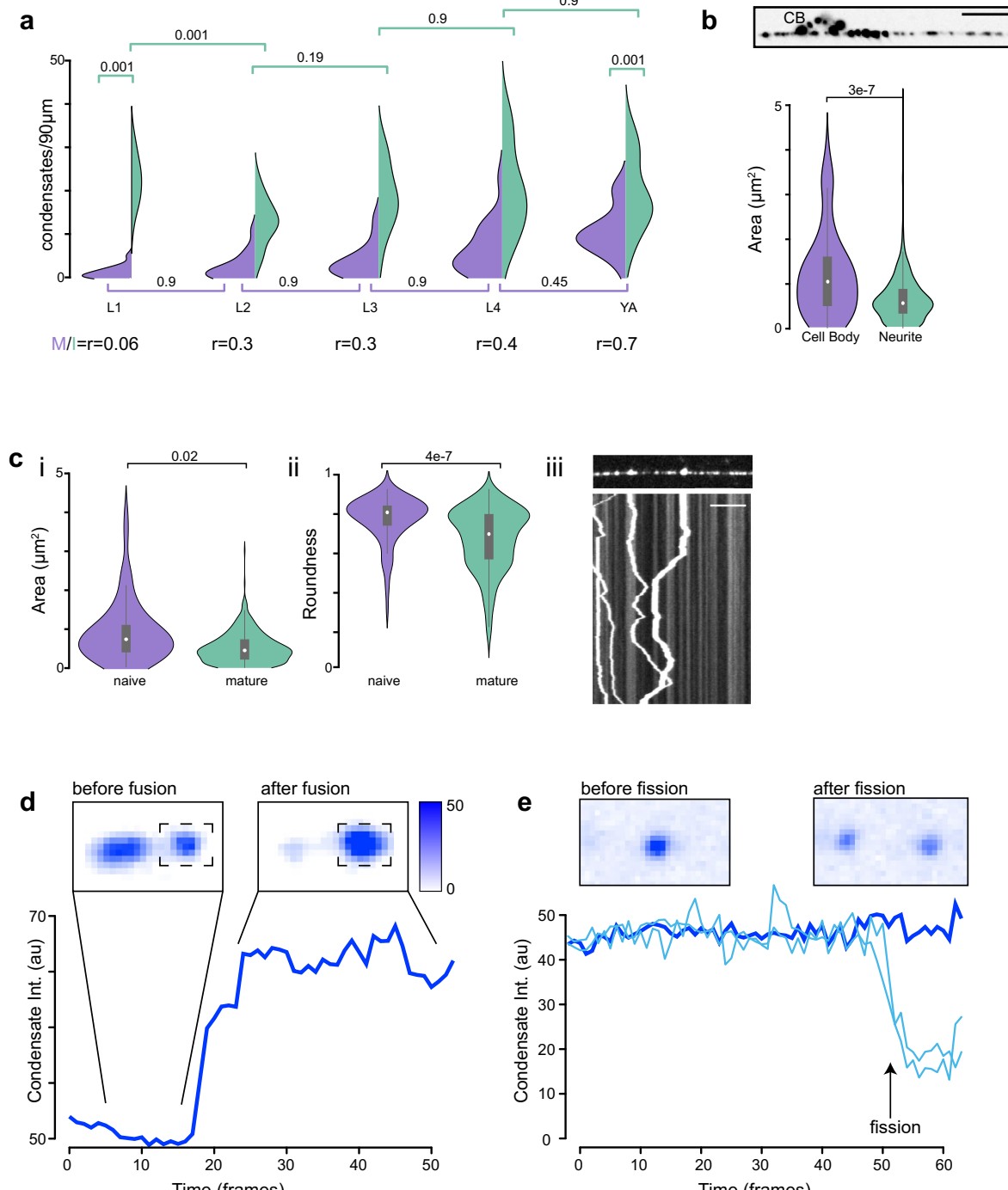

**Extended Data Fig. 2 | Characteristics of the mobile and immobile MEC-2 pool. a,** Number of mobile (purple) vs mature (green) condensates during all larval stages and first day of adulthood. The ratio r of mobile/static puncta increases significantly during development, at the same time as the length of the neurite increases. Whereas the number of static puncta remains fairly constant, the number of mobile condensates steadily increases beyond larval stage two, presumably to satisfy demand at distal sites of the growing neurites. N = 16, 18, 16, 20 and 18 TRNs from individual animals for L1, L2, L3, L4 and young adult stages, respectively. P-values derived from a t-test adjusted for multicomparison using the Tukey-Kramer post-hoc test for unbalanced data. **b,** Representative image of MEC-2 in the cell body and the size distribution inside the cell body and the neurite. Scale bar = 10 μm. N = 30 cell bodies and 22 neurites from individual animals. P-value derived from one-sided Kruskal–Wallis test. Within the violin plots, white circle mean±95% CI and grey box interquartile range Q1-Q3. **c,** Comparison of the (i) area and (ii) roundness between the naive, mobile pool and the static, mature pool. P-value derived from one-sided Kruskal–Wallis

test. iii) Representative image of a mobile punctum close to a static punctum. Representative for N = 24 TRNs from individual animals over 4 independent experiments. Scale bar = 10 μm, duration of the kymograph = 60 s. p-values derived from an unpaired, two-sided t-test. Within the violin plots, white circle and vertical bar indicate mean±95% CI and grey box interquartile range Q1-Q3. **d,** Evolution of the condensate intensity shown in the dotted square during the fusion event with the incoming condensate. Arrow points in the direction of movement towards the static condensate. Note the increase in intensity after fusion. Length of the box = 4μm. Representative of N = 8 droplets. **e,** Evolution of the condensate intensity shown in the box before and after fission event indicated with an arrow in the graph. Marine blue thick line indicates integrated intensity over the length of the quantification, showing that no material is lost. Celeste thin lines indicate intensities for each individual condensates before (same) and after fission. Note the decrease in intensity after fission. Length of the box = 5.5μm. Representative of N = 8 droplets.

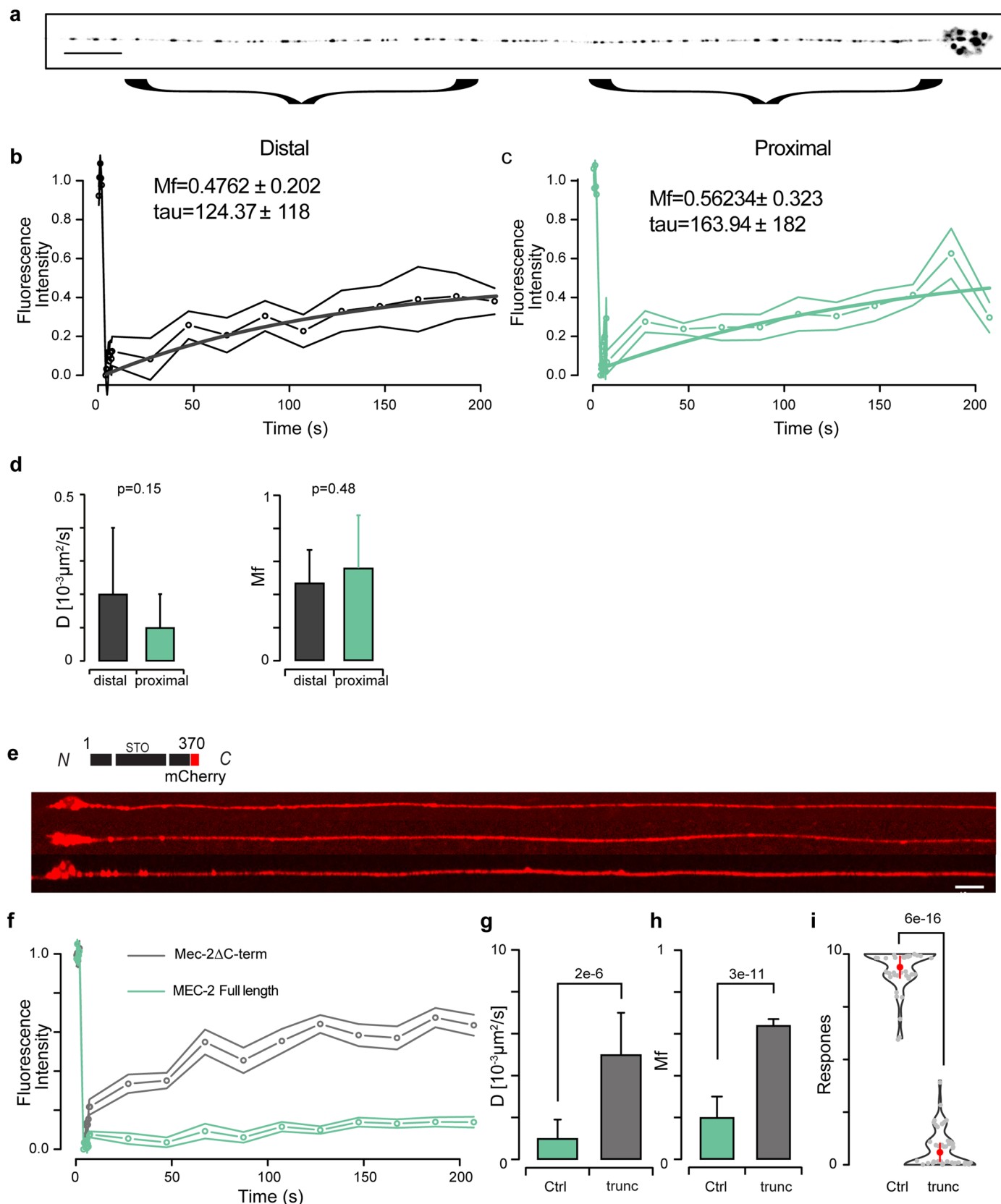

**Extended Data Fig. 3 | See next page for caption.**

**Extended Data Fig. 3 | Recovery dynamics and properties of immobile or truncated MEC-2 puncta. a**, Representative fluorescence image of an ALM TRN expressing wild-type MEC-2::mCherry. Cell body to the right. Scale bar = 10 μm. **b**, FRAP recovery dynamics (mean ± SEM) for distal, mature puncta (>150 μm from cell body). N = 11 condensates from 4 different individuals, pooled over 4 independent experiments. **c**, FRAP recovery dynamics (mean ± SEM) for proximal, mature puncta (<100 μm from cell body). N = 12 condensates from 8 different individuals, obtained from 4 independent experiments. **d**, Diffusion coefficient (D) and Mobile fraction (Mf) derived from an exponential fit to the experimental recovery curves in b and c. p-values derived from a two-sided t-test. **e**, Representative fluorescence image of an ALM TRN expressing truncated (trunc) MEC-2::mCherry. The same pattern as with the TEV cleaved construct

is observed. Scale bar = 10 μm. **f**, FRAP recovery dynamics (mean ± SEM) for WT puncta (N = 10 aggregates from 6 different individuals, pooled over 3 independent experiments) and truncated MEC-2 lacking the C-terminus (N = 12 aggregates from 7 different individuals pooled over 2 independent experiments). **g, h**, Timescale of (g) diffusion coefficient (D) and (h) mobile fraction (Mf) derived from an exponential fit to the experimental recovery curves in f. p-value derived from a two-sided t-test. **i**, Average number of responses (out of ten) from animals expressing wild-type (N = 50 individuals tested 10 times each, pooled over 3 independent experiments) and truncated MEC-2::mCherry (N = 60 individuals tested 10 times each, pooled over 3 independent experiments). Red circle indicates mean±95% CI. p = 6.e-16, two-sided Mann–Whitney U-test.

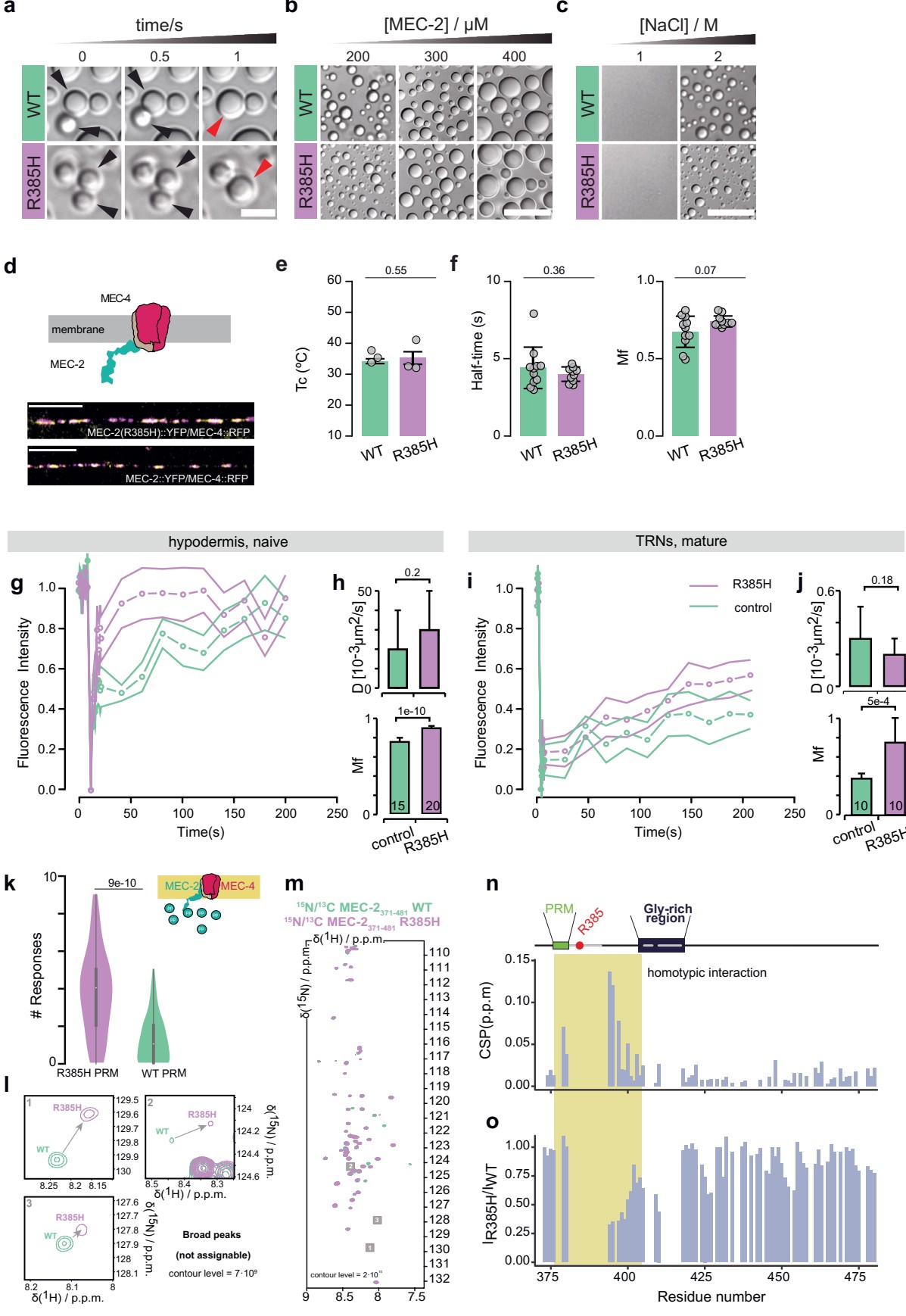

**Extended Data Fig. 4 | See next page for caption.**

**Extended Data Fig. 4 | MEC-2 proline-rich domain is essential for the sense of touch. a**, DIC microscopy images showing fusion events of 300 μM MEC-2 WT and R385H with 2 M NaCl at 37 °C. Scale bar = 5 μm. **b**, DIC microscopy images of MEC-2 WT and R385H at increasing protein concentrations of 200, 300 and 400 μM, with 2 M NaCl at 37 °C. Scale bar = 20 μm. Representative for N = 3 different field of view from one preparation. (**c**) DIC microscopy images of 200 μM MEC-2 WT and R385H with 1 and 2 M NaCl at 37 °C. Scale bar = 20 μm. Representative for N = 3 independent experiments. **d**, Representative images for at least N = 10 individuals of wild-type or MEC-2 (R385H)::YFP mutant in green and MEC-4::RFP in red. Colocalization indicated in yellow. Scale bar = 10 μm. **e**, Tc value of the apparent absorbance measurement as a function of temperature of 200 μM MEC-2 WT and R385H mutant with 2 M NaCl (N = 3 independent experiments). p-values derived from a two-sided t-test. Average temperature Mean ± SD. **f**, Recovery half-time and mobile fraction (Mf) of MEC-2 WT (N = 11 droplets) and R385H mutant (N = 10 droplets) quantified from one FRAP experiment *in vitro* of a 370 μM sample with 2 M NaCl at 20 °C. p-values derived from a two-sided t-test. Average half-time and Mf Mean ± SD. **g–j** Time course of the FRAP dynamics for control and R385H MEC-2 mutants after ectopic expression in the hypodermis (g) and TRNs (i). Mean ± SD **h,j** Diffusion

coefficient (D) of the recovery and mobile fraction (Mf) of the recovery for both wild type and R385H mutant MEC-2 after FRAP in the (h) hypodermis and (i) TRNs. N = 15 condensates from 7 different WT individuals over 4 independent experiments (g, h); 20 condensates from 6 different R385H mutant individuals over 6 independent experiments (g, h); 10 aggregates from 5 different WT individuals over 3 independent experiments (i, j) and 10 aggregates from 3 different R385H individuals over 2 independent experiments (i, j). p-values derived from a two-sided t-test. **k**, Touch response of wild type animals with an overexpression of R385H mutant or wild type proline-rich (PRM) MEC-2 motifs specifically in TRNs. Mean ± SD, N = 60 individuals per condition tested 10 times over 3 independent experiments. p-value derived from one-sided Kruskal–Wallis test. Scheme of the experiment at the top right. Within the violin plots, white circle and vertical bar indicate mean±95% CI and grey box interquartile range Q1-Q3. **l, m** Low-intensity (non-assignable) signals from 2D NMR spectra of the WT (green) and the R385H mutant (purple) MEC-2 C-terminal domain (l) derived from the 2D 1H-15N NMR correlation spectra of MEC-2 WT and R385H mutation (m). **n, o**, Chemical shift (CSP, mn) and intensity ratio (o) for each residue for the comparison between WT and R385H. The lower intensities around the PRM indicate a homotypic interaction between different MEC-2 (R385H) molecules.

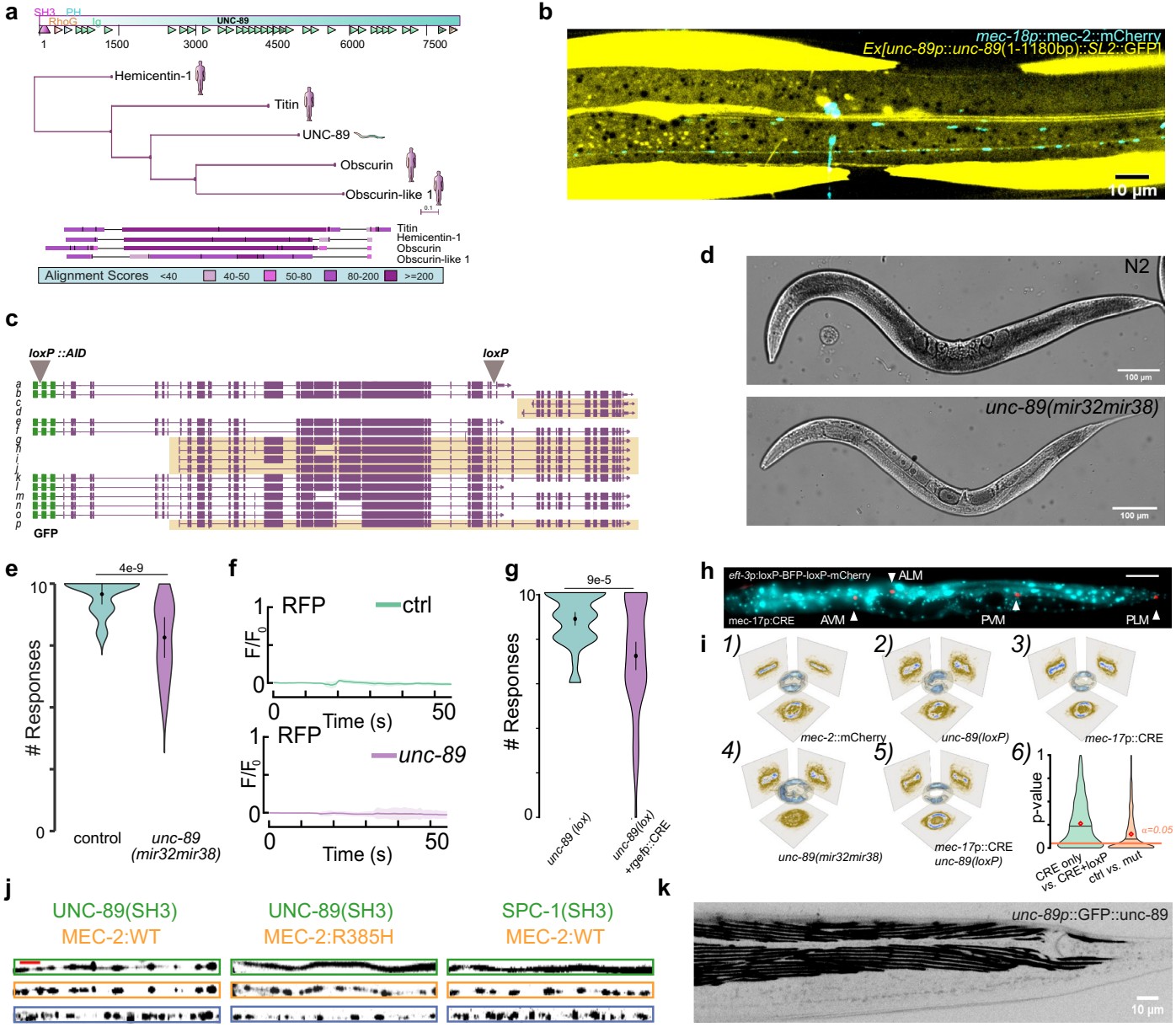

**Extended Data Fig. 5 | Non-muscular function of UNC-89. a,** Scheme of UNC-89 protein domains, BLAST against Homo sapiens genome and representation of the closest proteins Hemicentin-1, Titin, Obscurin and Obscurin-like 1 in a phylogenetic tree and protein alignment. **b,** Representative micrograph for N = 10 individuals of an animal expressing an *unc-89* promotor trap encompassing 4265 bp and the first 1180 bp of the genomic fragment (Exon1-Intron1-Exon2-Intron2-Exon3) showing expression of the largest isoforms in muscles and neurons. **c,** Genomic organization of GFP-tagged *unc-89* locus and location of the two loxP sites. Yellow shadow highlights the remaining isoforms in the *unc-89(mir32)* allele, which only knocks out the largest isoforms that contain the SH3 domain. *mir32* was generated using a frameshift causing an aberrant initiation site. **d,** Micrograph for N = 10 individuals comparing young adult N2 and *unc-89(mir32)* animals. Scale bar = 100 μm. **e,** Touch response of *unc-89(mir32)* KO allele compared to control wild type animals. Circle indicates mean, vertical bar indicates 95% CI, N = 60 individuals per condition tested 10 times each, over 3 independent experiments. p-value derived from a one-sided Kruskal–Wallis test. **f,** Fluorescence intensity vs time of the calcium-independent fluorophore in the mechanical stimulation experiment showed in Fig. 3b. **g,** Touch response of pan-neuronal (*rgef-1*p::CRE) knockout of UNC-89 compared to loxP-flanked control animals in absence of CRE recombinase. Circle indicates mean, vertical bar indicates 95% CI, N = 60 animals. p-value derived from a

one-sided Kruskal–Wallis test. **h,** Representative image of a *mec-17*p::CRE expressing animals with a CRE reporter (lox::BFP::lox::mCherry). Only cells in which CRE was active are shown in red, indicating successful excision of the BFP, leading to transcription of the mCherry reporter. See also ref. 88 for details and representative example of the pan-neuronal *rgef-1*p::CRE reporter. Scale bar = 50 μm. **i,** Quantification of animal locomotion using the Eigenworm approach for 1) MEC-2:mCherry transgenics; 2) loxP-flanked *unc-89* genomic locus; 3) animals expressing mec-17p::CRE; 4) mutant animals lacking the SH3-containing isoforms of UNC-89; 5) animals lacking UNC-89 in TRNs and, 6) distributions of bootstrapped p-value from a two-sided t-test for the comparison 3 vs 5 and 1 vs 4. Black horizontal line indicates median of 1000 independent p-value calculations, red diamond indicates the mean. Orange line is the alpha-level of significance. See Methods and ref. 88 for details. **j,** Representative images for N = 10 neurons of individual TRNs expressing a translational GFP fusion of the SH3 domain derived from UNC-89 and (i) MEC-2::.mCherry wild-type, MEC-2::(R385H)::mCherry mutant, and SPC-1 α-spectrin SH3 domain::GFP together with the wild type MEC-2::mCherry, and a scrambled version of MEC-2::mCherry. Scale bar in red for all panels = 5 μm. **k,** Representative fluorescence image for N = 10 individuals of an animal with an N-terminal GFP tag at the endogenous locus of *unc-89* in frame with the SH3 domain. Scale bar = 10 μm.

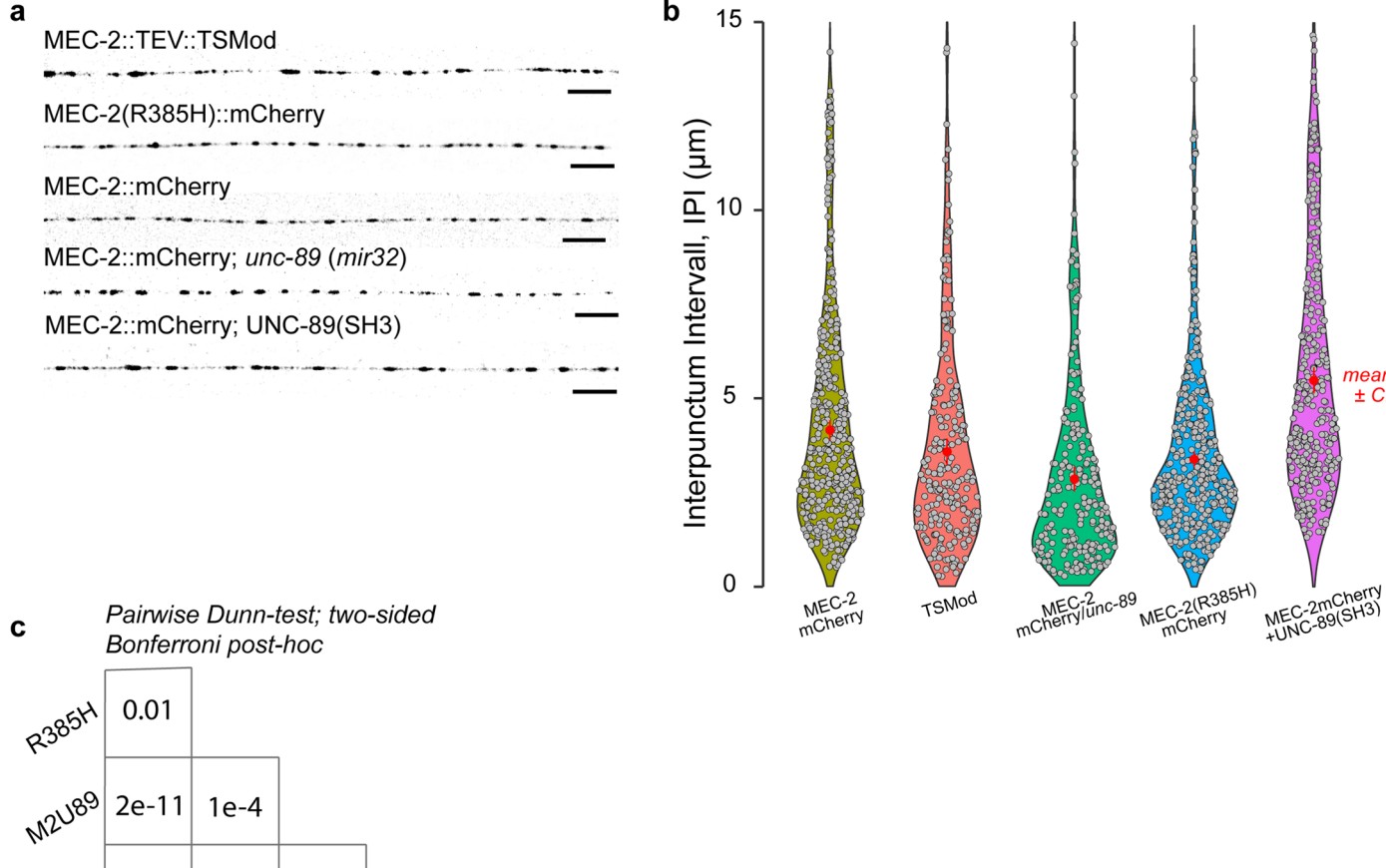

**Extended Data Fig. 6 | Distribution of MEC-2 in neurites of control and mutant animals. a**, Representative examples for the indicated MEC-2 alleles and genotypes. Scale bar = 10 μm. **b, c**, Violin plot for all mec-2 alleles and mutant backgrounds (b) and table indicating the p-values of the pairwise comparison (c) of their distribution from a two-sided Dunn's test of multiple comparisons using rank sums with Bonferroni's adjustment for multiple comparisons. The C-terminal truncated MEC-2 was not tested, as no interpunctum interval (IPI) could be extracted (IPI = 0 for a continuous distribution). Mean indicated as red circle with vertical bar indicating the 95% CI of the mean. N = 234 measurements from 5 MEC-2::mCherry individuals over 2 independent experiments, 4 TSMod individuals from 1 experiment, 8 *unc-89* individuals from 1 experiment, 7 R385H individuals from 1 experiment and 6 *unc-89* SH3 (overexpressed) individuals from 2 independent experiments.

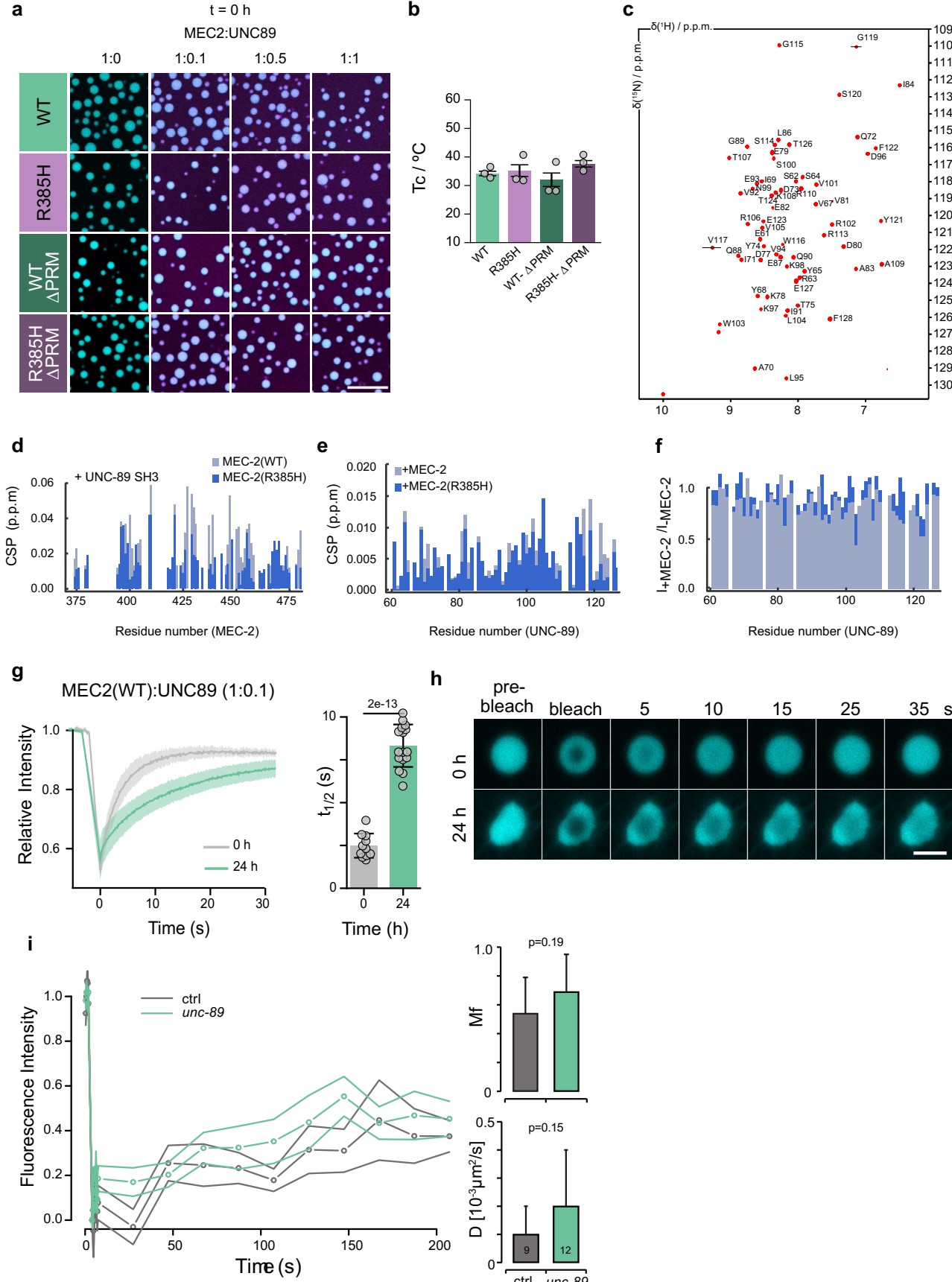

**Extended Data Fig. 7 | See next page for caption.**

**Extended Data Fig. 7 | *In vitro* phase separation of MEC-2 and binding to UNC-89 SH3 domain. a**, Confocal fluorescence microscopy images of 200 µM MEC-2 C-terminus (WT, MEC-2 (R385H), MEC-2 (ΔPRM); or MEC-2 (ΔPRM,R385H) mutant) labelled with Alexa Fluor 647 together with UNC-89 SH3 domain labelled with DyLight 488, at different molar ratios MEC-2:UNC-89 (as indicated in the figure), with 2 M NaCl at 37 °C. Pictures taken immediately after droplet formation (t = 0 h). Scale bar = 20 µm. Representative for N = 3 independent experiments. **b**, Tc value of the apparent absorbance measurement as a function of temperature of 200 µM MEC-2 WT and MEC-2 (R385H), MEC-2 (ΔPRM); or MEC-2 (ΔPRM,R385H) mutant proteins with 2 M NaCl (N = 3 independent measurements). Average Tc Mean ± SD. **c**, 61/62 HN assignment annotated 2D NMR spectrum of the UNC-89 SH3 domain (BMRD ID: 51490). The intensities of G119 and V117 are displayed in Fig. 4d. **d**, Chemical shift perturbations of MEC-2 C-terminal wild type (light blue) or R385H mutant (blue) upon binding to SH3 domain of UNC-89 (1:10 molar ratio). **e**, Chemical shift perturbations of UNC-89 upon binding to wild type (light blue) and R385H mutant (blue) MEC-2 (1:9 molar ratio). **f**, Intensity ratio of the NMR spectra of the SH3 domain of UNC-89 in the presence or the absence of the C-terminal domain of MEC-2 (WT (light blue) and R385H mutant (blue)). **g**, *In vitro* FRAP experiment for WT MEC-2:UNC-89 (1:0.1) condensates immediately after formation (t = 0 h, N = 11 condensates from one experiment) and after 24 hours incubation (t = 24 h, N = 16). The half-times of the recovery (t1/2) are displayed to the right. p-values derived from a two-sided t-test. Average fluorescence intensity and half-time Mean ± SD. **h**, Representative fluorescence microscopy images of the FRAP experiment in panel g. Scale bar=10µm. **i**, FRAP for MEC-2 static condensates in the *unc-89* background. At the right, timescale of Mobile fraction (Mf) and diffusion coefficient (D) derived from an exponential fit to the experimental recovery curves at the left. N = 9 condensates from 5 different WT individuals over 3 independent experiments and 12 condensates from 5 different *unc-89* individuals over 3 independent experiments. p-value derived from two-sided t-test.

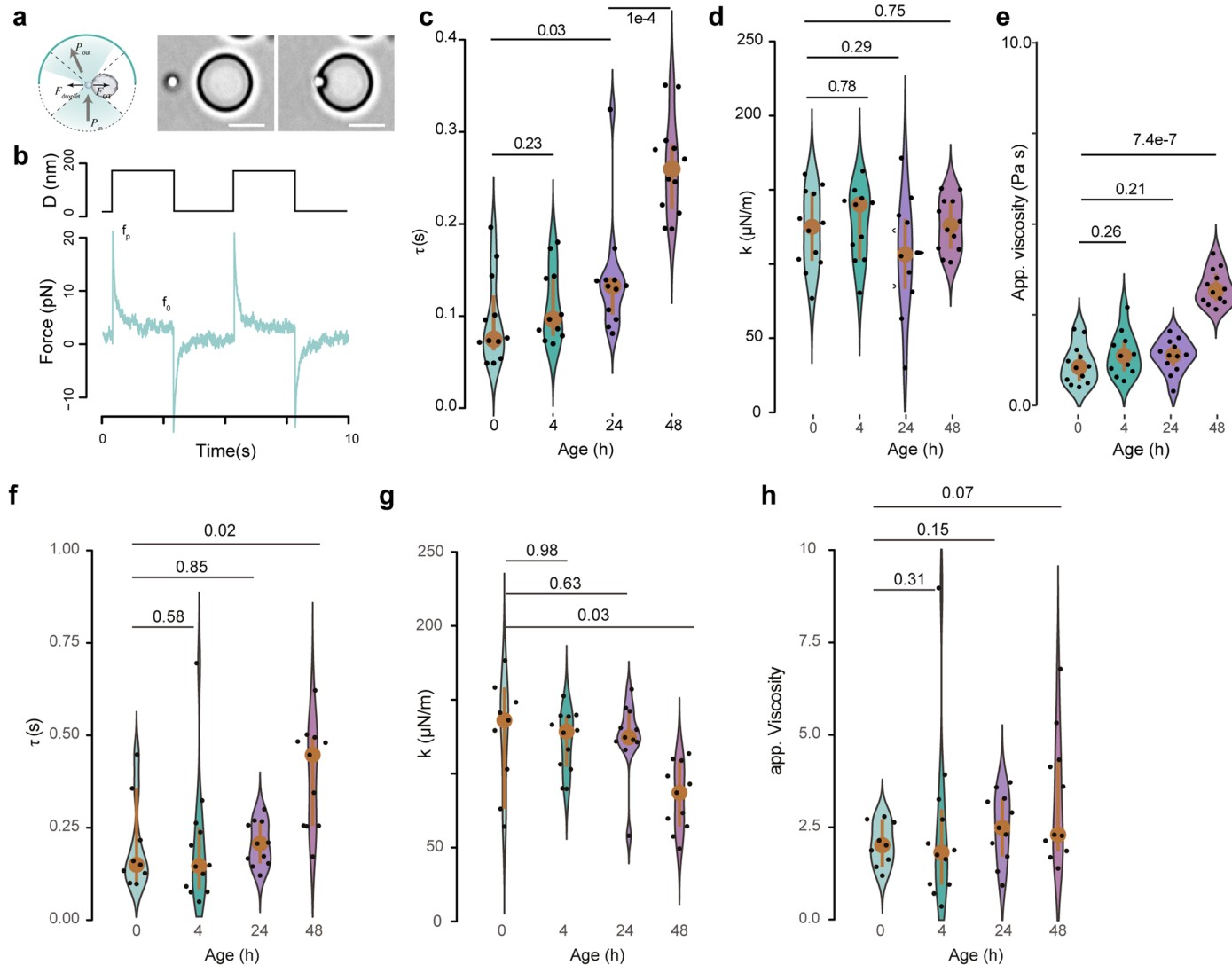

**Extended Data Fig. 8 | Step indentation reveals viscoelastic maturation of MEC-2 and its mutants. a**, Schematic of the optical tweezer based indentation assay, during which a trapped microsphere is driven onto an immobilized droplet. Two representative pictures showing the sphere before and after droplet contact. Scale bar = 5 μm. **b**, Representative force-time signal of a typical indentation test showing stress relaxation. Upper graph indicates trap trajectory, lower graph stress relaxation. fp and f0 are the peak and steady state force after relaxation, respectively. **c**, Time decay constant measured for step-stress relaxation experiments on protein condensates of increasing age. For panels c-e: N = 12 (0 h); 11 (4 h); 12 (24 h) and 12 (48 h) droplets from one experiment representative for 2 independent trials. **d**, Stiffness measured on the same protein condensates as in (c)). **e**, Viscosity (η = k·τ) of the droplets as derived from the measurements in (c) and (d). **f**, Time decay constant measured for step-stress relaxation experiments on protein condensates of increasing age, formed from MEC-2 (R385H) mutant. For panels f-h: N = 9 (0 h); 12 (4 h); 11 (24 h) and 11 (48 h) droplets from one experiment. **g**, Stiffness measured on the same protein condensates as in (f). **h**, Viscosity (η = k·τ) of the droplets as derived from the measurements in (f) and (g). In all panels, p-values above horizontal bar derived from pairwise two-sided non-parametric Mann–Whitney U-test with 0 h as control group or indicated otherwise. In all subpanels, red circle indicates the median±95% CI.

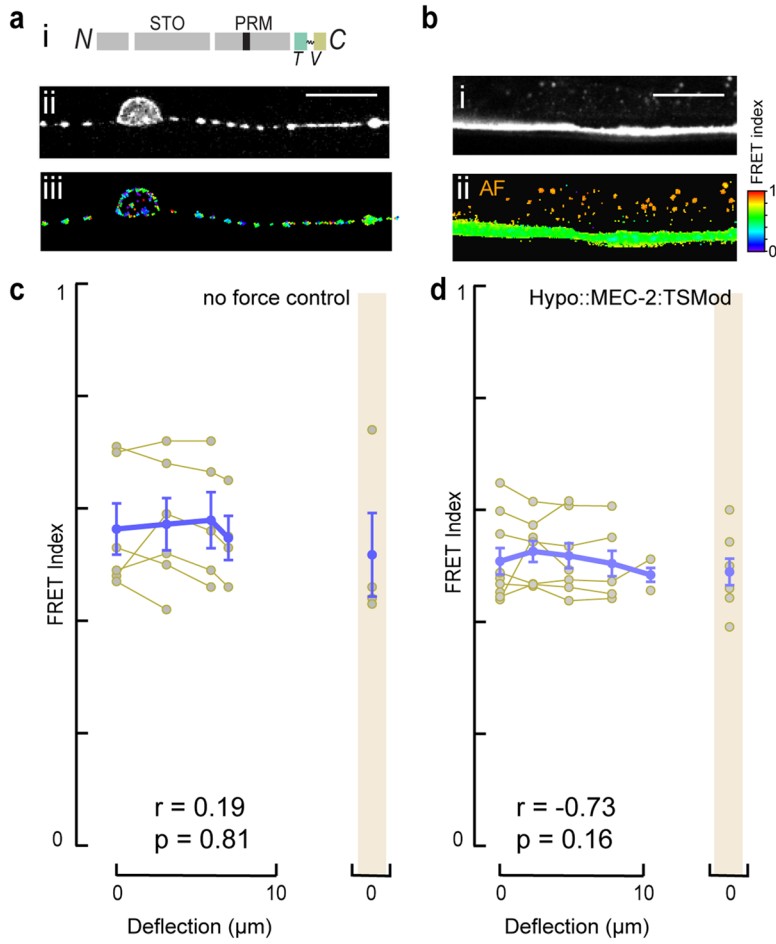

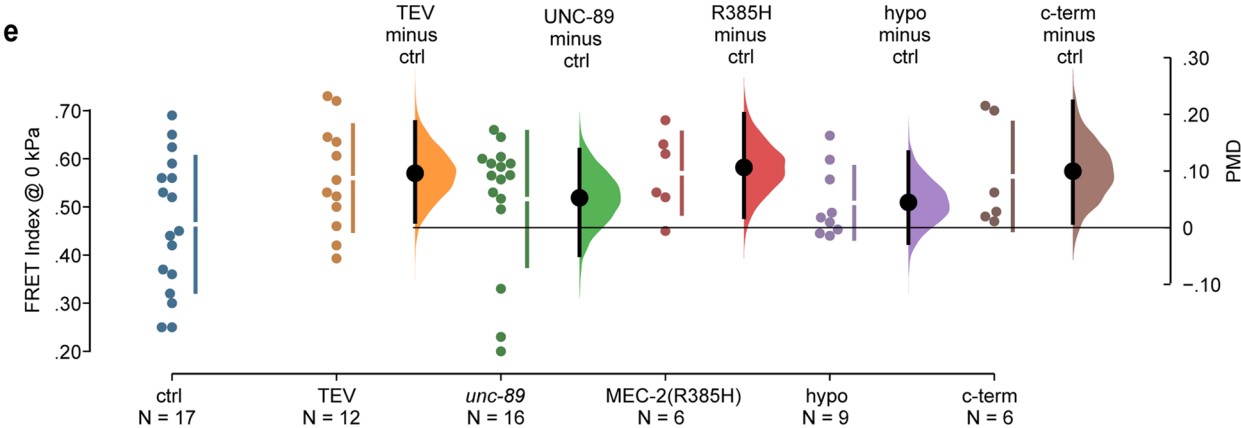

**Extended Data Fig. 9 | MEC-2::TSMod reports forces during body wall touch.**
**a**, i) Scheme of the C-terminally fused MEC-2::cTSMod as a no-force control, and the ii) representative acceptor image and iii) FRET map. STO, stomatin domain; T, donor fluorophore; V, acceptor fluorophore; PRM, proline-rich motif; TRAF, tumour necrosis factor receptor associated factor, TEV, Tobacco etch virus cleavage site. Scale bar = 10 μm. **b**, FRET imaging of MEC-2::TSMod ectopically expressed in the hypodermis. i) Representative acceptor image. ii) FRET map. AF indicates autofluorescence of the gut. Scale bar = 10 μm. **c,d**, FRET index changes with increasing pressure applied to the body wall of (c) C-terminally tagged MEC-2 (N = 6 animals pooled over 5 independent experiments) serving as a no-force control and (d) MEC-2::TSMod expressed in the hypodermis N = 9 animals pooled over 3 independent experiments) as a surrogate for naïve MEC-2.

Yellow lines represent sequences acquired for individual animals, purple circles the corresponding mean ± SEM. r = Pearson's correlation coefficient calculated between FRET vs. cuticle deformation. p-value indicates statistically significance of the correlation coefficient derived from a two-sided t-test. **e**, FRET index quantification for the indicated constructs derived from animals trapped inside the microfluidic chip in absence of a stimulating pressure. Each dot is a measurement from a single animal. The paired mean difference between ctrl and the experimental group is plotted on a floating axes on the right of each dot plot as a bootstrap sampling distribution. The mean difference is depicted as a dot; the 95% CI is indicated by the ends of the vertical error bar. Overlap of the CI interval with zero indicates insignificant effect size.

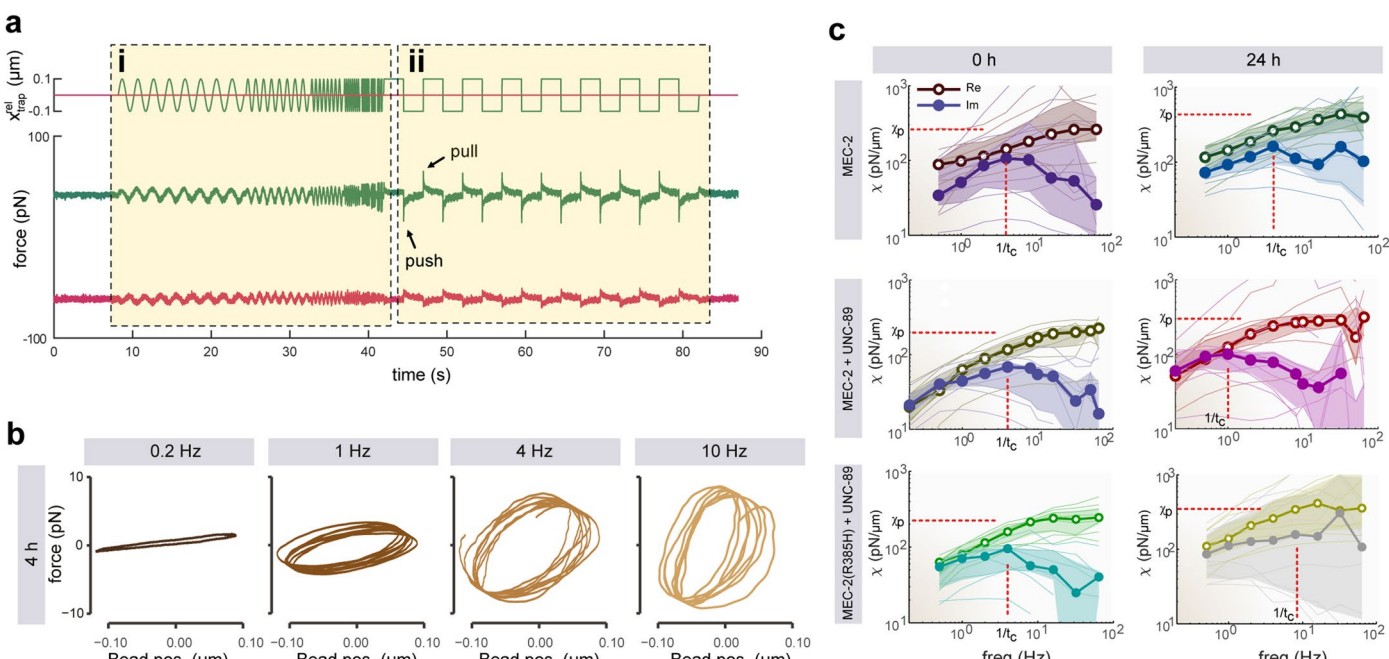

**Extended Data Fig. 10 | Active microrheology using oscillating optical traps.**
**a**, Typical force trace obtained from the sinusoidal oscillation series, followed
by step-stress relaxation measurements. Top - trap trajectories along the x
direction for trap 1 (green) and trap 2 (red). **b**, Lissajous figures acquired by active
microrheology on early MEC-2 condensates immediately after formation relating
the force and displacement for four different frequencies. **c**, Complex spring
constant of MEC-2 droplets. Open (filled) circles correspond to the storage (loss)
moduli. Lines are individual measurements and the shadowed area indicates the
±25% quantiles around the median. The back line indicates the spring constant of
the droplet at f = 0.

# Reporting Summary

## Statistics

For all statistical analyses, confirm that the following items are present in the figure legend, table legend, main text, or Methods section.

| n/a | Confirmed | |
|---|---|---|
| ☐ | ☒ | The exact sample size ($n$) for each experimental group/condition, given as a discrete number and unit of measurement |
| ☐ | ☒ | A statement on whether measurements were taken from distinct samples or whether the same sample was measured repeatedly |
| ☐ | ☒ | The statistical test(s) used AND whether they are one- or two-sided<br>*Only common tests should be described solely by name; describe more complex techniques in the Methods section.* |
| ☒ | ☐ | A description of all covariates tested |
| ☐ | ☒ | A description of any assumptions or corrections, such as tests of normality and adjustment for multiple comparisons |
| ☐ | ☒ | A full description of the statistical parameters including central tendency (e.g. means) or other basic estimates (e.g. regression coefficient) AND variation (e.g. standard deviation) or associated estimates of uncertainty (e.g. confidence intervals) |
| ☐ | ☒ | For null hypothesis testing, the test statistic (e.g. $F$, $t$, $r$) with confidence intervals, effect sizes, degrees of freedom and $P$ value noted<br>*Give P values as exact values whenever suitable.* |
| ☒ | ☐ | For Bayesian analysis, information on the choice of priors and Markov chain Monte Carlo settings |
| ☒ | ☐ | For hierarchical and complex designs, identification of the appropriate level for tests and full reporting of outcomes |
| ☐ | ☒ | Estimates of effect sizes (e.g. Cohen's $d$, Pearson's $r$), indicating how they were calculated |

*Our web collection on statistics for biologists contains articles on many of the points above.*

## Software and code

Policy information about availability of computer code

| | |
|---|---|
| Data collection | All data collection software (image acquisition software [HCImage version 4.4.2.7, μManager], NMR software (TopSpin version 4.0.8) , Optical tweezer LightAce software) is described in detail in the methods for each experiment and are open source or commercially available. No custom acquisition software was used. |
| Data analysis | Optical Tweezer and Calcium data were analyzed using custom-written Matlab scripts (Version 2021, Das et al, 2021); Ratiometric FRET data analysis was performed in IgorPro 6.37 as described in Krieg et al. 2014; and NMR data analyses were performed as described in detail using published open source software (CcpNmr version-3 and delta2D version-1). BLAST of protein sequences was performed using NCBI. FRAP analysis was performed with the easyFRAP online tool (Koulouras et al. 2018; version 1.11). Rheological analyses were perfomed using custom-written Matlab scripts (R2019b). Statistics and data plotting were done in R (version 4.2.2; 2022-10-31) with the package "ks" version 1.14.0 and Python (version 3). Image analysis was performed in ImageJ (version 1.53f51). All custom computer scripts are available on GitLab https://gitlab.icfo.net/rheo/Tweezers/droplet or supon request. No specialized standalone software was developed as part of this manuscript. |

For manuscripts utilizing custom algorithms or software that are central to the research but not yet described in published literature, software must be made available to editors and reviewers. We strongly encourage code deposition in a community repository (e.g. GitHub). See the Nature Portfolio guidelines for submitting code & software for further information.

## Data

Policy information about availability of data

All manuscripts must include a data availability statement. This statement should provide the following information, where applicable:
- Accession codes, unique identifiers, or web links for publicly available datasets
- A description of any restrictions on data availability
- For clinical datasets or third party data, please ensure that the statement adheres to our policy

The chemical shifts in the NMR data of MEC-2 and UNC-89 was deposited to Biological Magnetic Resonance Data Bank (https://bmrb.io/) under accession codes ID:51491 and ID:51490 respectively. Source data are provided with this study. The publicly available datasets used in this study are accessible through wormbase.org version WS289. Protein sequences were obtained from UniProt (O01761). Calcium imaging data of the animals under a mechanical stimulus has been deposited into zenodo.org under doi:10.5281/zenodo.8163972. All other data supporting the findings of this study are available from the corresponding author on reasonable request.

## Human research participants

Policy information about studies involving human research participants and Sex and Gender in Research.

| | |
|---|---|
| Reporting on sex and gender | n/a |
| Population characteristics | n/a |
| Recruitment | n/a |
| Ethics oversight | n/a |

Note that full information on the approval of the study protocol must also be provided in the manuscript.

# Field-specific reporting

Please select the one below that is the best fit for your research. If you are not sure, read the appropriate sections before making your selection.

☒ Life sciences ☐ Behavioural & social sciences ☐ Ecological, evolutionary & environmental sciences

For a reference copy of the document with all sections, see nature.com/documents/nr-reporting-summary-flat.pdf

# Life sciences study design

All studies must disclose on these points even when the disclosure is negative.

| | |
|---|---|
| Sample size | For behavioral data collection, a consistent number of 30 animals per genotype and condition was investigated. No a priori method or power analysis was applied to determine an optimal sample size. Sample size was chosen according to previous published standards (Krieg, NCB,2014), to remain experimentally feasible and avoid spurious overfitting. |
| Data exclusions | No statistical outliers have been identified. |
| Replication | Behavioral data is presented as a triplicate of 10 animals assayed in each experiment (each animal was tested 10 times). Replicates, in which control animals did not meet the a priori requirements (e.g. positive control not positive or negative control not negative such as  a failure of a positive control group to respond to blue light in ATR condition in an optogenetics experiment) were not considered in the analysis. All attempts at replication were successful. |
| Randomization | Whenever possible, experimental conditions were randomized to avoid history effects in the optical tweezer mechanics experiments. To avoid adaptation of the animals to repetitive touch stimuli, a minimum interstimulus intervall of 10s was applied. Animals from different genotypes in each experiment were subjected to the same treatment conditions on the same batch of NGM agar plates to control for variations coming from the food source or subtle differences in the environment conditions. No specific order was followed in any of the experiments unless specifically indicated in the methods. All other samples were not randomized. |
| Blinding | Behavioral experiments (manual scored touch tests) were performed blinded with respect to genotype and animal treatment. Other experiments were not performed blind to treatment or genotype, as they were performed with computer-assisted analysis. |

# Reporting for specific materials, systems and methods

We require information from authors about some types of materials, experimental systems and methods used in many studies. Here, indicate whether each material, system or method listed is relevant to your study. If you are not sure if a list item applies to your research, read the appropriate section before selecting a response.

## Materials & experimental systems

| n/a | Involved in the study |
|---|---|
| ☒ | ☐ Antibodies |
| ☒ | ☐ Eukaryotic cell lines |
| ☒ | ☐ Palaeontology and archaeology |
| ☐ | ☒ Animals and other organisms |
| ☒ | ☐ Clinical data |
| ☒ | ☐ Dual use research of concern |

## Methods

| n/a | Involved in the study |
|---|---|
| ☒ | ☐ ChIP-seq |
| ☒ | ☐ Flow cytometry |
| ☒ | ☐ MRI-based neuroimaging |

## Animals and other research organisms

Policy information about studies involving animals; ARRIVE guidelines recommended for reporting animal research, and Sex and Gender in Research

| | |
|---|---|
| Laboratory animals | Caenorhabditis elegans, N2 and mutants listed in Supplementary Table 4; Experiments were restricted to young adult hermaphrodites, unless otherwise indicated. |
| Wild animals | No wild animals have been used in this study. |
| Reporting on sex | Unless otherwise indicated, all experiments have been conducted on young adult hermaphrodite animals. |
| Field-collected samples | No field samples were used in this study. |
| Ethics oversight | Nematodes are excempt from ethics approval. |

Note that full information on the approval of the study protocol must also be provided in the manuscript.

