## [Peer Review File · Nature Cell Biology]

Peer Review Information

Journal: Nature Cell Biology

Manuscript Title: A MEC-2/Stomatin condensate liquid-to-solid phase transition controls neuronal mechanotransduction during touch sensing

Corresponding author name(s): Professor Michael Krieg

Editorial Notes:

Reviewer Comments & Decisions:

Decision Letter, initial version:

*Please delete the link to your author homepage if you wish to forward this email to co-authors.

Dear Professor Krieg,

I apologize for the delay. Your manuscript, "A rigidity phase transition of Stomatin condensates governs a switch from transport to mechanotransduction", has now been seen by 3 referees, who are experts in interneuron communication (referee 1); biomolecular condensation (referee 2); and mechanotransduction and mechanosensation in vivo (referee 3). As you will see from their comments (attached below) they find this work of potential interest, but have raised substantial concerns, which in our view would need to be addressed with considerable revisions before we can consider publication

in Nature Cell Biology.

Nature Cell Biology editors discuss the referee reports in detail within the editorial team, including the chief editor, to identify key referee points that should be addressed with priority, and requests that are overruled as being beyond the scope of the current study. To guide the scope of the revisions, I have listed these points below. We are committed to providing a fair and constructive peer-review process, so please feel free to contact me if you would like to discuss any of the referee comments further.

I should stress that the referees' concerns point to incomplete characterization of the rigidity phase transition in vivo and unclear mechanism and unclear links to mechanosensation which would need to be addressed with experiments and data, and reconsideration of the study for this journal and re-engagement of referees would depend on strength of these revisions.

In particular, it would be essential to:

A) Characterize the rigidity phase transition, rigidity percolation, and differential dynamics of MEC2 during development in vivo (all reviewers)

B) Assess the effects of this rigidity phase transition on mechanotransduction and/or mechanosensation (all reviewers)

C) All other referee concerns pertaining to strengthening existing data, providing controls, methodological details, clarifications and textual changes, should also be addressed.

D) Finally please pay close attention to our guidelines on statistical and methodological reporting (listed below) as failure to do so may delay the reconsideration of the revised manuscript. In particular please provide:

We would be happy to consider a revised manuscript that would satisfactorily address these points, unless a similar paper is published elsewhere, or is accepted for publication in Nature Cell Biology in the meantime.

- ensure that it conforms to our format instructions and publication policies (see below and www.nature.com/nature/authors/).

- provide a point-by-point rebuttal to the full referee reports verbatim, as provided at the end of this letter.

- provide the completed Editorial Policy Checklist (found here <https://www.nature.com/authors/policies/Policy.pdf>), and Reporting Summary (found here <https://www.nature.com/authors/policies/ReportingSummary.pdf>). This is essential for reconsideration of the manuscript and these documents will be available to editors and referees in the event of peer review. For more information see <http://www.nature.com/authors/policies/availability.html> or contact me.

Nature Cell Biology is committed to improving transparency in authorship. As part of our efforts in this direction, we are now requesting that all authors identified as 'corresponding author' on published papers create and link their Open Researcher and Contributor Identifier (ORCID) with their account on the Manuscript Tracking System (MTS), prior to acceptance. ORCID helps the scientific community achieve unambiguous attribution of all scholarly contributions. You can create and link your ORCID from the home page of the MTS by clicking on 'Modify my Springer Nature account'. For more information please visit please visit www.springernature.com/orcid.

[Redacted]

We would like to receive a revised submission within six months. We would be happy to consider a revision even after this timeframe, however if the resubmission deadline is missed and the paper is eventually published, the submission date will be the date when the revised manuscript was received.

We hope that you will find our referees' comments, and editorial guidance helpful. Please do not hesitate to contact me if there is anything you would like to discuss.

Best wishes,

Daryl Jason David

Daryl J.V. David, PhD

Senior Editor, Nature Cell Biology
Consulting Editor, Nature Communications
Nature Portfolio

Heidelberger Platz 3, 14197 Berlin, Germany
Email: daryl.david@nature.com

ORCID: <https://orcid.org/0000-0002-9253-4805>

Reviewers' Comments:

Reviewer #1:

Remarks to the Author:

Although a large body of work suggests that biomolecular condensates ensuing from liquid-liquid phase separation mature into various material states, how this aging process is controlled and if the naive and mature phases can have differential functions is currently unknown. Using *Caenorhabditis elegans* as a model, the authors demonstrated that MEC-2 Stomatin undergoes a rigidity phase transition during maturation from fluid to viscoelastic, glass-like condensates that facilitate either transport or mechanotransduction.

I agree that that the authors' data demonstrate a novel function for rigidity maturation during mechanotransduction. The authors used various *in vitro* and *in vivo* approaches and provided numerous data to support their claims, Therefore, I would like to support the eventual acceptance of this manuscript to NCB.

Nevertheless, I still think that the two key data are missing. I got an overall impression that the role for the UNC-89 SH3 domain in promoting MEC-2 maturation (rigidity phase transition) was not sufficiently demonstrated *in vivo*.

Majors:

1) Characterization of MEC-2(R385H) point mutant *in vivo*: In Fig. 1, the authors showed a rigidity phase transition of WT MEC-2 and its difference from the C-terminally truncated MEC-2 mutant. However, the authors did NOT show a sol-gel transition of MEC-2(R385H) mutant. This is critical, because this mutant is the key mutant the authors used for characterizing the interaction with UNC-89 SH3 domain in Fig. 3.

2) The authors stated in Abstract: "This switch is promoted by the SH3 domain of UNC 89/Titin/Obscurin through a direct interaction with MEC-2 and suggests a physiological role for a percolation transition in force transmission during body wall touch." I think that the key supporting data for this claim is missing. If this claim is current, we would expect a reduced sol-gel transition of WT MEC-2 (shown in Fig.1) in the absence of UNC-89. That is, I think that the authors should examine the rigidity phase transition of WT MEC-2 in the *unc-89* mutant background.

Minors:

1) line 250: weakens a heterotypic interaction between MEC-2 molecules that kinetically stabilized the MEC-2... Here, "heterotypic" interaction should read "homotypic" interaction?

2) line 251-255: these sentences seem to largely consist of the authors' speculation. It may be better to discuss these in the discussion section.

Reviewer #2:

Remarks to the Author:

In the article "A rigidity phase transition of Stomatin condensates governs a switch from transport to mechanotransduction", Sanfeliu-Cerdan and colleagues seek to understand how a shift from mobile and dynamic condensates to immobile and static condensates regulates force sensing and transduction in neurites. The authors use *C. elegans* and in vitro experiments to test the effect of a phase transition on mechanotransduction. The premise of this study is incredibly exciting and will be important for understanding how the biophysical properties of condensates regulate function outputs, including force sensing and transduction. The authors correlated in vitro and in vivo to reach their conclusion. However, there are several significant flaws with the manuscript in its current form which prevent this reviewer from supporting its publication at this time.

Major Concerns:

- 1) In vivo mature vs. naïve condensates; is the age of the condensate known or do the authors categorize them based on their behavior? Are naïve condensates static and positionally similar to mature condensates? Or are they static near the soma? One might imagine that the difference in recovery is not because naïve condensates are more fluid-like; rather it may be due to availability of unbleached molecules in the surrounding cytoplasm (i.e. a large pool in the body vs. a small pool further from the cell body). Because of this positional information relative to cell body is extremely important and comes into play in the experiments shown in Figure 4. Is FRET analyzed in naïve or mature condensates? Is there a FRET difference between the two types? Is one condensate type more likely to sense tension than the other? Most importantly, what are the local functional consequences of this tension sensing?
- 2) In Figure 1, What evidence is there that TEV-Cleaved MEC-2 is in fact cleaved? Can western blots be used to confirm cleavage and the percentage of MEC-2 protein that is cleaved? Does the cleaved MEC-2 protein maintain its structural domains or are they affected by the loss of the C-terminal? A western blot showing full cleavage of MEC-2 is necessary for assessing the degree of cleavage. Without knowing this, the conclusions from these experiments aren't solid. Quickly recovering C-terminal cleaved MEC-2 may be diffuse rather than in condensates, so comparing this protein to condensed protein isn't a proper comparison, it is simply comparing the dynamics of non-condensed MEC-2 with condensed MEC-2. In addition, why is a MEC-2 lacking the C-terminal not expressed rather than the TEV-cleavable protein? This seems like a more direct way of testing this hypothesis. If there is protein expression issues or viability issues caused by the expression of this MEC-2 variant, they should be stated in the text to show the logic for using TEV-MEC-2
- 3) The statistical analysis of comparing the means of quantified data is lacking. Instead of comparing means, the entire distribution of data points should be compared against distributions of data points within the groups of experiments. To accomplish this, a Kolmogorov-Smirnov test will suffice. Additionally, the colocalization analysis can be improved. The authors need to randomize their condensates within the regions of interest and test for colocalization to show that the colocalization is not a random occurrence. (See PMID 31268421 for more information regarding data analysis and colocalization).
- 4) The proline-rich motif experiments are confusing and need to be better described. It's unclear where the RtoH mutation is with respect to the PRM and what the PRM has to do with this analysis if the RtoH mutation is not in the 6th position (See PMID 18768933). The authors need to be more specific about this motif in the manuscript. The authors should also perform experiments with the negative control of no PRM needs to be performed with and without RtoH mutant if the mutation is outside of the PRM.
- 5) With regard to in vivo sensory experiments: How does the UNC-89 knockdown affect muscles given

Titin's role in muscle contraction? Couldn't it be that knocking Titin down prevents response simply because the mechanism of muscle contraction fails preventing movement? The authors need to show data that muscles are not affected by their knockdown. Also, does the calcium reduction scale with reduction in behavioral response (extended data 6e)? If it does, could this simply be the result of less muscle moved caused by the defect in Titin? Importantly, the authors description of the method "Gentle Body Touch Assays" needs to be clearer: what is the functional output? If movement of the *C. elegans* is the output, muscles certainly will play a role. If muscle physiology is altered by the UNC-89 knockdown, how will the authors parse muscle vs. TRNs?

6) In Figure 3G, the authors show that the RtoH mutation abrogates binding of the PRM by SH3. However, if this mutation abrogates binding, then the RtoH condensates in extended data 7a shouldn't dissolve at a 1:1 ratio. This strongly suggests that dissolution of the condensates is not due to binding. Rather, it is due something else. This contradiction will need to be solved.

7) What is the difference in tension sensing between naive and mature condensates (Figure 4D)? This should be stated because it will directly link physical properties with tension sensing. This is also key to the authors' claims that a phase transition alters mechanosensing.

8) Can MEC-2 transmit force or does this indicate that it can sense force? These are two distinct phenomena. The authors claim that it can transduce force when condensed, but they've only established force sensing.

Minor Comments:

1) In line 6 maturation... "transition" would be better here given the context of the work. Maturation suggests a one-way transition while it is plausible that some liquid to solid transitions are indeed reversible.

2) In line 15 liquid-like condensates... This isn't the case, it depends on viscoelastic properties of the condensate. Complex fluids can sustain mechanical forces as they approach a gel-like state.

3) In line 21 membrane organization... What is meant by membrane organization? lipids, proteins? be specific here.

4) In line 29 Stomatinn should be Stomatin.

5) In figure 1b, For all observed fusion, fission, and deformation events: Does fusion lead to a corresponding increase in condensate fluorescence? Does fission result in a corresponding loss of fluorescence? And is total condensate fluorescence effectively static in deformation events? These statistics and the number of corresponding events should be included.

6) In lines 48-49 Thus we reasoned... The structures do appear fluid-like but it may not necessarily be phase separated. Perhaps the authors at this point should state that this was their hypothesis based on these observations rather than reasoning that this is what the condensates are. The authors go on to support this hypothesis in the manuscript.

7) In lines 60-61 We hypothesize that the C-terminus... Why was this your hypothesis? What about the C-term leads you to this?

8) In line 87 and throughout ID protein should be IDP.

9) In lines 124-125 We reasoned that... The logic behind this statement should be stated to make it more clear for the reader.

10) In line 130 and throughout, PRiM should be PRM

11) In line 137, indicating that... the resolution of light microscopy isn't high enough to make this statement. Need proximity labeling or stained EM to confirm.

12) In line 148 LLPS behaviour... Do its viscoelastic properties change relative to WT? These experiments should be performed.

13) In line 158 sticky region proximal... What does this mean? Responding to Major comment 4 will be helpful here.

14) In line 159 defective touch response... Is this because the condensate is stickier, because the PRM

is inaccessible by SH3 domains, or something else?

15) In lines 161-162 accelerate biochemical reactions... Actin filament nucleation by BMCs is not because of increase concentration See PMID 27056844 and PMID 30846599. Rather the biochemical environment inside condensates is different than outside condensates and can promote biochemical reactions See PMID 29576456 and PMID 34035521.

16) In lines 210-211 through multivalent... This is a monovalent interaction, unless there are multiple PRMs in MEC-2 or SH3 domains on UNC-89 that can concomitantly interact.

17) Lines 221-223 and Extended data 6i... Are there other predicted binding partners that MEC-2 might interact with that could lead to this result?

18) Figure 3i, These structures need to be assessed for dynamics to see if the fibrils behave like solids (See PMID 26412307).

Reviewer #3:

Remarks to the Author:

Sanfeliu-Cerdán et al. have uncovered two novel results in their study of the MEC-2/stomatin protein. This protein had been shown to be required for gentle touch sensitivity in *C. elegans*. In this study the authors first demonstrate that the protein, especially that near the cell body appears to exist in a phase separated state. This work suggests that that state may be important for the distribution of the protein from the cell body to positions in the neurite. Their second important finding is that UNC-89/titin can interact with MEC-2 and is needed for touch sensitivity. Both results are important and should be of general interest. I would, however, like to see the manuscript modified, as suggested below.

Minor/Typos:

Abstract (line 4): "MEC-2 Stomatin" should be written "MEC-2/Stomatin"

line 29: Change "stomatinn" to "stomatin"

line 41 and throughout: The plural of "punctum" is "puncta" not "punctae"

line 258: Change "mecahnoreceptors" to "mechanoreceptors"

General comments:

line 20: "Stomatin family": the authors should describe the defining molecular characteristics of this family, for example, they speak of the "PHB domain" (line 63) without ever explaining what that is. This family is very large and, thus, their results might have far-reaching implications. A greater explanation of what this family is would be appreciated. Also, because the family is so large (with both eukaryote and prokaryotes having similar proteins), a better definition of what they are call stomatin is important. For example, do the authors only mean proteins that have the equivalent of MEC-2 Proline134, which has been shown to be very important for structure in stomatin, podocin, and MEC-2. They may also want to speak about other stomatin-like proteins in *C. elegans* and why they consider MEC-2 an ortholog.

lines 22-24 (MEC-2 interaction with MEC-4). The reference cited (#14) is incorrect (also on line 128 and possibly elsewhere). I suspect that they meant ref. 20. The authors, however, also consider the meaning of this interaction given the apparent lack of binding as observed with single molecules expressed in frog oocytes (Y. Chen et al. Proc. Natl. Acad. Sci. USA 112:11690-11695, 2015).

lines 24-26 ("Like in other Stomatin proteins..."): Extended Data Figure 1 does not show any of the data indicated by this sentence.

line 39: (Two distinct populations and Fig. 1a): The kymographs show either all "mature" and static or all "naïve" and moving and the diagram separates these two areas along the neurite. Given that the animals are normally touch sensitive over the entire length of the neurite, it is surprising that the mature puncta do not appear to be seen in the proximal neurite. If mature puncta coexist, they might be color coded in the image to distinguish them. Furthermore, the authors say that the naïve puncta are larger. A quantification of this difference would be a useful addition. Finally, the impression is that movement of these naïve particles occurs only in the proximal part of the cells. Does this change during development, i.e. are young larvae different from older animals or do these appear always in the same percentage of the neurite? Does this difference continue into the adult? The authors might comment on what this means for the development of the cell. What do aggregates look like in the cell body?

lines 51-55 ("Because the naïve pool. . .hypodermal cells...Fig. 1c"): These sentences are confusing. First, Fig. 1C appears to be data from the TRNs and not from hypodermal cells. Second, the authors should explain why hypodermal expression, i.e, without many of the other TRN-specific genes, e.g., *mec-4*, is a good model.

line 67: Here and elsewhere the authors use the word "indicate" when they might better say "suggest." An alternative hypothesis is that without the C-terminus, the protein does not fold correctly and misfolding is what causes the defect in condensation. Other examples are in lines 137, 142, and 209.

lines 75-77 (Fig. 1f): I am not sure what the data in this panel shows. The overexpression of TEV from the *mec-17* promoter appears to cause some touch insensitivity and the modified MEC-2 construct has even more. It would seem that the combination of the two could certainly give an even poorer touch response. This experiment would have been better with initial constructs that don't interfere with touch sensitivity.

line 82: While not disputing that AlphaFold 2 does not pick up structured domains at the N- and C-termini, I am not sure that the AlphaFold 2 data adds much because the program has problems with membrane proteins. The predictions of disorder seem a much better tool.

line 85 (Fig. 2b): I believe the readers would like more explanation of this panel. For example, presumably the boxed amino acids are those most common for disordered domains, but this information is not given. Also, what is the beige area in Fig. 2biii?

lines 214-215 ("Strikingly, we observed that both proteins sorted into the same punctae and colocalized along the neurite (Fig. 3e).") The image that is presented does not show what they say it does. The UNC-89 SH3 domain is expressed in most, if not all, of the neurite. I do not think that the data shown is compelling.

line 293: "Stomatin forms higher order oligomers" See also Huber et al. Proc. Natl. Acad. Sci. USA 103:17079-17086, 2006 for podocin and MEC-2.

Fig. 4 Should the top panel of 4b say donor? Is the TSMOD data the same as that shown in Fig. 1f? If so, this should be mentioned. What does the color code in 4d mean?

Discussion: I believe that the authors should enlarge their discussion. First, they really do not mention that MEC-2 and other proteins are membrane proteins and how phase separation acts with this localization. Second, they don't really expand their results to other members of this protein family. Such a discussion might be important given that the orientation of proteins within the member is different depending on whether they have the proline indicated above. For example, do proteins without this proline lack the unstructured domain?

Methods should be written concisely, but should contain all elements necessary to allow interpretation and replication of the results. As a guideline, Methods sections typically do not exceed 3,000 words. The Methods should be divided into subsections listing reagents and techniques. When citing previous methods, accurate references should be provided and any alterations should be noted. Information must be provided about: antibody dilutions, company names, catalogue numbers and clone numbers for monoclonal antibodies; sequences of RNAi and cDNA probes/primers or company names and catalogue numbers if reagents are commercial; cell line names, sources and information on cell line identity and authentication. Animal studies and experiments involving human subjects must be reported in detail, identifying the committees approving the protocols. For studies involving human subjects/samples, a statement must be included confirming that informed consent was obtained. Statistical analyses and information on the reproducibility of experimental results should be provided in a section titled "Statistics and Reproducibility".

All Nature Cell Biology manuscripts submitted on or after March 21 2016 must include a Data availability statement at the end of the Methods section. For Springer Nature policies on data availability see <http://www.nature.com/authors/policies/availability.html>; for more information on this particular policy see <http://www.nature.com/authors/policies/data/data-availability-statements-data-citations.pdf>. The Data availability statement should include:

- Accession codes for primary datasets (generated during the study under consideration and designated as "primary accessions") and secondary datasets (published datasets reanalysed during the study under consideration, designated as "referenced accessions"). For primary accessions data should be made public to coincide with publication of the manuscript. A list of data types for which submission to community-endorsed public repositories is mandated (including sequence, structure, microarray, deep sequencing data) can be found here <http://www.nature.com/authors/policies/availability.html#data>.
- Unique identifiers (accession codes, DOIs or other unique persistent identifier) and hyperlinks for datasets deposited in an approved repository, but for which data deposition is not mandated (see here for details <http://www.nature.com/sdata/data-policies/repositories>).
- At a minimum, please include a statement confirming that all relevant data are available from the authors, and/or are included with the manuscript (e.g. as source data or supplementary information), listing which data are included (e.g. by figure panels and data types) and mentioning any restrictions on availability.
- If a dataset has a Digital Object Identifier (DOI) as its unique identifier, we strongly encourage including this in the Reference list and citing the dataset in the Methods.

We recommend that you upload the step-by-step protocols used in this manuscript to the Protocol Exchange. More details can found at www.nature.com/protocolexchange/about.

All imaging data should be accompanied by scale bars, which should be defined in the legend. Cropped images of gels/blots are acceptable, but need to be accompanied by size markers, and to retain visible background signal within the linear range (i.e. should not be saturated). The boundaries of panels with low background have to be demarked with black lines. Splicing of panels should only be considered if unavoidable, and must be clearly marked on the figure, and noted in the legend with a statement on whether the samples were obtained and processed simultaneously. Quantitative comparisons between samples on different gels/blots are discouraged; if this is unavoidable, it should only be performed for samples derived from the same experiment with gels/blots were processed in parallel, which needs to be stated in the legend.

Figures should be provided at approximately the size that they are to be printed at (single column is 86 mm, double column is 170 mm) and should not exceed an A4 page (8.5 x 11"). Reduction to the scale that will be used on the page is not necessary, but multi-panel figures should be sized so that the whole figure can be reduced by the same amount at the smallest size at which essential details in each panel are visible. In the interest of our colour-blind readers we ask that you avoid using red and

green for contrast in figures. Replacing red with magenta and green with turquoise are two possible colour-safe alternatives. Lines with widths of less than 1 point should be avoided. Sans serif typefaces, such as Helvetica (preferred) or Arial should be used. All text that forms part of a figure should be rewritable and removable.

- For line art, graphs, charts and schematics we prefer Adobe Illustrator (.AI), Encapsulated PostScript (.EPS) or Portable Document Format (.PDF). Files should be saved or exported as such directly from the application in which they were made, to allow us to restyle them according to our journal house style.
- We accept PowerPoint (.PPT) files if they are fully editable. However, please refrain from adding PowerPoint graphical effects to objects, as this results in them outputting poor quality raster art. Text used for PowerPoint figures should be Helvetica (preferred) or Arial.
- We do not recommend using Adobe Photoshop for designing figures, but we can accept Photoshop generated (.PSD or .TIFF) files only if each element included in the figure (text, labels, pictures, graphs, arrows and scale bars) are on separate layers. All text should be editable in 'type layers' and line-art such as graphs and other simple schematics should be preserved and embedded within 'vector smart objects' - not flattened raster/bitmap graphics.
- Some programs can generate Postscript by 'printing to file' (found in the Print dialogue). If using an application not listed above, save the file in PostScript format or email our Art Editor, Allen Beattie for advice (a.beattie@nature.com).

SUPPLEMENTARY INFORMATION – Supplementary information is material directly relevant to the conclusion of a paper, but which cannot be included in the printed version in order to keep the

manuscript concise and accessible to the general reader. Supplementary information is an integral part of a Nature Cell Biology publication and should be prepared and presented with as much care as the main display item, but it must not include non-essential data or text, which may be removed at the editor's discretion. All supplementary material is fully peer-reviewed and published online as part of the HTML version of the manuscript. Supplementary Figures and Supplementary Notes are appended at the end of the main PDF of the published manuscript.

The total number of Supplementary Figures (not including the "unprocessed scans" Supplementary Figure) should not exceed the number of main display items (figures and/or tables (see our Guide to Authors and March 2012 editorial <http://www.nature.com/ncb/authors/submit/index.html#suppinfo>; <http://www.nature.com/ncb/journal/v14/n3/index.html#ed>). No restrictions apply to Supplementary Tables or Videos, but we advise authors to be selective in including supplemental data.

GUIDELINES FOR EXPERIMENTAL AND STATISTICAL REPORTING

REPORTING REQUIREMENTS – To improve the quality of methods and statistics reporting in our papers we have recently revised the reporting checklist we introduced in 2013. We are now asking all life sciences authors to complete two items: an Editorial Policy Checklist (found here <https://www.nature.com/authors/policies/Policy.pdf>) that verifies compliance with all required editorial policies and a reporting summary (found here <https://www.nature.com/authors/policies/ReportingSummary.pdf>) that collects information on experimental design and reagents. These documents are available to referees to aid the evaluation of the manuscript. Please note that these forms are dynamic 'smart pdfs' and must therefore be downloaded and completed in Adobe Reader. We will then flatten them for ease of use by the reviewers. If you would like to reference the guidance text as you complete the template, please access these flattened versions at <http://www.nature.com/authors/policies/availability.html>.

STATISTICS – Wherever statistics have been derived the legend needs to provide the n number (i.e.

the sample size used to derive statistics) as a precise value (not a range), and define what this value represents. Error bars need to be defined in the legends (e.g. SD, SEM) together with a measure of centre (e.g. mean, median). Box plots need to be defined in terms of minima, maxima, centre, and percentiles. Ranges are more appropriate than standard errors for small data sets. Wherever statistical significance has been derived, precise p values need to be provided and the statistical test used needs to be stated in the legend. Statistics such as error bars must not be derived from $n < 3$. For sample sizes of $n < 5$ please plot the individual data points rather than providing bar graphs. Deriving statistics from technical replicate samples, rather than biological replicates is strongly discouraged. Wherever statistical significance has been derived, precise p values need to be provided and the statistical test stated in the legend.

Author Rebuttal to Initial comments
--

Michael Krieg, Group leader Neurophotonics & Mechanical Systems Biology

ICFO - The Institute of Photonic Sciences

08860 Castelldefels,

Spain

Rebuttal

Review comments in black text; our response in blue

We thank the three anonymous reviewers as well as the editor for their appreciation of our work that is aimed to provide a mechanical mechanism for touch sensation, based on the rheological maturation of a phase-separated condensate involving MEC-2 stomatin and UNC-89 Titin. In addition, we established the binding interface with UNC-89 SH3 that promotes stiffening.

The reviewers provided detailed suggestions for improvement and requested additional data and analyses to clarify several aspects of the study, especially regarding the in vivo mechanism of stiffening. To address the concerns of the referees, we have now provided new data on multiple levels as detailed in the responses to the detailed referee comments. In short, we

1. measured the appearance of mobile and immobile MEC-2 condensates during development of the animals from the first larval stage to the young adult
2. provided a detailed characterization of the MEC-2 condensate size and mobility
3. have performed FRAP experiments on various mutant conditions of MEC-2 and UNC-89 to follow how the PRM influences recovery dynamics in vivo
4. have characterized the binding between both proteins by NMR, have explored the effect of proline mutations on condensate dynamics and sorting experiments in vitro and touch in vivo
5. performed a complete set of FRET tension experiments on various MEC-2 und UNC-89 mutant backgrounds
6. improved statistical reporting and image analyses

7. acquired a complete characterization of the condensate rheology to extract the complex G modulus aimed at understanding how the frequency dependent material properties change with condensate age and are influenced by their composition. This new data add significant mechanistic power to the study and provides one explanation why faster stimuli evoke larger currents in vivo.
8. improved the discussion and relevant references as suggested throughout.

We do believe that the review process was highly constructive and that the suggested additional analyses and experiments added a significant amount of descriptive and mechanistic insight to the presented problem. We very much hope that the reviewers see the effort we put into answering the outstanding questions to eliminate any additional concerns, but are also open for further criticism to further improve the work if needed.

Reviewer 1

Although a large body of work suggests that biomolecular condensates ensuing from liquid-liquid phase separation mature into various material states, how this aging process is controlled and if the naive and mature phases can have differential functions is currently unknown. Using *Caenorhabditis elegans* as a model, the authors demonstrated that MEC-2 Stomatin undergoes a rigidity phase transition during maturation from fluid to viscoelastic, glass-like condensates that facilitate either transport or mechanotransduction.

I agree that that the authors' data demonstrate a novel function for rigidity maturation during mechanotransduction. The authors used various in vitro and in vivo approaches and provided numerous data to support their claims, Therefore, I would like to support the eventual acceptance of this manuscript to NCB.

We thank the reviewer for their positive comments.

Nevertheless, I still think that the two key data are missing. I got an overall impression that the role for the UNC-89 SH3 domain in promoting MEC-2 maturation (rigidity phase transition) was not sufficiently demonstrated in vivo.

Majors:

1. Characterization of MEC-2(R385H) point mutant in vivo: In Fig. 1, the authors showed a rigidity phase transition of WT MEC-2 and its difference from the C-terminally truncated MEC-2 mutant.

However, the authors did NOT show a sol-gel transition of MEC-2(R385H) mutant. This is critical, because this mutant is the key mutant the authors used for characterizing the interaction with UNC-89 SH3 domain in Fig. 3.

The reviewer raised an important point and we are now including the data for the sol to gel transition *in vivo*. In particular we addressed the problem from multiple angles:

- Recovery dynamics in naive puncta: We performed FRAP on the ectopically expressed MEC-2(R385H) mutation in the hypodermis, where *unc-89* is not expressed, and observe that the puncta initially recovered faster, but to the same extent (Extended Data Fig. 4i). This suggests that the MEC-2(R385H) mutation has a strong effect in the recovery dynamics of the naive MEC-2 condensate.
 - We have now also presented data which suggests that MEC-2(R385H) has lower recovery dynamics in mature condensates within TRNs as compared to the wild type MEC-2 protein *in vivo*. We have included this data, together with the other characterization of the MEC-2(R385H) mutant as a new Fig. 2 and Extended Data Fig. 4h.
 - We had previously performed FRET in the R385H mutation, which was included in old Fig. 4 of the first submission, and found that the mutant MEC-2 cannot sustain mechanical stress during body wall touch as efficiently as the wild type condensates. To characterize this in greater detail we have now repeated these measurements with the MEC-2 tension sensor expressed in the hypodermis, which normally does not express *unc-89* (Extended Data Fig. 10). In addition, we performed FRET measurements under increasing pressures delivered to the body wall of microfluidically immobilized animals with a C-terminally truncated MEC-2 protein expressed in TRNs - a construct from which we initially showed that it does not form discrete condensates and remains in a fluid-like state (as determined by FRAP, see Fig. 1). In none of these cases did we observe significant force propagation.
2. The authors stated in Abstract: “This switch is promoted by the SH3 domain of UNC 89/Titin/Obscurin through a direct interaction with MEC-2 and suggests a physiological role for a percolation transition in force transmission during body wall touch.” I think that the key supporting data for this claim is missing. If this claim is current, we would expect a reduced sol-gel transition of WT MEC-2 (shown in Fig. 1) in the absence of UNC-89. That is, I think that the authors should examine the rigidity phase transition of WT MEC-2 in the *unc-89* mutant background.

Similar to the points above, we present new data to underline the initial observation that UNC-89 promotes the rigidity transition.

- We performed FRET measurements of wild type MEC-2::TSMOD in the UNC-89 mutants under mechanical stimulation and found that the absence of UNC-89 leads to a loss of force transmission, as seen by an unchanged FRET value with increasing pressure delivered to the body wall. We have included this as a new Fig. 6. However, we did not detect noticeable changes in FRAP dynamics of MEC-2 puncta in TRNs in absence of UNC-89 (Extended Data Fig. 6d). This may be due to the fact that MEC-2 condensates in this mutant background are usually smaller and have a smaller IPI (ED Figure 6). Also, the possibility exists that other SH3 domains compensate and that *unc-89* determines the elastic properties of these condensates, which are inaccessible by FRAP measurements. To address this limitation, we performed the FRAP measurements of wild type MEC-2 condensates in the hypodermis, which does not express UNC-89 and does not undergo a rigidity transition, as evidenced by FRET imaging of a hypodermally expressed MEC-2::TSMOD (see point above, Extended Data Fig. 10d). In this condition, we found that MEC-2 recovery half-time is reduced and mobile fraction is increased when UNC-89 is not expressed (Fig. 1e).

Minors: 1) line 250: weakens a heterotypic interaction between MEC-2 molecules that kinetically stabilized the MEC-2... Here, “heterotypic” interaction should read “homotypic” interaction?

We thank the referee for pointing this out. We have edited this statement in the text during the revision.

2) line 251-255: these sentences seem to largely consist of the authors’ speculation. It may be better to discuss these in the discussion section.

We have now moved these statements to the discussion as suggested by the referee.

Reviewer 2

In the article “A rigidity phase transition of Stomatin condensates governs a switch from transport to mechanotransduction”, Sanfeliu-Cerdan and colleagues seek to understand how a shift from mobile and dynamic condensates to immobile and static condensates regulates force sensing and transduction in neurites. The authors use *C. elegans* and in vitro experiments to test the effect of a phase transition on mechanotransduction. The premise of this study is incredibly exciting and will be important for

understanding how the biophysical properties of condensates regulate function outputs, including force sensing and transduction. The authors correlated in vitro and in vivo to reach their conclusion. However, there are several significant flaws with the manuscript in its current form which prevent this reviewer from supporting its publication at this time.

Major Concerns:

1. In vivo mature vs. naive condensates; is the age of the condensate known or do the authors categorize them based on their behavior?

We categorized them solely based on their behavior and their morphology within the neurites. We have added the detailed characterization of their appearance as Extended Data Fig. 1. For example we have found that the MEC-2 condensates of the mobile, naive pool are bigger in size and more spherical than the immobile, mature pool. One interpretation that has also been propagated in the recent literature is that more spherical condensates display properties of a liquid like material. Specifically, we write on lines 56-57: “The clusters of the naive pool were on average bigger and more spherical than those of the mature pool (Extended Data Fig. 1a).”

Are naive condensates static and positionally similar to mature condensates? Or are they static near the soma?

We apologize for the confusion and noticed that the wording used in the first version could easily be misinterpreted as if the two populations are mutually exclusive and non overlapping. We have now clarified this in the new version and replaced the kymograph in Fig. 1 to clearly demonstrate this. As described in the main text, we categorized the condensates based on their mobility behavior and, based on this definition, naive condensates are more mobile and occupy overlapping territories with the mature condensates. This means that even proximal to the cell body we will find static puncta, and occasionally, far away from the cell body we will find mobile puncta. We also found more mobile puncta in longer axons, indicating that directed transport is more prevalent, or at least, easier to detect in long neurites of adults animals vs shorter neurites in larval stages. We have added this as Extended Data Fig. 1.

One might imagine that the difference in recovery is not because naive condensates are more fluid-like; rather it may be due to availability of unbleached molecules in the surrounding cytoplasm

(i.e. a large pool in the body vs. a small pool further from the cell body).

We thank the referee for bringing up this important point. We have now performed FRAP on static puncta close to the cell body (within 50-100 μm) and far away from it ($>150 \mu\text{m}$) and could not find differences in their recovery dynamics. We interpret this finding that the recovery is mostly due to internal exchange of molecules within a MEC-2 puncta, e.g. between the bleached and unbleached spot, rather than due to an exchange between the puncta and the cytoplasm. We now included this data as Extended Data Figure 2. Specifically we write in lines 80-85: "Importantly, the delayed recovery of the static condensates did not depend on the position on the neurite with respect to the cell body when we photobleached individual MEC-2 condensate proximal ($<100 \mu\text{m}$) and distal ($>200 \mu\text{m}$) to the cell body (Extended Data Fig. 2). This suggests that mature puncta do not readily recover through transport from the dilute phase, illustrating that they are kinetically trapped condensates with viscoelastic properties distinct to those of the mobile MEC-2 puncta."

Because of this positional information relative to cell body is extremely important and comes into play in the experiments shown in Figure 4. Is FRET analyzed in naive or mature condensates? Is there a FRET difference between the two types?

Due to the mobility of the puncta in the naive form, which is incompatible with the slow FRET image acquisition, we were not able to perform FRET measurements on the mobile phase of the MEC-2 condensates. However, we have now added data for the FRET measurements in the hypodermis, of MEC-2::TSMMod in the UNC-89 mutant background and a C-terminally truncated MEC-2, all of which we initially hypothesized to have a delayed rigidity transition. Indeed, we found that MEC-2::TSMMod does not report changes under external touch in the microfluidic devices when expressed in the hypodermis (Extended Data Fig. 10), is C-terminally truncated or when UNC-89 is not expressed in TRNs. Since per definition soft viscoelastic or purely viscous structures cannot sustain mechanical stresses longer than their characteristic relaxation time scale, we reasoned that only wild type MEC-2 expressed in TRNs is able to transfer force. We have also in depth characterized the rheological properties of pure MEC-2 condensates and their mixtures with UNC-89 (Fig. 5).

Is one condensate type more likely to sense tension than the other? Most importantly, what are the local functional consequences of this tension sensing? We do not have a direct answer to this question but can speculate the following. The observed ballistic motion of the mobile MEC-2 con-

densates is typical for motorprotein driven transport, in particular active transport on microtubules, and thus consistent for condensate resident in the axonal cytoplasm. We thus speculate that only the immobile but not the mobile condensates associate with the membrane and hence the functional mechano-electrical transduction channel MEC-4 (Extended Data Fig. 4f). Consequently, we reasoned that the immature, mobile MEC-2 fraction cannot transduce tension into neuronal signaling. Because we have no method established that measures intermolecular force transfer between MEC-2 and MEC-4, we are unable to visualize this process, but we strongly favor the idea that MEC-2 stabilizes the open conformation of MEC-4 when it is set under tension. Moreover, we do not expect local differences in tension sensing, as the mobile pool overlaps with the immobile pool in the proximal regions of the neurite. This interpretation is consistent with prior work that established that local force applications to the skin of the animal can elicit neurophysiological responses close to the cell body and in distal parts[1].

2. In Figure 1, What evidence is there that TEV-Cleaved MEC-2 is in fact cleaved? Can western blots be used to confirm cleavage and the percentage of MEC-2 protein that is cleaved? Does the cleaved MEC-2 protein maintain its structural domains or are they affected by the loss of the C-terminal? A western blot showing full cleavage of MEC-2 is necessary for assessing the degree of cleavage. Without knowing this, the conclusions from these experiments aren't solid. Quickly recovering C-terminal cleaved MEC-2 may be diffuse rather than in condensates, so comparing this protein to condensed protein isn't a proper comparison, it is simply comparing the dynamics of non-condensed MEC-2 with condensed MEC-2. In addition, why is a MEC-2 lacking the C-terminal not expressed rather than the TEV-cleavable protein? This seems like a more direct way of testing this hypothesis. If there is protein expression issues or viability issues caused by the expression of this MEC-2 variant, they should be stated in the text to show the logic for using TEV-MEC-2.

We thank the referee for this critical comment. We have now expressed a constitutive C-terminally truncated version of MEC-2 and observed the same effects similar to the TEV-cleaved protein. In short, the truncated version does not form condensates or stable puncta along the neurites and is not able to transduce mechanical touch. In addition, it recovers faster in the FRAP experiments. We have included this as a new Extended Data Figure 3. Specifically, we write in lines 107-108: "importantly, we made the same observations with a constitutive truncated MEC-2 construct lacking the C-terminal domain (Extended Data Fig. 3)."

3. The statistical analysis of comparing the means of quantified data is lacking. Instead of comparing means, the entire distribution of data points should be compared against distributions of data points within the groups of experiments. To accomplish this, a Kolmogorov-Smirnov test will suffice.

We have included the statistical comparison to guide the judgement of significant differences. In addition to the commonly used binary decision based on the p-value and level of significance alpha, we have included judgments based on effect sizes as outlined in ref. [2], where ever possible. Additionally, the colocalization analysis can be improved. The authors need to randomize their condensates within the regions of interest and test for colocalization to show that the colocalization is not a random occurrence. (See PMID 31268421 for more information regarding data analysis and colocalization).

We thank the referee for pointing this out. We have now collected more data points and performed a randomization test (See Methods and Extended Data Fig. 5k). The results can be seen in Fig. 3e and confirm our initial assessment.

4. The proline-rich motif experiments are confusing and need to be better described. It's unclear where the RtoH mutation is with respect to the PRM and what the PRM has to do with this analysis if the RtoH mutation is not in the 6th position (See PMID 18768933). The authors need to be more specific about this motif in the manuscript. The authors should also perform experiments with the negative control of no PRM needs to be performed with and without RtoH mutant if the mutation is outside of the PRM.

We have now repeatedly mentioned in the manuscript where the mutation is located and included schematics wherever relevant (Fig. 2f). As the referee noted, many proline rich motifs obtain specificity from a conserved arginine (or other charged residues) at the 7 or 8th position, with respect to the first proline in the motif PPxxPxx(x)R. To directly answer the questions raised, we expressed and purified the MEC-2 C-terminal domain without the proline rich motif (AAxxA) and measured the SH3 binding by NMR experiments. As expected, in the absence of the proline rich motif, SH3 binding is abolished both in the wild type or R385H MEC-2 isoforms. Additionally, droplet co-partitioning measured by fluorescence microscopy in vitro indicate that mutating the proline rich motif in MEC-2 has a strong impact on droplet dynamics. We repeated droplet co-partitioning and observed that the AAxxA mutants did not evolve into fibrillar-like species after 24h of incubation with UNC89, neither in the wild type nor in the R385H backgrounds. We have added these data to Fig. 4. We also showed that these prolines play an important role during touch

in vivo: We generated new CRISPR mutants that replace these prolines with alanine in the genome of *C. elegans* and demonstrated that these edits lead to a complete loss of touch sensation (see text lines 252-256 and 265-270 for details).

5. With regard to in vivo sensory experiments: How does the UNC-89 knockdown affect muscles given Titin's role in muscle contraction? Couldn't it be that knocking Titin down prevents response simply because the mechanism of muscle contraction fails preventing movement? The authors need to show data that muscles are not affected by their knockdown.

The reviewer brings up a relevant concern, as titin is mostly expressed in muscles. However, we have performed the sensory stimulation experiments in *unc-89* mutants that are locally restricted to only the touch receptor neurons, meaning that all other tissues are UNC-89 intact. We achieved this using an intersectional strategy in which we flanked the genomic *unc-89* locus with loxP sites and expressed CRE enzyme cell specifically in TRNs using a promoter with NO overlap with other cells of the nervous system and muscles. We confirmed the CRE specificity with a CRE-activity reporter and found 100% recombination efficiency in this study and in our previous study (see also ref. [3]). We then recorded short videos and confirmed that these cell/neuron specific mutants do not have a locomotion defect. We included these data as an additional Extended Data Fig. 4i.

Also, does the calcium reduction scale with reduction in behavioral response (extended data 6e)? If it does, could this simply be the result of less muscle moved caused by the defect in Titin? Importantly, the authors description of the method "Gentle Body Touch Assays" needs to be clearer: what is the functional output? If movement of the *C. elegans* is the output, muscles certainly will play a role. If muscle physiology is altered by the UNC-89 knockdown, how will the authors parse muscle vs. TRNs? See comment above about cell-specific, intersectional CRE strategy.

6. In Figure 3G, the authors show that the RtoH mutation abrogates binding of the PRM by SH3. However, if this mutation abrogates binding, then the RtoH condensates in extended data 7a shouldn't dissolve at a 1:1 ratio. This strongly suggests that dissolution of the condensates is not due to binding. Rather, it is due something else. This contradiction will need to be solved. We thank the reviewer for this comment and have repeated these experiments at varying ratios of MEC-2 and UNC-89 ranging from 1:0 to 1:1 and found that the droplets do not dissolve even up to 1:1 ratio for neither the wild type nor the R385H mutant. We also used the PPxxP to AAxxA MEC-2 mutants that completely abrogate binding to UNC-89 and obtained the same result. Altogether,

these results indicate that binding of UNC-89 to MEC-2 does not promote droplet dissolution. A possible explanation for the former results is that the samples phase separate at 35 °C and the images were taken at 37 °C, therefore slight changes in protein or salt concentrations given by the intrinsic error in protein concentration determination or pipetting could give rise to a change in the temperature at which the droplets form. To diminish these errors, we further concentrated our stock solutions close to their solubility limit to prepare the new samples. We have changed the text accordingly and updated the Extended Data Fig. 7 with the new data. Altogether, the data suggests that the sticky region that potentially drives the LLPS in MEC-2 is located next to the proline-rich motif. At the same time, the R385H mutation influences the low-affinity binding to the SH3 domain of UNC-89, and interferes with the delicate balance between intra- and inter-molecular interactions of MEC-2 that lead to the observed rigidification of condensates both in vitro and in vivo.

7. What is the difference in tension sensing between naive and mature condensates (Figure 4D)? This should be stated because it will directly link physical properties with tension sensing. This is also key to the authors' claims that a phase transition alters mechanosensing. As stated above, we have performed the FRET measurements in condensates in the hypodermis, which are significantly more liquid-like than the static condensates in TRNs, based on their FRAP recovery dynamics. We found that the pressure induced FRET values of the hypodermally expressed MEC-2::TSMOD are significantly higher than in control tension sensors expressed in TRNs. Our analysis also suggests that FRET values of the MEC-2::TSMOD expressed in the hypodermis are insensitive to body wall touch, indicating that MEC-2 expressed in ectopic tissues cannot bear mechanical stress. In addition, we have performed FRET on the TEV-cleaved MEC-2::TSMOD, which does not contain the motif needed for rigidity transition and found significantly higher FRET values. We have now included these results in Fig. 6 and Extended Data Fig. 10.
8. Can MEC-2 transmit force or does this indicate that it can sense force? These are two distinct phenomena. The authors claim that it can transduce force when condensed, but they've only established force sensing. Critically, our FRET measurement showed that a mechanical insult to the body of the animals propagates into the neuron where MEC-2 is set under tension. The FRET cassette is thus a calibrated indicator that MEC-2 experiences a force. However, we cannot predict if MEC-2 transmits this force to the pore-forming subunit of the ion channel. To address the

question of the referee directly, we have now thoroughly determined the complex shear modulus of the MEC-2 condensates as a function of age, composition and strain rate in a dual trap optical tweezer assay. In this assay, the condensate is sandwiched between two beads of which one is driven at a given frequency while measuring the transmitted force with the second, passive bead. The measured quantities yield the frequency dependent complex shear modulus G . The complex shear modulus provides important information about the conservative (G') and dissipative (G'') properties of a material and characterize the timescales and magnitudes of how well this material can 'store' or 'dissipate' mechanical energy. From these experiments we observed that MEC-2/UNC-89 obtains a higher storage modulus and lower loss modulus after 24h after condensate formation. We conclude that MEC-2 becomes more rigid and obtains a higher viscosity (sustains force for longer time), which is accelerated in presence of functional UNC-89. This viscoelastic ageing, or maturation, is not visible when UNC-89 and the mutant MEC-2 are allowed to co-condensate. We have now included these results in Fig. 5 and Extended Data Fig. 9. In addition, we have provided a Supplementary Discussion with more details in the experimental procedures, assumptions and implications of these measurements.

In short, these results have an important implication about the function of mechanoreceptors in vivo: It has been known for 20 years that TRNs are more sensitive to high frequency vibrations than to low step indentations (see ref [4, 5, 6]), which implicates a strong frequency dependent response, similar to what was observed in mammalian Pacinian corpuscle [7]. Several hypotheses were put forward that proposed a mechanical filter, present in the hypodermis ([8, 9]), which give rise to the transmission of fast stimuli but dampens slow stimuli. Our results showing that slow frequencies are effectively dissipated and dampened is the first proof that mechanoreceptor complexes and their associated proteins show a frequency dependent loss and storage modulus with importance in mechanotransduction.

Minor Comments:

- In line 6 maturation. . . “transition” would be better here given the context of the work. Maturation suggests a one-way transition while it is plausible that some liquid to solid transitions are indeed reversible.

Done

- In line 15 liquid-like condensates. . . This isn't the case, it depends on viscoelastic properties of the condensate. Complex fluids can sustain mechanical forces as they approach a gel-like state.

Done

- In line 21 membrane organization. . . What is meant by membrane organization? lipids, proteins? be specific here.

Done

- In line 29 Stomatinn should be Stomatin.

Done

- In figure 1b, For all observed fusion, fission, and deformation events: Does fusion lead to a corresponding increase in condensate fluorescence? Does fission result in a corresponding loss of fluorescence? And is total condensate fluorescence effectively static in deformation events? These statistics and the number of corresponding events should be included. We have now included the quantification of the fluorescence intensity of representative fusion and fission events and added this analysis to Extended Data Fig. 1d, e, respectively.

- In lines 48-49 Thus we reasoned. . . The structures do appear fluid-like but it may not necessarily be phase separated. Perhaps the authors at this point should state that this was their hypothesis based on these observations rather than reasoning that this is what the condensates are. The authors go on to support this hypothesis in the manuscript.

We thank the referee for suggesting a better wording of our sentences. Specifically, we now write in lines 66-68: "Based on these observations we hypothesized that MEC-2 exists as phase separated biomolecular condensates *in vivo* with spatially distinct material properties."

- In lines 60-61 We hypothesize that the C-terminus. . . Why was this your hypothesis? What about the C-term leads you to this?

As outlined in the introduction and Supplementary Fig. 1, many stomatin proteins, including MEC-2 have unstructured terminal domain with low sequence complexity. We make this explicitly clear in this section and write: "Biomolecular condensation is often driven by phase transitions of intrinsically disordered regions with low sequence complexity. We hypothesized."

- In line 87 and throughout ID protein should be IDP.

We have now defined this abbreviation.

- In lines 124-125 We reasoned that... The logic behind this statement should be stated to make it more clear for the reader.

We have now explicitly explained this statement and write: "displays strain-rate dependent mechanical properties. We speculate that these properties are important for mechanotransduction and have pronounced consequences on the frequency selection during touch sensation *in vivo*."

- In line 130 and throughout, PRiM should be PRM

Done

- In line 137, indicating that... the resolution of light microscopy isn't high enough to make this statement. Need proximity labeling or stained EM to confirm.

We have rewritten the conclusion about this experiment, which was to test that the mutant MEC-2 still sorts into the same puncta together with MEC-4, as expected from the wild type protein which was shown to directly interact with the MEC-2 proteins. We write: "Because many *mec-2* point mutations disrupt MEC-2 trafficking and thus localization into discrete puncta along the neurite [10], we investigated if the MEC-2 R385H mutant could still colocalize with MEC-4, the pore-forming subunit of the MeT channel. Importantly, mutated MEC-2 (R385H) localized to TRN dendrites indistinguishably to wild type MEC-2 *in vivo* and colocalized with the MeT channel MEC-4 (Extended Data Fig. 4f), suggesting that it retained the capability to partition into MEC-4 positive puncta in mature condensates along the neurite."

- In line 148 LLPS behaviour... Do its viscoelastic properties change relative to WT? These experiments should be performed.

We have now performed the step relaxation measurements on the condensates composed of the mutant MEC-2(R385H) proteins and the active microrheology on condensates composed of mutant MEC-2 and UNC-89 mixtures. In contrast to droplet composed of mixtures with wild type MEC-2 and UNC-89, these mutant droplet did not mature over the course of 24 h. We present these results in the new Figure 5 and Extended Data Fig. 9, as also described above (lines 286-296 main text).

- In line 158 sticky region proximal... What does this mean? Responding to Major comment 4 will be helpful here.

We have extensively edited this text and repeated the phase separation experiments *in vitro* to clarify any confusion. We also removed any references with unclear terminology, such as 'sticky'.

In the discussion we write: "... binding of the UNC89 SH3 domain to the PRM would trigger the liquid-to-solid transition because this region of sequence, or residues in its vicinity, preserves the liquid character by interacting with a yet to be identified aggregation-prone motif in the C-terminal domain of MEC-2 by a mechanism known as heterotypic buffering[11]."

- In line 159 defective touch response. . . Is this because the condensate is stickier, because the PRM is inaccessible by SH3 domains, or something else?

In the flow of the arguments, we did not elaborate the mechanism for the defective touch at this position in the text yet and this statement is meant to summarize the main experimental findings of the preceding paragraphs. We addressed the possible mechanism how this point mutation gives rise to a defective touch response with involvement of the 'sticky' region in the sections that follow. Please also see our answer to the previous comment above.

- In lines 161-162 accelerate biochemical reactions. . . Actin filament nucleation by BMCs is not because of increase concentration See PMID 27056844 and PMID 30846599. Rather the biochemical environment inside condensates is different than outside condensates and can promote biochemical reactions See PMID 29576456 and PMID 34035521.

We thank the referee for pointing this out and suggesting these references to underline our statements. We have added them to the manuscript at the appropriate sections.

- In lines 210-211 through multivalent. . . This is a monovalent interaction, unless there are multiple PRMs in MEC-2 or SH3 domains on UNC-89 that can concomitantly interact.

We have deleted multivalent, as it is irrelevant for the statements that follow.

- Lines 221-223 and Extended data 6i. . . Are there other predicted binding partners that MEC-2 might interact with that could lead to this result?

We have now removed this panel from the Figure as it does not add more conclusive data to the manuscript. We have thoroughly interrogated the binding with more direct methods.

- Figure 3i, These structures need to be assessed for dynamics to see if the fibrils behave like solids (See PMID 26412307).

We have performed FRAP experiments on a MEC-2 wild type sample with UNC-89 (1:0.1 ratio) both at time 0 and 24 h of incubation. In order to properly compare the dynamics between the former (sample with droplets) and the latter (sample with aged droplets and fibrillar species) we

bleached, for the latter, the aged droplets. We found that after 24h the protein molecules diffuse slower in the droplets than at time 0h, indicating that they are more viscous on these fibril in vitro and observed that they recover slower than the naive mixtures in vitro (see Extended Data Fig. 8). We also performed optical tweezer step deformations directly on the fibers (see Figure below) and found that they display properties of a non-dissipative solid, with a strong resting force after stress relaxation compared to naive droplets.

Referee Fig. 1: MEC-2 fibers behave as solids. **a**, Image of a microsphere in contact with a fiber. **b**, Lissajou figures of the sinusoidal indentations showing a perfect solid response without noticeable hysteresis. **c**, Force response during repetitive step indentations.

Reviewer #3:

Sanfeliu-Cerdn et al. have uncovered two novel results in their study of the MEC-2/stomatin protein. This protein had been shown to be required for gentle touch sensitivity in *C. elegans*. In this study the authors first demonstrate that the protein, especially that near the cell body appears to exist in a phase separated state. This work suggests that that state may be important for the distribution of the protein from the cell body to positions in the neurite. Their second important finding is that UNC-89/titin can interact with MEC-2 and is needed for touch sensitivity. Both results are important and should be of general interest. I would, however, like to see the manuscript modified, as suggested below.

Minor/Typos:

- Abstract (line 4): “MEC-2 Stomatin” should be written “MEC-2/Stomatin”
We have changed this notation throughout the text.
- line 29: Change “stomatinn” to “stomatin”
Done
- line 41 and throughout: The plural of “punctum” is “puncta” not “puncta”
We thank the referee for pointing this out. We have changed this throughout the text.
- line 258: Change “mecahnoreceptors” to “mechanoreceptors” Thank you for pointing this out.
Done.

General comments:

1. line 20: “Stomatin family”: the authors should describe the defining molecular characteristics of this family, for example, they speak of the “PHB domain” (line 63) without ever explaining what that is. This family is very large and, thus, their results might have far-reaching implications. A greater explanation of what this family is would be appreciated. Also, because the family is so large (with both eukaryote and prokaryotes having similar proteins), a better definition of what they are call stomatin is important. For example, do the authors only mean proteins that have the equivalent of MEC-2 Proline134, which has been shown to be very important for structure in stomatin, podocin, and MEC-2.

We have now clarified what we mean by stomatin family and replaced the term ‘PHB domain’ with the ‘stomatin domain’. However, we would like to omit the specific reference to the conserved prolines, as we fear that this might cause confusion to the reader, as our focus in this article deals with another proline residue in a highly conserved SH3-binding motif of the unstructured tail. Specifically we write in line 21-32: ”Stomatins are highly conserved membrane-associated scaffolding proteins that are required for membrane mechanics[12] and modulate ion channel activity[13, 14, 15]. This family of proteins is characterized by a stereotypic stomatin domain found in all members and the ability to bind cholesterol through a conserved residue in the N-terminal domain[16]. The family diverged from a larger superfamily, the SPFH (Stomatin, Prohibitin, Flotillin, HflK/HflC domain), which is found in all phyla including Archaea as well as in mitochondria. In the mouse, STOML3

is important for membrane stiffening, mechanical sensitivity [12] and the sense of touch [17]. *C. elegans* contains eight different members of this protein family, all expressed in neurons, including UNC-24 and MEC-2, which are found in the touch receptor neurons (TRNs). The homolog MEC-2 ... ”.

They may also want to speak about other stomatin-like proteins in *C. elegans* and why they consider MEC-2 an ortholog.

We have now replaced the term ‘ortholog’ with the umbrella term ‘homolog’ to avoid confusion and potential concerns about its mode of divergence.

2. lines 22-24 (MEC-2 interaction with MEC-4). The reference cited (#14) is incorrect (also on line 128 and possibly elsewhere). I suspect that they meant ref. 20. The authors, however, also consider the meaning of this interaction given the apparent lack of binding as observed with single molecules expressed in frog oocytes (Y. Chen et al. Proc. Natl. Acad. Sci. USA 112:11690-11695, 2015).

We have added the following statement to the discussion, which reflects the referee’s suggestion to incorporate the mentioned important findings from oocytes, line 403: ”Our data is consistent with the previous observation that MEC-2 interacts transiently with MEC-4 in oocytes[18], where it may modulate the stiffness of cholesterol-enriched platforms surrounding the ion channel[19].”

3. lines 24-26 (“Like in other Stomatin proteins...”): Extended Data Figure 1 does not show any of the data indicated by this sentence.

We have modified our statements according to the suggestion of the referee. We write starting in line 33: “The N and the C-termini of the *C. elegans* Stomatins have regions of sequence predicted to be intrinsically disordered (Supplementary Fig. 1), with low sequence complexity, composed of repetitive sequences of proline, glycines and serines and little homology to other family members.”

4. line 39: (Two distinct populations and Fig. 1a): The kymographs show either all “mature” and static or all “naive” and moving and the diagram separates these two areas along the neurite. Given that the animals are normally touch sensitive over the entire length of the neurite, it is surprising that the mature puncta do not appear to be seen in the proximal neurite. If mature puncta coexist, they might be color coded in the image to distinguish them.

We thank the referee for pointing this out and apologize for the confusion, as we chose an imprecise wording of our observation (see also answer to Ref #1 above). Indeed, mature puncta are seen all

along the neurite and we have mentioned this explicitly in the main text. The mobile pool, however, can be predominantly seen in the vicinity of the cell body and its relative proportion increases with the developmental age of the animal, and thus correlates with the neurite length. We have now replaced the kymograph in Figure 1a with one that better visualizes the co-existence of the mature and naive condensate and added the quantification of the relative proportion of these pools. In general, and this is what we see also in young larvae, mature puncta form throughout the neurite and even very early in development.

Furthermore, the authors say that the naive puncta are larger. A quantification of this difference would be a useful addition.

We have now added the quantification of the projected area and the roundness of the two populations to the supplementary data in the manuscript. The comparison can be found in Extended Data Fig. 1a. Indeed, naive puncta are more spherical, congruent with the previous description of liquid like character of phase separated condensates. Mature puncta are more elongated, suggesting a deviation from a liquid like character.

Finally, the impression is that movement of these naive particles occurs only in the proximal part of the cells. Does this change during development, i.e. are young larvae different from older animals or do these appear always in the same percentage of the neurite? Does this difference continue into the adult? The authors might comment on what this means for the development of the cell.

We thank the referee for the interesting question. We have imaged the MEC-2 dynamics in early larvae and young adult animals and observed a qualitatively similar behavior. Even in early larvae, mature and mobile puncta coexist while most of the dynamics can be found close to the cell body. However, in older developmental stages, the relative fraction of the mobile pool in reference to the static pool increases which is in line with our hypothesis that the mobility is associated with transport into the neurites. We have added the quantification of the relative contribution of the mobile pool vs immobile pool to the Extended Data Fig. 1b.

What do aggregates look like in the cell body?

Aggregates in the cell body are round and larger in size. We have now added the quantification as Extended Data Fig. 1c. Specifically we write in line 61: “In all developmental stages, we also found MEC-2 clusters in the cell body, which were larger and more mobile than those found in neurites”.

5. lines 51-55 (“Because the naive pool. . .hypodermal cells. . .Fig. 1c”): These sentences are confus-

ing. First, Fig. 1C appears to be data from the TRNs and not from hypodermal cells. Second, the authors should explain why hypodermal expression, i.e., without many of the other TRN-specific genes, e.g., *mec-4*, is a good model.

We have now clarified in the text why we used hypodermal cells as a model to investigate the behavior of MEC-2 in a heterologously expressed cell. In addition to the absence of TRN, and neuron-specific genes that are necessary for MEC-2 function and dynamics, such as MEC-4 ion channels and the cytoskeleton (including MEC-12, UNC-89), the MEC-2 condensates in the hypodermal cells were primarily static and did not show a directed transport, which facilitate the 3 min long recording during the FRAP experiment.

6. line 67: Here and elsewhere the authors use the word “indicate” when they might better say “suggest.” An alternative hypothesis is that without the C-terminus, the protein does not fold correctly and misfolding is what causes the defect in condensation. Other examples are in lines 137, 142, and 209.

We thank the referee for pointing this out and have changed the wording as suggested.

7. lines 75-77 (Fig. 1f): I am not sure what the data in this panel shows. The overexpression of TEV from the *mec-17* promoter appears to cause some touch insensitivity and the modified MEC-2 construct has even more. It would seem that the combination of the two could certainly give an even poorer touch response. This experiment would have been better with initial constructs that don't interfere with touch sensitivity.

We agree with the referee and generated new transgenics in which we inserted the TEV cleavage site without the fluorescent protein. Importantly, in absence of the TEV protease, this transgenic behaves almost like wild type animals, however, becomes severely touch insensitive when the TEV protease is co-expressed. This loss in touch sensitivity coincides with a loss of punctate/condensate formation, indicating successful cleavage. We have included this new data in Fig. 1 and corroborated this analysis with a C-terminally truncated construct as suggested by Referee 2 (see Extended Data Figure 3).

8. line 82: While not disputing that AlphaFold 2 does not pick up structured domains at the N- and C-termini, I am not sure that the. AlphaFold 2 data adds much because the program has problems with membrane proteins. The predictions of disorder seem a much better tool.

We agree that the same prediction with two different model provides increased confidence. In

our case, the AlphaFold prediction seems to be correct as we have directly shown with NMR experimental data that the MEC-2 C-terminus, that is not part of the transmembrane domain, does not adopt a fixed structure in solution. For this reason, AlphaFold prediction provides a clear and reliable visual representation of the disordered nature of the C-terminal region of MEC-2.

9. line 85 (Fig. 2b): I believe the readers would like more explanation of this panel. For example, presumably the boxed amino acids are those most common for disordered domains, but this information is not given. Also, what is the beige area in Fig. 2biii?

We have now added more information to the figure legends to facilitate the readers understanding. We have also added explanation that the beige area corresponds to the C-terminal unstructured domain.

10. lines 214-215 (“Strikingly, we observed that both proteins sorted into the same puncta and colocalized along the neurite (Fig. 3e).”) The image that is presented does not show what they say it does. The UNC-89 SH3 domain is expressed in most, if not all, of the neurite. I do not think that the data shown is compelling.

We agree, the representative images and overlays were not very convincing. We have thus collected better data (Extended Data fig. 5k) and presented the two channel separately in grey scale, to avoid misinterpretation of colocalization. We have also performed an unbiased quantification (see also comment above) and found a more significant colocalization for the UNC-89::SH3 domain with MEC-2, but not with the mutant MEC-2(R385H). We should note that the UNC-89 construct is not driven from its endogenous promoter but from a TRN-specific element, which likely leads to a higher expression a spillover into the neurite. Importantly, the higher intensity where MEC-2 is found suggests a higher colocalization of the UNC-89 construct.

11. line 293: “Stomatin forms higher order oligomers” See also Huber et al. Proc. Natl. Acad. Sci. USA 103:17079-17086, 2006 for podocin and MEC-2.

Thank you, we have added this reference to backup our statement.

12. Fig. 4 Should the top panel of 4b say donor? Is the TSMOD data the same as that shown in Fig. 1f? If so, this should be mentioned. What does the color code in 4d mean?

We have now redesigned the FRET figure and added significant more data to elaborate on the tension propagation mechanism. At the same time, we hope we have addressed all the issues raised in this point.

13. Discussion: I believe that the authors should enlarge their discussion. First, they really do not mention that MEC-2 and other proteins are membrane proteins and how phase separation acts with this localization.

We thank the referee for this suggestions. We have now expanded on the previous speculations and write starting in line 397: "Because MEC-2 is a membrane-associated protein with cholesterol binding ability and membrane anchors[20, 21], the membrane itself could direct the assembly process and the slowed diffusion within the plasma membrane might further delimit the extent and timescale of MEC-2 formation[22]. Conversely, the solid transition might itself produce a stress that reshapes the neuronal membrane [23, 24] and induce constriction of the axonal caliper[25], thus reposition the MeT channel complex close to cytoskeletal elements that are critical for mechanosensation [26]."

Second, they don't really expand their results to other members of this protein family. Such a discussion might be important given that the orientation of proteins within the member is different depending on whether they have the proline indicated above. For example, do proteins without this proline lack the unstructured domain?

We have now added a short discussion to other stomatin members, a discussion that includes the notion that many stomatin proteins are membrane associated and regulate ion channels including Piezo ion channels. We also surveyed the potential PRM sites on various stomatin homologs and collated this information in Extended Data Fig 1, which shows the location and sequence of potential SH3 binding domains. However, the mentioned conserved proline is likely not an SH3 binding motif, as it is buried within the folded PHB domain and, as the referee explained, might be even associated with the membrane. Thus, these sites are indicated in italics in the Figure and not discussed any further. Specifically we write starting in line 415: "MEC-2 is only one member of a large family of Stomatin proteins found in all animal phyla and even in bacteria and mitochondria. Despite their high sequence homology, a suggested unifying function is still elusive [16]. Many other stomatin members (Extended Data Fig. 1) share unstructured domains with predicted proline-rich SH3 binding sites at their respective termini, that might select cell-specific function and localization. Due to their low conservation, the context of these sequences needs to be evaluated whether or not heterotypic buffering could underlie their regulatory functions."

Rebuttal References

- [1] Katta, S., Sanzeni, A., Das, A., Vergassola, M., Goodman, M. B., Progressive recruitment of distal MEC-4 channels determines touch response strength in *C. elegans*. *The Journal of general physiology* **151**, 1213–1230 (2019).
- [2] Ho, J., Tumkaya, T., Aryal, S., Choi, H., Claridge-Chang, A., Moving beyond P values: data analysis with estimation graphics. *Nature Methods* **16**, 565–566 (2019).
- [3] Das, R., et al., An asymmetric mechanical code ciphers curvature-dependent proprioceptor activity. *Science Advances* **7**, 1–20 (2021).
- [4] Suzuki, H., et al., In vivo imaging of *C. elegans* mechanosensory neurons demonstrates a specific role for the MEC-4 channel in the process of gentle touch sensation. *Neuron* **39**, 1005–1017 (2003).
- [5] Eastwood, A. L., et al., Tissue mechanics govern the rapidly adapting and symmetrical response to touch. *Proceedings of the National Academy of Sciences* **113**, E2471–E2471 (2015).
- [6] Nekimken, A. L., et al., Pneumatic stimulation of *C. elegans* mechanoreceptor neurons in a microfluidic trap. *Lab Chip* **17**, 1116–1127 (2017).
- [7] Loewenstein, W. R., Mendelson, M., Components of receptor adaptation in a Pacinian corpuscle. *The Journal of Physiology* **177**, 377–397 (1965).
- [8] Katta, S., Krieg, M., Goodman, M. B., Feeling Force: Physical and Physiological Principles Enabling Sensory Mechanotransduction. *Annual Review of Cell and Developmental Biology* **31**, 347–371 (2015).
- [9] Sanzeni, A., et al., Somatosensory neurons integrate the geometry of skin deformation and mechanotransduction channels to shape touch sensing. *eLife* **8**, 1–44 (2019).
- [10] Zhang, S., et al., MEC-2 Is Recruited to the Putative Mechanosensory Complex in *C. elegans* Touch Receptor Neurons through Its Stomatin-like Domain. *Current Biology* **131**, 1888–1896 (2004).
- [11] Mathieu, C., Pappu, R. V., Paul Taylor, J., Beyond aggregation: Pathological phase transitions in neurodegenerative disease. *Science* **370**, 56–60 (2020).

- [12] Qi, Y., et al., Membrane stiffening by STOML3 facilitates mechanosensation in sensory neurons. *Nature Communications* **6**, 8512 (2015).
- [13] Brand, J., et al., A stomatin dimer modulates the activity of acid-sensing ion channels. *The EMBO Journal* **31**, 3635–3646 (2012).
- [14] Poole, K., Moroni, M., Lewin, G. R., Sensory mechanotransduction at membrane-matrix interfaces. *Pflügers Archiv - European Journal of Physiology* 1–12 (2014).
- [15] Price, M. P., Thompson, R. J., Eshcol, J. O., Wemmie, J. A., Benson, C. J., Stomatin modulates gating of acid-sensing ion channels. *Journal of Biological Chemistry* **279**, 53886–53891 (2004).
- [16] Lapatsina, L., Brand, J., Poole, K., Daumke, O., Lewin, G. R., Stomatin-domain proteins. *European Journal of Cell Biology* **91**, 240–245 (2012).
- [17] Wetzel, C., et al., A stomatin-domain protein essential for touch sensation in the mouse. *Nature* **445**, 206–209 (2007).
- [18] Chen, Y., Bharill, S., Isacoff, E. Y., Chalfie, M., Subunit composition of a DEG/ENaC mechanosensory channel of *Caenorhabditis elegans*. *Proceedings of the National Academy of Sciences* **4**, 201515968 (2015).
- [19] Anishkin, A., Kung, C., Stiffened lipid platforms at molecular force foci. *Proceedings of the National Academy of Sciences of the United States of America* **110**, 4886–4892 (2013).
- [20] Huber, T. B., et al., Podocin and MEC-2 bind cholesterol to regulate the activity of associated ion channels. *Proceedings of the National Academy of Sciences of the United States of America* **103**, 17079–17086 (2006).
- [21] Brown, A. L., Liao, Z., Goodman, M. B., MEC-2 and MEC-6 in the *Caenorhabditis elegans* sensory mechanotransduction complex: Auxiliary subunits that enable channel activity. *Journal of General Physiology* **131**, 605–616 (2008).
- [22] Snead, W. T., et al., Membrane surfaces regulate assembly of ribonucleoprotein condensates. *Nature Cell Biology* **24**, 461–470 (2022).
- [23] Alberti, S., Gladfelter, A., Mittag, T., Considerations and Challenges in Studying Liquid-Liquid Phase Separation and Biomolecular Condensates. *Cell* **176**, 419–434 (2019).

- [24] Bergeron-Sandoval, L.-P., et al., Proteins with prion-like domains can form viscoelastic condensates that enable membrane remodeling and endocytosis. *PNAS* **118**, e211378911 (2021).
- [25] Pan, X., Zhou, Y., Hotulainen, P., Meunier, F. A., Wang, T., The axonal radial contractility: Structural basis underlying a new form of neural plasticity. *BioEssays* **43**, 1–13 (2021).
- [26] Bounoutas, A., Chalfie, M., Touch sensitivity in *Caenorhabditis elegans*. *Pflügers Archiv - European Journal of Physiology* **454**, 691–702 (2007).

Decision Letter, first revision:

*Please delete the link to your author homepage if you wish to forward this email to co-authors.

Dear Professor Krieg,

I do apologize once again for this very long delay. While Reviewer #3 was unable to review your revisions, as mentioned, Reviewer #1 has assessed your responses to Reviewer #3's previous concerns. Thank you as well for sending today your proposed responses to the current Reviewer concerns, which we have now discussed within the editorial team.

Your manuscript, "A rigidity phase transition of Stomatin condensates governs a switch from transport to mechanotransduction", has now been seen by 2 of our original referees, who are experts in interneuron communication (referee 1); biomolecular condensation (referee 2). As you will see from their comments (attached below) they find this work of interest, but have raised some important points. Although we are also very interested in this study, we believe that their concerns should be addressed before we can consider publication in Nature Cell Biology.

Nature Cell Biology editors discuss the referee reports in detail within the editorial team, including the chief editor, to identify key referee points that should be addressed with priority, and requests that are overruled as being beyond the scope of the current study. To guide the scope of the revisions, I have listed these points below. We are committed to providing a fair and constructive peer-review process, so please feel free to contact me if you would like to discuss any of the referee comments further.

In particular, it would be essential to:

A) Clarify concerns about unclear MEC2 dynamics and condensation, and their effects on mechanosensation (Reviewer #1)

B) Address concerns about potential physiological relevance of these condensates (Reviewer #1's assessment of your replies to Reviewer #3's previous concerns)

C) All other referee concerns pertaining to strengthening existing data, providing controls, methodological details, clarifications and textual changes, should also be addressed.

D) Finally please pay close attention to our guidelines on statistical and methodological reporting (listed below) as failure to do so may delay the reconsideration of the revised manuscript. In particular please provide:

- a Supplementary Table including all numerical source data in Excel format, with data for different

figures provided as different sheets within a single Excel file. The file should include source data giving rise to graphical representations and statistical descriptions in the paper and for all instances where the figures present representative experiments of multiple independent repeats, the source data of all repeats should be provided.

We therefore invite you to take these points into account when revising the manuscript. In addition, when preparing the revision please:

- ensure that it conforms to our format instructions and publication policies (see below and www.nature.com/nature/authors/).
- provide a point-by-point rebuttal to the full referee reports verbatim, as provided at the end of this letter.
- provide the completed Editorial Policy Checklist (found here <https://www.nature.com/authors/policies/Policy.pdf>), and Reporting Summary (found here <https://www.nature.com/authors/policies/ReportingSummary.pdf>). This is essential for reconsideration of the manuscript and these documents will be available to editors and referees in the event of peer review. For more information see <http://www.nature.com/authors/policies/availability.html> or contact me.

Nature Cell Biology is committed to improving transparency in authorship. As part of our efforts in this direction, we are now requesting that all authors identified as 'corresponding author' on published papers create and link their Open Researcher and Contributor Identifier (ORCID) with their account on the Manuscript Tracking System (MTS), prior to acceptance. ORCID helps the scientific community achieve unambiguous attribution of all scholarly contributions. You can create and link your ORCID from the home page of the MTS by clicking on 'Modify my Springer Nature account'. For more information please visit www.springernature.com/orcid.

[Redacted]

We would like to receive the revision within four weeks. If submitted within this time period, reconsideration of the revised manuscript will not be affected by related studies published elsewhere, or accepted for publication in Nature Cell Biology in the meantime. We would be happy to consider a revision even after this timeframe, but in that case we will consider the published literature at the time of resubmission when assessing the file.

We hope that you will find our referees' comments, and editorial guidance helpful. Please do not hesitate to contact me if there is anything you would like to discuss.

Best wishes,

Daryl Jason David

Daryl Jason Verzosa David, PhD

Senior Editor, Nature Cell Biology
Nature Portfolio

Heidelberger Platz 3, 14197 Berlin, Germany
Email: daryl.david@nature.com
ORCID: <https://orcid.org/0000-0002-9253-4805>

Reviewers' Comments:

Reviewer #1:

Remarks to the Author:

In the revised manuscript, the authors have performed two key experiments I requested in my original review. I appreciate the authors' efforts to address my concerns. However, the results obtained seem somewhat contradictory to the conclusion the authors suggested in this manuscript.

In the first experiment I requested, the authors performed FRAP experiments for R385H mutant in comparison with WT in TRN neurons (Extended Fig. 4g,h) and hypodermis (Extended Data Fig. 4i).

On lines 155-157, the authors state: The recovery in FRAP experiments of the immobile puncta formed by mature mutant MEC-2 was similar to that of formed by the wild type during the course of our experiments (Extended Data Fig. 4g,h). However, this statement is inaccurate. The real graph of Extended Data Fig. 4h shows significant increases in the time for half recovery ($t_{1/2}$, $p = 0.03$) and the mobile fraction (Mf, $p = 5e^{-4}$) in the R385H mutant compared with WT. In TRN neurons, endogenous UNC-89 is present. UNC-89 SH3 domain binds proline rich motif (PRM) of WT better than that of R385H mutant, which should facilitate the maturity (rigidity phase transition) of MEC2 WT more than the R385H mutant. Thus, $t_{1/2}$ of WT should be larger than that of R385H mutant, but the result is opposite. The authors need to explain this apparent discrepancy.

As described on lines 157-159 (However, the naive pool corresponding to mutant MEC-2 (R385H) expressed in the hypodermis recovered significantly faster than wild type (Extended Data Fig. 4i)), the R385H mutant is more mobile than WT by nature presumably due to the reduced homotypic interaction (Extended Data Fig. 4n). Thus, it is puzzling why the R385H mutant in TRN neurons exhibits more rigidity (slower recovery) than WT in Extended Data Fig. 4h.

In the second experiment I requested, the authors performed FRAP experiments for WT MEC2 in the WT vs unc-89 mutant (Extended Data Fig. 6d, e, f). My naive expectation is, WT MEC2 proteins are more fluid in the unc-89 mutant than in WT. The authors did not find significant difference between these two conditions and stated on lines 220-223: However, we did not detect significant changes in

the recovery dynamics after photobleaching puncta of the unc-89 mature pool (Extended Data Fig. 6d), suggesting either compensation with other SH3 domains⁵⁸ or that passive rheological properties of MEC-2 are not affected by the unc-89 knockout (see below, and ref. 59).

Assuming other SH3 domains compensate for the maturity of MEC-2 in the unc-89 knockout, then why does unc-89 knockout exhibit a phenotype in touch response in Fig. 3a?

I must admit that I am not the expert of the rigidity phase transition or mec-2 genetics in *C. elegans*. However, most readers of NCB are cell biologists who are the outside of the field like this reviewer. I admit that data of the in vitro experiments are strong enough. Nevertheless, I feel that the rigidity phase transition of MEC-2 proteins (in the presence or absence of UNC-89) in vivo (examined by FRAP experiments) does not seem to show a strong correlation with their functional outcome (touch response) as the authors indicated.

I have been asked to comment on authors' responses to Reviewer #3's previous concerns. Overall, I think that the authors responded to them in satisfactory manners.

However, I have specific comments on the authors' responses to Reviewer #3's previous concerns:

1) Reviewer #3's concern #4: line 39: (Two distinct populations and Fig. 1a): The kymographs show either all "mature" and static or all "naive" and moving and the diagram separates these two areas along the neurite. Given that the animals are normally touch sensitive over the entire length of the neurite, it is surprising that the mature puncta do not appear to be seen in the proximal neurite. If mature puncta coexist, they might be color coded in the image to distinguish them.

I think that this is an important criticism. I am not an expert of MEC2 biology and naively misunderstood that the distal part of TRN is more sensitive to touch than proximal part...

To respond to Reviewer #3's concern, the authors replaced Fig. 1a with a new one. The previous Fig 1a looks like "all or nothing type" of distinct populations, which is no longer valid in a new Fig. 1a... With new Fig.1a and the notion that animals are normally touch sensitive over the entire length of the neurite, then the physiological significance/impact of differential localization of naive vs mature MEC-2 proteins in TRN neurites seems less clear than that in the original submission.

2) Reviewer #3's concern #10: lines 214-215 ("Strikingly, we observed that both proteins sorted into the same puncta and colocalized along the neurite (Fig. 3e).") The image that is presented does not show what they say it does. The UNC-89 SH3 domain is expressed in most, if not all, of the neurite. I do not think that the data shown is compelling.

I think that this is also an important criticism. The authors responded: We agree, the representative images and overlays were not very convincing. We have thus collected better data (Extended Data fig. 5k) and presented the two channel separately in grey scale, to avoid misinterpretation of colocalization. We have also performed an unbiased quantification (see also comment above) and found a more significant colocalization for the UNC-89::SH3 domain with MEC-2, but not with the mutant MEC-2(R385H). We should note that the UNC-89 construct is not driven from its endogenous promoter but from a TRN-specific element, which likely leads to a higher expression a spillover into the neurite. Importantly, the higher intensity where MEC-2 is found suggests a higher colocalization of the UNC-89 construct.

I think that the authors' response is quite satisfactory. Yet, in kymograph of Fig. 3e there is only one mobile condensate of MEC-2 WT::mCherry and no UNC-89 SH3::GFP puncta shows a co-migration. If the authors' claim is correct, in the proximal region, we would anticipate a significant number of mobile puncta of MEC2 WT::mCherry that co-migrate with UNC-89 SH3::GFP puncta. I wonder whether there is such data. If so such data would strengthen the authors' argument of colocalization.

Reviewer #2:

Remarks to the Author:

In their revised manuscript, the authors performed extensive experiments and analysis to improve their manuscript. They also sufficiently responded to the concerns of this reviewer. The manuscript as currently written provides a convincing and complete picture of how the material properties of the MEC-2 condensate can be changed over time and accelerated by a binding partner, UNC-89, to affect condensate function in mechanotransduction. This reviewer commends the authors on their effort and fully supports the publication of this manuscript in Nature Cell Biology. The results of this study will impact numerous fields in the biological sciences and provide a new lens through which to view the relationship between biophysical properties and condensate functions.

GUIDELINES FOR SUBMISSION OF NATURE CELL BIOLOGY ARTICLES

ARTICLE FORMAT

AUTHOR AFFILIATIONS – should be denoted with numerical superscripts (not symbols) preceding the names. Full addresses should be included, with US states in full and providing zip/post codes. The

corresponding author is denoted by: "Correspondence should be addressed to [initials]."

ABSTRACT – should not exceed 150 words and should be unreferenced. This paragraph is the most visible part of the paper and should briefly outline the background and rationale for the work, and accurately summarize the main results and conclusions. Key genes, proteins and organisms should be specified to ensure discoverability of the paper in online searches.

TEXT – the main text consists of the Introduction, Results, and Discussion sections and must not exceed 3500 words including the abstract. The Introduction should expand on the background relating to the work. The Results should be divided in subsections with subheadings, and should provide a concise and accurate description of the experimental findings. The Discussion should expand on the findings and their implications. All relevant primary literature should be cited, in particular when discussing the background and specific findings.

REFERENCES – are limited to a total of 70 in the main text and Methods combined,. They must be numbered sequentially as they appear in the main text, tables and figure legends and Methods and must follow the precise style of Nature Cell Biology references. References only cited in the Methods should be numbered consecutively following the last reference cited in the main text. References only associated with Supplementary Information (e.g. in supplementary legends) do not count toward the total reference limit and do not need to be cited in numerical continuity with references in the main text. Only published papers can be cited, and each publication cited should be included in the numbered reference list, which should include the manuscript titles. Footnotes are not permitted.

Methods should be written concisely, but should contain all elements necessary to allow interpretation and replication of the results. As a guideline, Methods sections typically do not exceed 3,000 words. The Methods should be divided into subsections listing reagents and techniques. When citing previous methods, accurate references should be provided and any alterations should be noted. Information must be provided about: antibody dilutions, company names, catalogue numbers and clone numbers for monoclonal antibodies; sequences of RNAi and cDNA probes/primers or company names and

catalogue numbers if reagents are commercial; cell line names, sources and information on cell line identity and authentication. Animal studies and experiments involving human subjects must be reported in detail, identifying the committees approving the protocols. For studies involving human subjects/samples, a statement must be included confirming that informed consent was obtained. Statistical analyses and information on the reproducibility of experimental results should be provided in a section titled "Statistics and Reproducibility".

All Nature Cell Biology manuscripts submitted on or after March 21 2016, must include a Data availability statement as a separate section after Methods but before references, under the heading "Data Availability". For Springer Nature policies on data availability see <http://www.nature.com/authors/policies/availability.html>; for more information on this particular policy see <http://www.nature.com/authors/policies/data/data-availability-statements-data-citations.pdf>. The Data availability statement should include:

- Accession codes for primary datasets (generated during the study under consideration and designated as "primary accessions") and secondary datasets (published datasets reanalysed during the study under consideration, designated as "referenced accessions"). For primary accessions data should be made public to coincide with publication of the manuscript. A list of data types for which submission to community-endorsed public repositories is mandated (including sequence, structure, microarray, deep sequencing data) can be found here <http://www.nature.com/authors/policies/availability.html#data>.
- Unique identifiers (accession codes, DOIs or other unique persistent identifier) and hyperlinks for datasets deposited in an approved repository, but for which data deposition is not mandated (see here for details <http://www.nature.com/sdata/data-policies/repositories>).
- At a minimum, please include a statement confirming that all relevant data are available from the authors, and/or are included with the manuscript (e.g. as source data or supplementary information), listing which data are included (e.g. by figure panels and data types) and mentioning any restrictions on availability.
- If a dataset has a Digital Object Identifier (DOI) as its unique identifier, we strongly encourage including this in the Reference list and citing the dataset in the Methods.

We recommend that you upload the step-by-step protocols used in this manuscript to the Protocol Exchange. More details can found at www.nature.com/protocolexchange/about.

DISPLAY ITEMS – main display items are limited to 6-8 main figures and/or main tables. For Supplementary Information see below.

FIGURES – Colour figure publication costs \$395 per colour figure. All panels of a multi-panel figure must be logically connected and arranged as they would appear in the final version. Unnecessary figures and figure panels should be avoided (e.g. data presented in small tables could be stated briefly in the text instead).

All imaging data should be accompanied by scale bars, which should be defined in the legend. Cropped images of gels/blots are acceptable, but need to be accompanied by size markers, and to

retain visible background signal within the linear range (i.e. should not be saturated). The boundaries of panels with low background have to be demarked with black lines. Splicing of panels should only be considered if unavoidable, and must be clearly marked on the figure, and noted in the legend with a statement on whether the samples were obtained and processed simultaneously. Quantitative comparisons between samples on different gels/blots are discouraged; if this is unavoidable, it has to be performed for samples derived from the same experiment with gels/blots were processed in parallel, which needs to be stated in the legend.

- For line art, graphs, charts and schematics we prefer Adobe Illustrator (.AI), Encapsulated PostScript (.EPS) or Portable Document Format (.PDF). Files should be saved or exported as such directly from the application in which they were made, to allow us to restyle them according to our journal house style.
- We accept PowerPoint (.PPT) files if they are fully editable. However, please refrain from adding PowerPoint graphical effects to objects, as this results in them outputting poor quality raster art. Text used for PowerPoint figures should be Helvetica (preferred) or Arial.
- We do not recommend using Adobe Photoshop for designing figures, but we can accept Photoshop generated (.PSD or .TIFF) files only if each element included in the figure (text, labels, pictures, graphs, arrows and scale bars) are on separate layers. All text should be editable in 'type layers' and line-art such as graphs and other simple schematics should be preserved and embedded within 'vector smart objects' - not flattened raster/bitmap graphics.
- Some programs can generate Postscript by 'printing to file' (found in the Print dialogue). If using an application not listed above, save the file in PostScript format or email our Art Editor, Allen Beattie for advice (a.beattie@nature.com).

Regardless of format, all figures must be vector graphic compatible files, not supplied in a flattened raster/bitmap graphics format, but should be fully editable, allowing us to highlight/copy/paste all text and move individual parts of the figures (i.e. arrows, lines, x and y axes, graphs, tick marks, scale bars etc). The only parts of the figure that should be in pixel raster/bitmap format are photographic images or 3D rendered graphics/complex technical illustrations.

All placed images (i.e. a photo incorporated into a figure) should be on a separate layer and independent from any superimposed scale bars or text. Individual photographic images must be a minimum of 300+ DPI (at actual size) or kept constant from the original picture acquisition and not

decreased in resolution post image acquisition. All colour artwork should be RGB format.

Unprocessed scans of all key data generated through electrophoretic separation techniques need to be presented in a supplementary figure that should be labeled and numbered as the final supplementary figure, and should be mentioned in every relevant figure legend. This figure does not count towards the total number of figures and is the only figure that can be displayed over multiple pages, but should be provided as a single file, in PDF or TIFF format. Data in this figure can be displayed in a relatively informal style, but size markers and the figures panels corresponding to the presented data must be indicated.

The total number of Supplementary Figures (not including the “unprocessed scans” Supplementary Figure) should not exceed the number of main display items (figures and/or tables (see our Guide to Authors and March 2012 editorial <http://www.nature.com/ncb/authors/submit/index.html#suppinfo>; <http://www.nature.com/ncb/journal/v14/n3/index.html#ed>). No restrictions apply to Supplementary Tables or Videos, but we advise authors to be selective in including supplemental data.

GUIDELINES FOR EXPERIMENTAL AND STATISTICAL REPORTING

REPORTING REQUIREMENTS – To improve the quality of methods and statistics reporting in our papers we have recently revised the reporting checklist we introduced in 2013. We are now asking all life sciences authors to complete two items: an Editorial Policy Checklist (found here <https://www.nature.com/authors/policies/Policy.pdf>) that verifies compliance with all required editorial policies and a Reporting Summary (found here <https://www.nature.com/authors/policies/ReportingSummary.pdf>) that collects information on experimental design and reagents. These documents are available to referees to aid the evaluation of the manuscript. Please note that these forms are dynamic 'smart pdfs' and must therefore be downloaded and completed in Adobe Reader. We will then flatten them for ease of use by the reviewers. If you would like to reference the guidance text as you complete the template, please access these flattened versions at <http://www.nature.com/authors/policies/availability.html>.

Author Rebuttal, first revision:

We would like to thank the two referees for their encouraging comments and their help in improving the presentation of the results. We have now responded to the remaining concerns of reviewer #1. Our answers can be read below, highlighted in blue and in the main text of the manuscript highlighted in green. We hope that these explanations are clear and that modifying the manuscript accordingly lead to clearer understanding of the underlying mechanisms.

Reviewers' comments:

Reviewer #1 (Remarks to the Author):

In the revised manuscript, the authors have performed two key experiments I requested in my original review. I appreciate the authors' efforts to address my concerns. However, the results obtained seem somewhat contradictory to the conclusion the authors suggested in this manuscript.

We appreciate the reviewer's thorough analysis and thank him/her for identifying points that indeed merit clarification in the manuscript. Our response to the specific points can be seen in darkblue.

1. In the first experiment I requested, the authors performed FRAP experiments for R385H mutant in comparison with WT in TRN neurons (Extended Fig. 4g,h) and hypodermis (Extended Data Fig. 4i).

In the original revision, the reviewer suggested to review and revisit the sol-gel transition *in vivo*, which we agree was insufficiently explored. In addition to the mentioned FRAP experiment, we would like to emphasize that we have conducted several experiments to further explore this mechanism *in vivo* and *in vitro*:

- (a) We performed FRAP on wt and MEC-2 mutant puncta in TRNs. We observed a significant higher Mf for the mutant MEC-2, with little changes on the diffusion time scale (see our explanation below).
- (b) We performed FRAP on wt and MEC-2 mutant puncta in the hypodermis. We have repeated these experiments and display their results in ED Fig. 4, including their average recovery curves and the Mf and diffusion coefficient.
- (c) We measured FRET-force relation of MEC-2 condensates under different conditions (R385H mutant, in the hypodermis, in unc-89 mutant and a c-terminally truncated version). Consistently, we observed a lower ability of MEC-2 to bear mechanical load in these genetic manipulations.

On lines 155-157, the authors state: *The recovery in FRAP experiments of the immobile puncta formed by mature mutant MEC-2 was similar to that of formed by the wild type during the course of our experiments (Extended Data Fig. 4g,h).*

However, this statement is inaccurate. The real graph of Extended Data Fig. 4h shows significant increases in the time for half recovery ($t_{1/2}$, $p = 0.03$) and the mobile fraction (Mf, $p = 5e-4$) in the R385H mutant compared with WT. In TRN neurons, endogenous UNC-89 is present. UNC-89 SH3 domain binds proline rich motif (PRM) of WT better than that of R385H mutant, which should facilitate the maturity (rigidity phase transition) of MEC2 WT more than the R385H mutant. Thus, $t_{1/2}$ of WT should be larger than that of R385H mutant, but the result is opposite. The authors need to explain this apparent discrepancy.

The Reviewer makes a fair point: since R385H MEC-2 does not interact with the UNC-89 SH3 domain, an interaction that triggers the liquid-to-solid transition, one could expect that the R385H MEC-2 condensates recover fluorescence faster (lower $t_{1/2}$), and to a larger extent (higher Mf) than WT MEC-2 condensates in FRAP experiments. In response to this we would like to make a technical point and some scientific points.

First we would like to emphasize that the difference in mobile fraction is larger, statistically more robust and in agreement with the expectation of the Reviewer: indeed all data points in the normalized recovery curve, except one, are higher for the mutant than for WT MEC-2. We could argue that the result is therefore closer to expectation but prefer to conclude that the difference in rate of recovery is small: indeed, after adjusting the recovery half-time by the size of the bleach spot, the diffusion coefficients of mutant and WT MEC-2 are indistinguishable ($p=0.2$, two-sided Welch test). Even though this adjustment does not affect the overall conclusion, it affects the relative difference in the descriptive variable, especially when bleach spots differ substantially (as in Fig. 1g). For consistency, we have now changed the plots and replaced the recovery half time for the diffusion coefficient throughout the manuscript.

Why is the diffusion unchanged between the mutant and the wildtype condensates? As FRAP recovery measures the diffusion and mobility of the labeled molecular species, in our case MEC-2, it provides information about the viscosity and fluidity of the associated condensates. Indeed, although the rate of diffusion seems not altered by the mutation, the higher mobile fraction (Mf) obtained for the mutant suggests it recovers fluorescence to a larger extent than the WT, indicating that a significant fraction of WT MEC-2 is not diffusive. Although this observation is in agreement with our conclusion that WT has undergone a liquid-to-solid transition we consider that the important point below is more relevant to the discussion. Specifically, we write in the manuscript: *“We examined the impact of the MEC-2 mutation on both mature and naive condensates using FRAP. As expected, the overall recovery dynamics of MEC-2 within the naive pool in the hypodermis was faster and more complete than that of MEC-2 in the mature pool (Extended Data Fig. 4g-j). Additionally, we observed that the mutant MEC-2 exhibited a higher mobile fraction (Mf) in both the mature and naive populations compared to the wild-type MEC-2 (Extended Data Fig. 4h, j). However, the diffusion coefficient derived from the recovery curves of mature puncta formed by the mutant MEC-2 remained unchanged compared to those formed by the wild-type (Extended Data Fig. 4i, j), which mirrored the findings obtained in the in vitro experiments (Extended Data Fig. 4f). Notably, while FRAP recovery provides insights into the mobility of MEC-2 within the condensate, and thus viscosity ($D \propto \eta^{-1}$), it does not offer information about elastic properties. Together, despite the unchanged diffusion rate caused by the mutation, the higher mobile fraction (Extended Data Fig. 4j) observed in the mutant indicates that a significant portion of wild-type MEC-2 does not participate in the recovery process within mature condensates.”*

A further point that we would like to add to the discussion is that in the immobile, mature condensates of mutant and WT MEC-2 studied by FRAP in Extended Data Fig. 4g,h the liquid-to-solid transition has already taken place. It may seem counter-intuitive that the condensates formed by mutant MEC-2 mature despite it not interacting with UNC-89 but, as recently proposed for other proteins (Ranganathan and Shakhnovich (2020) eLife; Mittag and Pappu (2022) Mol Cell), the naive MEC-2 condensates are metastable viscoelastic fluids with time-dependent material properties even in the absence of this interaction.

As recently shown (Mathieu, Pappu and Taylor (2020) Science; Alshareedah et al (2023) bioRxiv) the sequences of phase separating proteins provide kinetic stability to the condensates so that, despite their metastability, they remain dynamic liquid (naive) in the relevant biological timescale - a striking example of this is provided by a manuscript authored by some of the co-authors of the current manuscript, currently under review (Garcia-Cabau, Bartomeu et al (2023) bioRxiv). In this scenario UNC-89 allows mechanosensation because, by interacting with MEC-2, it decreases the kinetic stability of the condensates, thus triggering their maturation at the time required for mechanosensation. Indeed, in conditions when the UNC-89 SH3 domain is overexpressed, we noticed substantially lower number of naive, mobile MEC-2 condensates (e.g. kymograph in Fig. 3e and also see our response to comments below).

We acknowledge that these concepts were not sufficiently developed in the revised version, in part because they relate to recent developments in this field, and thank Reviewer #1 for bringing them up during the review process. Specifically we write now in the discussion: *“Our data suggest a plausible molecular mechanism*

for the rigidification of the condensates: in this mechanism binding of the UNC-89 SH3 domain to the PRM would trigger the liquid-to-solid transition because the PRM, or residues in its vicinity, preserves the liquid character by interacting with a yet-to-be identified aggregation-prone motif in the C-terminal domain of MEC-2 by a mechanism known as heterotypic buffering (Matthieu et al, 2020). Binding of the UNC-89 SH3 domain to the PRM would, by competing against the interaction responsible for buffering, expose the aggregation-prone motif, thus leading to the liquid-to-solid transition. Indeed, in the absence of UNC-89, MEC-2 forms stiff condensates after 48 h in vitro but the addition of UNC-89 SH3 domain, at a 1:0.1 molar ratio, allows the transition to occur in only a few hours. This is also reflected in vivo, even though MEC-2 can form mature condensates with distinct mechanical properties in absence of UNC-89, because the overexpression of UNC-89 SH3 domain increases the relative size of the mature pool, in expense of the naive one. In summary, we propose that UNC-89 regulates mechanosensation because, by interacting with MEC-2, it triggers the maturation of the MEC-2 condensates at the times and locations where it is required for this process”

2. As described on lines 157-159

However, the naive pool corresponding to mutant MEC-2 (R385H) expressed in the hypodermis recovered significantly faster than wild type (Extended Data Fig. 4i). the R385H mutant is more mobile than WT by nature presumably due to the reduced homotypic interaction (Extended Data Fig. 4n). Thus, it is puzzling why the R385H mutant in TRN neurons exhibits more rigidity (slower recovery) than WT in Extended Data Fig. 4h.

In addition to our response to the comments above, we would like to emphasize that FRAP recovery rates measure the diffusion of the mobile species and, as such, allows measuring sample viscosity; however, they *per se* do not measure changes in elasticity or rigidity. In other words, these experiments are necessary but not sufficient to study the changes in mechanical rigidity caused by liquid-to-solid transitions. This apparent discrepancy will be resolved later in the text with the results from the experiments presented in Fig. 5 and Fig. 6. For this reason, we have performed MEC-2::TSMOD experiments to measure tensions *in vivo* and indeed found that the condensates formed by R385H MEC-2 have a substantially lower mechanical tension and a reduced ability to bear mechanical load (Fig. 6). We also provided an extended discussion on why the passive diffusion timescales differ from the viscoelastic relaxation timescales in the rheology experiments in the supplementary discussion section of the SI text.

3. In the second experiment I requested, the authors performed FRAP experiments for WT MEC2 in the WT vs unc-89 mutant (Extended Data Fig. 6d, e, f). My naive expectation is, WT MEC2 proteins are more fluid in the unc-89 mutant than in WT. The authors did not find significant difference between these two conditions and stated on lines 220-223: However, we did not detect significant changes in the recovery dynamics after photobleaching puncta of the unc-89 mature pool (Extended Data Fig. 6d), suggesting either compensation with other SH3 domains⁵⁸ or that passive rheological properties of MEC-2 are not affected by the unc-89 knockout (see below, and ref. 59). Assuming other SH3 domains compensate for the maturity of MEC-2 in the unc-89 knockout, then why does unc-89 knockout exhibit a phenotype in touch response in Fig. 3a?

To explain the apparent discrepancy noted by Reviewer #1 we would like to highlight that, similarly to the reasoning above, FRAP provides information about the mobility of the labeled molecular species in the condensate, not about the material properties of the condensate: the tension sensor experiments provide a more informative result, one that describes the rigidity transition and its connection with the sense of touch. What we can say from our data is that the diffusion is different between the naive and mature condensates, but once they have undergone the rigidity transition the diffusion does not depend on UNC-89 anymore. In addition, we should consider that these results come from different experimental set-ups and can therefore have different outcomes: during touch, in the behavior assay, and in the microfluidic device, the MEC-2/SH3 complex is under mechanical tension, whereas during the FRAP experiments the fluorescence recovery is an equilibrium

process that takes place in the absence of mechanical tension. These experiments reveal that the absence of the SH3 domain becomes phenotypic under mechanical load, not under equilibrium conditions and similar observations have been made in the field of mechanobiology, where passive and active processes have been compared directly side-by-side (Jawerth et al. (2020) Science). In this particular study, the authors detected a liquid-to-solid transition after a 24-48h with passive microrheology, while active microrheology is able to detect the rigidity transition already after a few hours.

4. I must admit that I am not the expert of the rigidity phase transition or *mec-2* genetics in *C. elegans*. However, most readers of NCB are cell biologists who are the outside of the field like this reviewer. I admit that data of the *in vitro* experiments are strong enough. Nevertheless, I feel that the rigidity phase transition of MEC-2 proteins (in the presence of absence of UNC-89) *in vivo* (examined by FRAP experiments) does not seem to show a strong correlation with their functional outcome (touch response) as the authors indicated.

As explained above, the interaction between UNC-89 and MEC-2 accelerates the phase transition which leads to a change in condensate mobility and rigidity. The absence of UNC-89, however, does not seem to alter MEC-2::mCherry diffusion once the condensates are matured. With our data derived from a MEC-2::TSMOD FRET-tension sensor, included during the revision process, we show in Fig. 6 how the tension of MEC-2 condensates depends on its interaction with UNC-89. In animals lacking *unc-89*, significantly less tension developed in the MEC-2 condensates and respond less to external touch. We commented on these on experiments in the original response letter, but acknowledge that they were not sufficiently well explained.

Reviewer 1 on Reviewer 3

I have been asked to comment on authors' responses to Reviewer #3's previous concerns. Overall, I think that the authors responded to them in satisfactory manners.

However, I have specific comments on the authors' responses to Reviewer #3's previous concerns:

1. Reviewer #3's concern #4: line 39: (Two distinct populations and Fig. 1a): The kymographs show either all "mature" and static or all "naive" and moving and the diagram separates these two areas along the neurite. Given that the animals are normally touch sensitive over the entire length of the neurite, it is surprising that the mature puncta do not appear to be seen in the proximal neurite. If mature puncta coexist, they might be color coded in the image to distinguish them.

I think that this is an important criticism. I am not an expert of MEC2 biology and naively misunderstood that the distal part of TRN is more sensitive to touch than proximal part... To respond to Reviewer #3's concern, the authors replaced Fig. 1a with a new one. The previous Fig 1a looks like "all or nothing type" of distinct populations, which is no longer valid in a new Fig. 1a. With new Fig.1a and the notion that animals are normally touch sensitive over the entire length of the neurite, then the physiological significance/impact of differential localization of naive vs mature MEC-2 proteins in TRN neurites seems less clear than that in the original submission.

We responded to the previous suggestion of referee 3 and made it clearly visible that the two populations are not mutually exclusive but overlapping. As requested by the referee, we replaced the two separate kymographs with a continuous that highlights the behavior of the mobile and immobile pool within a single neuron, instead of color coding them. Even in the old figure panel the immobile pool was visible but the reason for the apparent absence in the old picture is likely the fact that the mobile pool has a much higher fluorescence brightness and thus obscures the immobile pool in the first kymographs. To better visualize that these two populations overlap, we have replaced the two original kymographs with one continuous example. In addition to their difference in mobility of the naive and mature pool, we have shown that the MEC-2 condensation and maturation is critical for touch, as the c-terminal truncation with defects in condensation strongly interferes

with touch sensitivity (Fig. 1; ED Fig. 3). In absence of UNC-89 and the c-terminal tail, MEC-2 condensates do not sustain mechanical touch in the in-vivo FRET-tension sensor experiments. Specifically we wrote: *“To determine if the MEC-2 C-terminal domain and, especially, its propensity to phase separate into distinct condensates relates to its function in sensing mechanical touch we assayed the effect of the TEV cleavage on aversive behavior. Consistent with our hypothesis, the conditional C-terminally truncated MEC-2 was unable to transduce external touch into avoidance behavior (Fig. 1h); importantly, we made the same observations with a constitutive truncated MEC-2 construct lacking the C-terminal domain (Extended Data Fig. 3). Taken together, our results indicate that MEC-2 forms liquid-like phase-separated condensates that undergo a dynamic switch along the neurite and suggest a key role of the C-terminal domain in condensation and in the maturation, localization and function of the resulting condensates. ”*

2. Reviewer #3's concern #10: lines 214-215 (“Strikingly, we observed that both proteins sorted into the same puncta and colocalized along the neurite (Fig. 3e).”) The image that is presented does not show what they say it does. The UNC-89 SH3 domain is expressed in most, if not all, of the neurite. I do not think that the data shown is compelling.

I think that this is also an important criticism. The authors responded: *We agree, the representative images and overlays were not very convincing. We have thus collected better data (Extended Data fig. 5k) and presented the two channel separately in grey scale, to avoid misinterpretation of colocalization. We have also performed an unbiased quantification (see also comment above) and found a more significant colocalization for the UNC-89::SH3 domain with MEC-2, but not with the mutant MEC-2(R385H). We should note that the UNC-89 construct is not driven from its endogenous promotor but from a TRN-specific element, which likely leads to a higher expression a spillover into the neurite. Importantly, the higher intensity where MEC-2 is found suggests a higher colocalization of the UNC-89 construct.*

I think that the authors' response is quite satisfactory. Yet, in kymograph of Fig. 3ei there is only one mobile condensate of MEC-2 WT::mCherry and no UNC-89 SH3::GFP puncta shows a co-migration. If the authors' claim is correct, in the proximal region, we would anticipate a significant number of mobile puncta of MEC2 WT::mCherry that co-migrate with UNC-89 SH3::GFP puncta. I wonder whether there is such data. If so such data would strengthen the authors' argument of colocalization.

The reviewer brings up an important point which needs further explanation in the text. To recapitulate, our working model up to this point was that binding of MEC-2 to UNC-89 accelerates a rigidity transition from the naive, mobile to the mature, immobile pool. The kymograph shows important features that we have described before: the immobile, mature MEC-2 puncta co-localize with UNC-89 whereas the mobile MEC-2 puncta do not. Our interpretation for this observation that UNC-89 is not found on mobile MEC-2 is because it accelerates the rigidity transition as we have also shown in vitro. Thus the existence of UNC-89 and mobile MEC-2 is mutually exclusive. Indeed, we observed substantially less naive, mobile MEC-2 condensates in the animal overexpressing the UNC-89:SH3 domain (see kymograph in Fig. 3e i and the quantification). We do think that this direct comparison is a compelling piece of information knitting together that the MEC-2/UNC-89 interaction is critical for the rigidity phase transition in vivo and associated difference in mechanical properties. Together, the kymograph, combined with the co-localization analysis makes a stronger point and supports the model that UNC-89 induces the rigidity transition and is thus found only associated to the immobile MEC-2 phase. Specifically we write: *“As seen in the kymograph, the colocalization was only observed for the static, immobile condensates, whereas the mobile MEC-2 puncta are not found to colocalize with UNC-89. This directly supports our finding that UNC-89 governs the transitions from mobile to immobile MEC-2 condensates. Indeed, we found significantly less naive, mobile MEC-2 condensates (compare Fig 1b, Fig. 3e, f) in TRNs expressing the UNC-89:SH3 protein, and a higher IPI (Extended Data Fig. 6a-c), leading to a lower mature pool compared to wt neurons in distal neurites (wt vs UNC-89::SH3, $p=0.015$, one-sided t-test). This effect appeared to be specific to the UNC-89:SH3 domain, as the SH3 domain of α -spectrin SPC-1 did not interfere*

with formation of the mobile condensates (Fig. 3f).”

Reviewer #2 (Remarks to the Author):

In their revised manuscript, the authors performed extensive experiments and analysis to improve their manuscript. They also sufficiently responded to the concerns of this reviewer. The manuscript as currently written provides a convincing and complete picture of how the material properties of the MEC-2 condensate can be changed over time and accelerated by a binding partner, UNC-89, to affect condensate function in mechanotransduction. This reviewer commends the authors on their effort and fully supports the publication of this manuscript in Nature Cell Biology. The results of this study will impact numerous fields in the biological sciences and provide a new lens through which to view the relationship between biophysical properties and condensate functions.

Decision Letter, second revision:

Our ref: NCB-A49055B

14th June 2023

Dear Dr. Krieg,

I'm writing on behalf of my colleague Dr Daryl David, who is out of the office.

Thank you for submitting your revised manuscript "A rigidity phase transition of Stomatin condensates governs a switch from transport to mechanotransduction" (NCB-A49055B). It has now been seen by the original Referee #1 and their comments are below. The reviewer finds that the paper has improved in revision, and therefore we'll be happy in principle to publish it in Nature Cell Biology, pending minor revisions to comply with our editorial and formatting guidelines.

****Please note that the current version of your manuscript is in a PDF format. Could you please email me a copy of the file in an editable format (Microsoft Word or LaTeX), as we can not proceed with PDFs at this stage?***

Once we have received your Word file, we will be performing detailed checks on your paper and will send you a checklist detailing our editorial and formatting requirements in about 1-2 weeks. Please do not upload the final materials or make any revisions until you receive this additional information from us.

Thank you again for your interest in Nature Cell Biology. Please do not hesitate to contact me if you have any questions.

Sincerely,

Melina

Melina Casadio, PhD
Senior Editor, Nature Cell Biology
ORCID ID: <https://orcid.org/0000-0003-2389-2243>

Reviewer #1 (Remarks to the Author):

While a large body of work suggests that biomolecular condensates ensuing from liquid-liquid phase separation mature into various material states, how this aging process is controlled and if the naive and mature phases can have differential functions is currently unknown. Using *Caenorhabditis elegans* as a model, the authors demonstrated that MEC-2 Stomatin undergoes a rigidity phase transition during maturation from fluid to viscoelastic, glass-like condensates that facilitate either transport or mechanotransduction.

I agree that that the authors' data demonstrate a novel function for rigidity maturation during mechanotransduction. The authors used various in vitro and in vivo approaches and provided numerous data to support their claims.

Through the 2nd round of review process, the authors thoroughly answered my concerns. Therefore, I support the acceptance of this manuscript to NCB.

Decision Letter, final checks:

Our ref: NCB-A49055B

7th July 2023

Dear Dr. Krieg,

Thank you for your patience as we've prepared the guidelines for final submission of your Nature Cell Biology manuscript, "A rigidity phase transition of Stomatin condensates governs a switch from transport to mechanotransduction" (NCB-A49055B). Please carefully follow the step-by-step instructions provided in the attached file, and add a response in each row of the table to indicate the changes that you have made. Please also check and comment on any additional marked-up edits we have proposed within the text. Ensuring that each point is addressed will help to ensure that your revised manuscript can be swiftly handed over to our production team.

In recognition of the time and expertise our reviewers provide to Nature Cell Biology's editorial process, we would like to formally acknowledge their contribution to the external peer review of your manuscript entitled "A rigidity phase transition of Stomatin condensates governs a switch from transport to mechanotransduction". For those reviewers who give their assent, we will be publishing their names alongside the published article.

Nature Cell Biology offers a Transparent Peer Review option for new original research manuscripts submitted after December 1st, 2019. As part of this initiative, we encourage our authors to support increased transparency into the peer review process by agreeing to have the reviewer comments, author rebuttal letters, and editorial decision letters published as a Supplementary item. When you submit your final files please clearly state in your cover letter whether or not you would like to participate in this initiative. Please note that failure to state your preference will result in delays in accepting your manuscript for publication.

Cover suggestions

As you prepare your final files we encourage you to consider whether you have any images or illustrations that may be appropriate for use on the cover of Nature Cell Biology.

Nature Cell Biology has now transitioned to a unified Rights Collection system which will allow our Author Services team to quickly and easily collect the rights and permissions required to publish your work. Approximately 10 days after your paper is formally accepted, you will receive an email in providing you with a link to complete the grant of rights. If your paper is eligible for Open Access, our Author Services team will also be in touch regarding any additional information that may be required to arrange payment for your article.

Please note that *Nature Cell Biology* is a Transformative Journal (TJ). Authors may publish their research with us through the traditional subscription access route or make their paper immediately open access through payment of an article-processing charge (APC). Authors will not be required to make a final decision about access to their article until it has been accepted. Find out more about Transformative Journals

Please use the following link for uploading these materials:
[Redacted]

Best regards,

Kendra Donahue
Staff
Nature Cell Biology

On behalf of

Daryl Jason Verzosa David, PhD

Senior Editor, Nature Cell Biology
Nature Portfolio

Heidelberger Platz 3, 14197 Berlin, Germany
Email: daryl.david@nature.com
ORCID: <https://orcid.org/0000-0002-9253-4805>

Reviewer #1:

Remarks to the Author:

While a large body of work suggests that biomolecular condensates ensuing from liquid-liquid phase separation mature into various material states, how this aging process is controlled and if the naive and mature phases can have differential functions is currently unknown. Using *Caenorhabditis elegans* as a model, the authors demonstrated that MEC-2 Stomatin undergoes a rigidity phase transition during maturation from fluid to viscoelastic, glass-like condensates that facilitate either transport or mechanotransduction.

I agree that that the authors' data demonstrate a novel function for rigidity maturation during mechanotransduction. The authors used various in vitro and in vivo approaches and provided numerous data to support their claims.

Through the 2nd round of review process, the authors thoroughly answered my concerns. Therefore, I support the acceptance of this manuscript to NCB.

Author Rebuttal, second revision:

We would like to thank the editor and the anonymous referees for their time and effort in providing fruitful comments and their dedication to the scientific process. We are grateful to them and acknowledge their contribution in helping to improve our scientific findings and their interpretation.

Reviewers' comments:

Reviewer #1 (Remarks to the Author):

While a large body of work suggests that biomolecular condensates ensuing from liquid-liquid phase separation mature into various material states, how this aging process is controlled and if the naive and mature phases can have differential functions is currently unknown. Using *Caenorhabditis elegans* as a model, the authors demonstrated that MEC-2 Stomatin undergoes a rigidity phase transition during maturation from fluid to viscoelastic, glass-like condensates that facilitate either transport or mechanotransduction.

I agree that the authors' data demonstrate a novel function for rigidity maturation during mechanotransduction. The authors used various *in vitro* and *in vivo* approaches and provided numerous data to support their claims.

Through the 2nd round of review process, the authors thoroughly answered my concerns. Therefore, I support the acceptance of this manuscript to NCB.

We appreciate the reviewer's thorough analysis and thank him/her for identifying points that indeed merit clarification in the manuscript. Thanks a lot for the support throughout this process.

Final Decision Letter:

Dear Dr Krieg,

I am pleased to inform you that your manuscript, "A MEC-2/Stomatin condensate liquid-to-solid phase transition controls neuronal mechanotransduction during touch sensing", has now been accepted for publication in Nature Cell Biology.

Please note that *Nature Cell Biology* is a Transformative Journal (TJ). Authors may publish their research with us through the traditional subscription access route or make their paper immediately open access through payment of an article-processing charge (APC). Authors will not be required to make a final decision about access to their article until it has been accepted. Find out more about Transformative Journals

Authors may need to take specific actions to achieve compliance with funder and

institutional open access mandates. If your research is supported by a funder that requires immediate open access (e.g. according to Plan S principles) then you should select the gold OA route, and we will direct you to the compliant route where possible. For authors selecting the subscription publication route, the journal's standard licensing terms will need to be accepted, including self-archiving policies. Those licensing terms will supersede any other terms that the author or any third party may assert apply to any version of the manuscript.

If you have not already done so, we strongly recommend that you upload the step-by-step protocols used in this manuscript to the Protocol Exchange (www.nature.com/protocolexchange), an open online resource established by Nature Protocols that allows researchers to share their detailed experimental know-how. All uploaded protocols are made freely available, assigned DOIs for ease of citation and are fully searchable through nature.com. Protocols and Nature Portfolio journal papers in which they are used can be linked to one another, and this link is clearly and prominently visible in the online versions of both papers. Authors who performed the specific experiments can act as primary authors for the Protocol as they will be best placed to share the methodology details, but the Corresponding Author of the present research paper should be included as one of the authors. By uploading your Protocols to Protocol Exchange, you are enabling researchers to more readily reproduce or adapt the methodology you use, as well as increasing the visibility of your protocols and papers. You can also establish a dedicated page to collect your lab Protocols. Further information can be found at www.nature.com/protocolexchange/about

With kind regards,

Daryl